

# Generalized Lieb-Schultz-Mattis theorem
# on bosonic symmetry protected topological phases

**Shenghan Jiang[1,2], Meng Cheng[3], Yang Qi[4,5,6⋆] and Yuan-Ming Lu[7]**

**1** Department of Physics and Institute for Quantum Information and Matter,
California Institute of Technology, Pasadena, California 91125, USA
**2** Kavli Institute for Theoretical Sciences, University of Chinese Academy of Sciences,
Beijing 100190, China
**3** Department of Physics, Yale University, New Haven, CT 06511-8499, USA
**4** Center for Field Theory and Particle Physics, Department of Physics,
Fudan University, Shanghai 200433, China
**5** State Key Laboratory of Surface Physics, Fudan University, Shanghai 200433, China
**6** Collaborative Innovation Center of Advanced Microstructures, Nanjing 210093, China
**7** Department of Physics, Ohio State University, Columbus, Ohio 43210, USA

⋆ qiyang@fudan.edu.cn

## Abstract

**We propose and prove a family of generalized Lieb-Schultz-Mattis (LSM) theorems for symmetry protected topological (SPT) phases on boson/spin models in any dimensions. The "conventional" LSM theorem, applicable to e.g. any translation invariant system with an odd number of spin-1/2 particles per unit cell, forbids a symmetric short-range-entangled ground state in such a system. Here we focus on systems with no LSM anomaly, where global/crystalline symmetries and fractional spins within the unit cell ensure that any symmetric SRE ground state must be a non-trivial SPT phase with anomalous boundary excitations. Depending on models, they can be either strong or "higher-order" crystalline SPT phases, characterized by non-trivial surface/hinge/corner states. Furthermore, given the symmetry group and the spatial assignment of fractional spins, we are able to determine all possible SPT phases for a symmetric ground state, using the real space construction for SPT phases based on the spectral sequence of cohomology theory. We provide examples in one, two and three spatial dimensions, and discuss possible physical realization of these SPT phases based on condensation of topological excitations in fractionalized phases.**

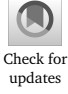

# 1   Introduction

Since the discovery of topological band insulators [1,2], symmetry-protected topological (SPT) phases have attracted considerable research interests both theoretically and experimentally [3–5]. Unlike conventional phases of matter within the Landau paradigm which are fully captured by their symmetries, SPT phases exhibit quantized bulk response functions as well as topological boundary excitations protected by symmetry. Examples of SPT phases include topological insulators and superconductors of weakly interacting electrons, which can be realized in semiconductors and other electronic materials [1, 2]. On the other hand, there also exists a large family of SPT phases in bosonic systems which necessitate strong interactions to be stabilized [3–5], such as 1D Haldane phase [6–8] and 2D bosonic integer quantum Hall states [9–11]. The requirement of strong and often complex interactions significantly hinders the physical realization of those phases. One nature question is: how to identify an interacting physical system that is likely to host these bosonic SPT phases?

Symmetries of a quantum many-body system, when combined with certain restrictions on representations on the many-body Hilbert space, can strongly constrain the nature of possible ground states of the (possibly interacting) system. One famous example is the Lieb-Schultz-Mattis (LSM) theorem [12], which forbids a symmetric gapped ground state in any translational-invariant one-dimensional (1D) spin-1/2 chain. The LSM theorem has been generalized to higher spatial dimensions and other symmetries, where certain space group symmetries (such as translation, mirror, and nonsymmorphic symmetries) forbid symmetric short-range-entangled (SRE) ground states in a generic interacting Hamiltonian within this many-body Hilbert space [13–23].

Recently, a new class of LSM-type theorems for SPT phases has been discovered [24–26], where magnetic translation symmetries combined with certain restrictions on the Hilbert space dictate that any symmetric SRE ground state must be a non-trivial (strong) SPT phase in two spatial dimensions (2D). In contrast to the conventional LSM theorems, here an SRE ground state is allowed, but it has to be a non-trivial SPT phase, i.e. with symmetry-protected topological excitations on its boundary.

In this work, we generalize these 2D results to other spatial dimensions and other internal/crystalline symmetries, where we coin these new theorems as "SPT-LSM" theorems. In a broader context of classifying SPT phases protected by both crystalline and internal symmetries, SPT-LSM theorems highlight an important feature in this problem, namely the classification depends on the precise manner in which the total symmetry group is represented in local (i.e. "on-site") Hilbert space. This is quite different from pure internal symmetry groups, where all one needs to know is the group structure.

The goal of this work is to provide a general framework to systematically construct and prove SPT-LSM theorems in an interacting quantum many-body system with both internal and

crystalline symmetries, together with certain restrictions on its local Hilbert space. For example, we consider the cases where a fractional spin is assigned to certain high-symmetry locations. This framework allows us to derive more SPT-LSM theorems with various symmetries in various spatial dimensions, which can guide the future search for physical realizations of SPT phases in strongly correlated systems.

The constraints on ground states in conventional LSM theorems can be understood more systematically in terms of matching 't Hooft anomalies, since these theorems can be interpreted using bulk-boundary correspondence of crystalline SPT phases in one dimension higher [19, 20]. In other words, the ground state of a lattice system satisfying the conditions of a certain LSM theorem must be able to "resolve" the anomaly. However, this interpretation does not directly carry over to SPT-LSM theorems discussed here, as the fact that a SRE ground state exists means the (fictitious) higher-dimensional bulk is topologically trivial. This difficulty calls for a new angle to systematically understand SPT-LSM theorems.

In this work, we propose a general approach to construct SPT-LSM theorems from the "conventional" LSM theorems. The procedure is based on the observation that all SPT-LSM theorems obtained so far require that lattice symmetries (e.g. translation symmetries [24–26]) are realized "projectively". One starts from a conventional LSM theorem, where the total symmetry group is a direct product of the lattice and internal symmetries. In such a system, one may realize a symmetric gauge theory. The LSM anomaly matching condition implies that symmetries have to be implemented non-trivially in the gauge theory. In particular, we will look for a realization where the gauge charge transforms projectively under the lattice symmetry (but linearly under the internal symmetry). Once such a gauge theory is identified, condensing gauge charges (binding with some symmetry charges) leads to a SPT phase and a candidate SPT-LSM system, where the gauge symmetry now becomes a global symmetry. We then prove this SPT-LSM theorem in a more rigorous approach, based on (i) a real-space construction of crystalline SPT phases [20, 27–31] and (ii) in certain cases entanglement-spectrum-based argument [18, 24] to find the precise constraints on the ground state. Using this method, we devise several new examples of SPT-LSM theorems in various dimensions, going beyond the known results in two ways. Firstly, we show by example how such theorems can be established with just point-group symmetries and internal symmetries (without translation symmetries). Secondly, we establish examples of SPT-LSM theorems for strong SPT phases in 3D. We also notice that the gauge-charge condensation picture approach can be used as a tool to guide the design of spin/boson Hamiltonians to realize SPT phases.

This paper is organized as follows. In Section 2, we warm up with a 1D example of SPT-LSM system enforced by a mirror reflection symmetry. An SPT phase is realized as the ground state of an exact solvable model. We discuss how various methods can be used to approach this problem, including entanglement-spectrum-based argument, decorated domain wall picture, group cohomology calculation as well as the real space construction. All these methods can be generalized to higher dimensions. In Section 3, we propose a general framework to approach and prove all SPT-LSM theorems, using real-space construction based on spectral sequence of cohomology theory. Examples of higher dimensional SPT-LSM systems are given in Section 4. Finally, in Section 5, we summarize and discuss future directions.

Technical details and more mathematical formulations are summarized in the Appendices. Here we highlight some appendices which may be interesting in their own right. Appendix B generalizes fixed point wavefunctions proposed in Ref. [32], described by the mathematical framework of equivariant cohomology, which gives a classification and construction of bosonic topological crystalline phases. Appendix C presents an algorithm to calculate bosonic SPT phases in a SPT-LSM system based on equivariant cohomology and we apply this algorithm to compute examples given in Section 4. Appendix D give an overview on how to obtain SPT phases by condensing topological excitations in fractionalized phases.

## 2 A 1D example

In this section, we provide a simple 1D spin chain system as an example for SPT-LSM systems. We study this 1D example from various points of view:

1. We provide an exact solvable model, which realize the non-trivial SPT phases with dangling fractional spins at the open edge.

2. Based on entanglement properties for a generic symmetric SRE phase in 1d, we then argue that the non-trivial SPT phase persists beyond the exact solvable model. In fact, gapped quantum phases supported by the 1D spin chain system (with the specific symmetries and local Hilbert space) either spontaneously break symmetry, or if symmetric, must be non-trivial SPT phases. We call such systems that enforce non-trivial SPT phases as SPT-LSM systems.

3. We show that the symmetry enforced SPT phase can also be understood from decorated domain wall picture: by condensing domain walls from a specific spontaneously symmetry breaking (SSB) phase, we either obtain another SSB phase, or get a non-trivial SPT phase.

4. We perform some calculations based on group cohomology, which is related to the general framework of real space construction for the non-trivial SPT phase. It is also shown that the non-trivial SPT phase presented here is beyond the conventional cohomological classification of bosonic crystalline phases obtained in Ref. [33, 34].

Various methods applied to this simple 1D example can all be generalized to higher dimensions and other symmetry groups. We will give a general framework and examples in higher dimensions of SPT-LSM systems in the next few sections.

### 2.1 Models and symmetry

Let us consider a 1D chain system, where the local Hilbert space is isomorphic to $\mathbb{C}^{16} \sim \mathbb{C}^4 \otimes \mathbb{C}^4$. The 16 basis states are labeled as $|\alpha, \beta\rangle$, with $\alpha, \beta = 0, 1, 2, 3$.

This system hosts global $Z_4^g \times Z_4^h$ symmetry, whose generators are labeled as $g$ and $h$ respectively. Meanwhile, it also preserves translation symmetry $T_{2x}$ and mirror reflection symmetry $\sigma$, as shown in Fig. 1. Local Hilbert space consists of one fractional spin on each reflection axis (i.e. each site in Fig. 1), each forming a projective representation of the $Z_4^g \times Z_4^h$ symmetry group.

There are four inequivalent projective representations of $Z_4^g \times Z_4^h$, which are classified by the second cohomology group: $H^2[Z_4^g \times Z_4^h, U(1)] = Z_4$. These four inequivalent projective representation are characterized by commutation relation between $W_g$ and $W_h$, with $W$ labeling the representation:

$$W_g^4 = W_h^4 = 1, \quad W_g W_h = i^\eta \cdot W_h W_g, \tag{1}$$

where $\eta = 0, 1, 2, 3$.

In our setup, there are two sites (and hence two reflection axes) per unit cell, and the Hilbert space on each site forms a $\eta = 2$ projective representation. Crucially, we require that the system has the site-centered inversion symmetry $\sigma$. Note that fractional spins are not well defined without $\sigma$, as we can always fuse two sites to get an integer spin.

To be more concrete, we introduce $4 \times 4$ matrix $\mu$'s and $\nu$'s, which acts as

$$\mu^z|\alpha, \beta\rangle = i^\alpha|\alpha, \beta\rangle, \quad \mu^x|\alpha, \beta\rangle = |[\alpha + 1], \beta\rangle;$$
$$\nu^z|\alpha, \beta\rangle = i^\beta|\alpha, \beta\rangle, \quad \nu^x|\alpha, \beta\rangle = |\alpha, [\beta + 1]\rangle, \tag{2}$$

where $[\alpha] \equiv \alpha \bmod 4$. In particular, we have $\mu^z \mu^x = i \mu^x \mu^z$ as well as $\nu^z \nu^x = i \nu^x \nu^z$.

The onsite symmetry action can be expressed as tensor product of unitary operators on local Hilbert space, which reads

$$
\begin{aligned}
W_g(j) &= \mu_j^x \otimes \nu_j^x\,; \\
W_h(j) &= \begin{cases} \mu_j^z \otimes \nu_j^z\,, & j \text{ even} \\ (\mu_j^z)^3 \otimes (\nu_j^z)^3\,, & j \text{ odd}\,. \end{cases}
\end{aligned}
\tag{3}
$$

Notice that $W_g$ and $W_h$ anti-commute on a single site $j$:

$$
\begin{aligned}
W_g(j)^4 &= W_h(j)^4 = 1\,, \\
W_g(j)W_h(j) &= -W_h(j)W_g(j)\,.
\end{aligned}
\tag{4}
$$

Compared with Eq. (7), we conclude that $Z_4^g \times Z_4^h$ acts projectively on the local Hilbert space with $\eta = 2$.

We also list action of lattice symmetries, including site-center reflection $\sigma$ and two lattice-spacing translation $T_{2x}$:

$$
\begin{aligned}
\sigma &: \mu_j \to \nu_{-j}\,, \quad \nu_j \to \mu_{-j}\,; \\
T_{2x} &: \mu_j \to \mu_{j+2}\,, \quad \nu_j \to \nu_{j+2}\,,
\end{aligned}
\tag{5}
$$

where we ignore the $z/x$ superscripts of $\mu_j(\nu_j)$ for brevity. Notice that a unit cell consists of two sites, so in total transforms linearly under $Z_4^g \times Z_4^h$. The symmetry group and local Hilbert space are presented in Fig. 1.

Now, let us propose an exact solvable model defined by summation of commuting projectors, which reads

$$
H = \sum_j (1 - P_{j+1/2}^x P_{j+1/2}^z)\,,
\tag{6}
$$

where

$$
\begin{aligned}
P_{j+1/2}^x &= \frac{1}{4}\left(1 + \nu_j^x \mu_{j+1}^x + (\nu_j^x \mu_{j+1}^x)^2 + (\nu_j^x \mu_{j+1}^x)^3\right)\,, \\
P_{j+1/2}^z &= \frac{1}{4}\left(1 + \nu_j^z (\mu_{j+1}^z)^3 + (\nu_j^z \mu_{j+1}^z)^2 + (\nu_j^z)^3 \mu_{j+1}^z\right)\,.
\end{aligned}
\tag{7}
$$

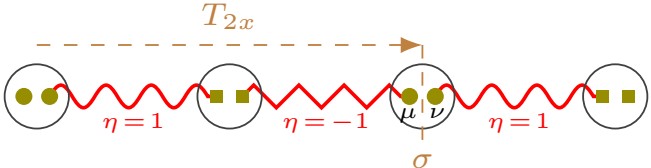

Figure 1: The 1D chain system, with local Hilbert space isomorphic to $\mathbb{C}^4 \otimes \mathbb{C}^4$. The onsite symmetry group for this system is $Z_4^g \times Z_4^h$, and act projectively on each site, as shown in Eq. (3). And the lattice symmetries include translational symmetry with two lattice spacing $T_{2x}$, as well as reflection along site center $\sigma$. Ground state for Hamiltonian in Eq. (6) is AKLT-like, which hosts non-trivial edge states.

The local projector $1 - P_{j+1/2}^z P_{j+1/2}^x$ projects out a single state of Hilbert space at $j$ and $j+1$ as

$$
\left[1 - P_{j+1/2}^x P_{j+1/2}^z\right]\left(\sum_{a=0}^3 |\nu_j^z = a, \mu_{j+1}^z = a\rangle\right) = 0\,.
\tag{8}
$$

Thus, there is a unique zero-energy ground state in systems with periodic boundary condition, which reads

$$|\Psi\rangle = \bigotimes_j \left[ \sum_{a=0}^{3} |v_j^z = a, \mu_{j+1}^z = a\rangle \right]. \tag{9}$$

It is easy to check that all other excited states is separated by a finite energy gap.

Since there is no degeneracy in the ground state manifold, we conclude that the ground state has no long-range order. In other words, it must be a symmetric SRE phase.

Next we show this ground state belongs to a non-trivial SPT phase. To see this, we put the system on an open boundary system with $2n$ sites, and then the Hamiltonian reads

$$H_{open} = \sum_{j=0}^{2n-1} (1 - P_{j+1/2}^x P_{j+1/2}^z). \tag{10}$$

We point out that although this open chain system breaks both $T_{2x}$ and $\sigma$ symmetry, it shares the same bulk properties as the closed chain system, since the open chain construction is derived from the closed chain. Thus, it is reasonable to study boundary modes of the open chain as a manifestation of its bulk properties.

It is straightforward to see that there are dangling spins on boundaries: $\mu$ spin on left edge and $v$ spin on right edge. Due to these dangling spins, the ground state degeneracy on the open boundary system equals $16 = 4 \times 4$. These dangling spins are actually protected by $Z_4^g \times Z_4^h$ symmetry. To see this, we identify boundary symmetry operators as

$$\begin{aligned} W_g^L = \mu^x, \quad W_h^L = \mu^z; \\ W_g^R = v^x, \quad W_h^R = v^z. \end{aligned} \tag{11}$$

Thus, the $W^{L/R}$ forms $\eta = -1$ projective representation of $Z_4^g \times Z_4^h$. And due to spatial separation, local symmetric perturbations only lead to exponentially small splitting of the degeneracy.

As shown in Fig. 1, the ground state construction is similar as the famous AKLT wavefunction [35]: an $\eta = 2$ local state is decomposed to two $\eta = 1(-1)$ states, and the $\eta = 1$ state from the left/right site pairs with the $\eta = -1$ state from the right/left site. Consequently, if one puts the system on an open boundary, there will be an unpaired $\eta = \pm 1$ zero mode at each boundary, where $\pm$ sign depends on the even/oddness of the edge site.

We point out a key difference of this 1D system from the spin-one system. In a $S = 1$ spin chain with $SO(3)$ symmetry, depending on the interaction, one may either obtain a non-trivial SPT phase or obtain a trivial symmetric phase. The non-trivial SPT phase has AKLT-type wavefunction, while the trivial symmetric phase can be constructed by splitting local spin-one into two integer spins and then pair spins in one bond. However, for the system discussed above, the only splitting consistent with the site-centered inversion is to split an $\eta = 2$ representation into two $\eta = \pm 1$, which always yields a non-trivial SPT phase. We will argue that this result is in fact completely general and holds beyond the exact solvable models.

## 2.2 Entanglement based argument

By analyzing entanglement spectrum, We now argue that gapped symmetric ground states for this system must be non-trivial SPT phases.

Given a quantum many-body wavefunction $|\psi\rangle$ on a 1D chains, one can "cut" the system into two halves along bond $(j, j + 1)$ and do Schmidt decomposition as

$$|\psi\rangle = \sum_{\alpha=1}^{N} \lambda_\alpha |\phi_\alpha^L\rangle \otimes |\phi_\alpha^R\rangle, \tag{12}$$

where $N$ is named as Schmidt rank, and $\lambda$'s are positive number characterizing entanglement between left and right parts. Here, we require $\lambda_1 \geq \lambda_2 \geq \cdots \geq \lambda_N$. $|\phi_\alpha^{L/R}\rangle$ are left/right entanglement states (respect to bond $(j, j+1)$). If $|\psi\rangle$ describes gapped symmetric ground state, the entanglement spectrum, defined as $\{-\ln \lambda_\alpha^2 \mid \alpha = 1, 2, \cdots\}$, is also gapped in the thermodynamic limit of a 1D system.

In general, entanglement states transform projectively under the on-site symmetry group. Namely, for $g \in SG_{onsite}$,

$$U_g^{L/R}|\phi_\alpha^{L/R}\rangle = \sum_\beta (V_g^{L/R})_{\alpha\beta}|\phi_\beta^{L/R}\rangle, \tag{13}$$

where $U_g^{L/R}$ is the physical symmetry action restricted in the left/right system while $V_g^{L/R}$ is an $N \times N$ unitary matrix, which forms a projective representation of $SG_{\text{onsite}}$.

For $Z_4^g \times Z_4^h$ group, according to Eq. (1), we can express projective representation of $V_g^{L/R}$ as

$$V_g^{L/R} \cdot V_h^{L/R} = \exp\left(i\frac{\pi}{2} \cdot \eta_{j+1/2}^{L/R}\right) V_h^{L/R} \cdot V_g^{L/R}, \tag{14}$$

where $\eta_{j+1/2}^{L/R} \in \{0, 1, 2, 3\}$. One may wonder if $V^{L/R}$ can be divided to several blocks, where different block represents different projective representations. However, states with blocked diagonalized $V^{L/R}$ are in fact cat states, and thus spontaneously break symmetries [36].

$Z_4^g \times Z_4^h$ global symmetry requires

$$\eta_{j+1/2}^R + \eta_{j+1/2}^L = 0 \bmod 4. \tag{15}$$

Suppose we move the entanglement cut by one site, we expect that

$$\eta_{j+1/2}^L = \eta_{j-1/2}^L + \eta_0, \tag{16}$$

where $\eta_0$ labels site projective representation. This is because the Schmidt states at the two cuts are related by adding the intervening sites, and we also assume that the ground state is short-range correlated.

Reflection symmetry along site $j$ requires

$$\eta_{j-1/2}^R = \eta_{j+1/2}^L. \tag{17}$$

By solving above equations, we find $2\eta_{j+1/2}^L + \eta_0 = 0$. The existence of a symmetric gapped ground state means that this equation is solvable, so $\eta_0$ must be even, leaving two possibilities $\eta_0 = 0, 2$. For $\eta_0 = 0$, one finds a trivial solution $\eta^{L/R} = 0$, corresponding to a trivial gapped phase.

For $\eta_0 = 2$ which is the system we have in hand, the solutions are $\eta_{2j-1/2}^{L/R} = \eta_{2j+1/2}^{R/L} = \pm 1$. The nonzero $\eta^{L/R}$ also indicates edge modes transform projectively under $Z_4^g \times Z_4^h$ symmetry, and the symmetric phases obtained must be non-trivial SPT phases.

Lastly, if $\eta_0$ is 1 or 3, there is no integer solution for $\eta^{L/R}$, meaning that a gapped symmetric state is impossible. Indeed, in this case, in each unit cell, there is non-trivial projective representation with $\eta = 2$, satisfying the condition of a conventional LSM theorem and excluding gapped symmetric phases.

## 2.3 SPT phases from domain wall condensation

In the following, we will construct possible symmetric phases of this systems by considering condensation of domain walls.

First, let us perform the $Z_4$ version of Kramers-Wannier duality. We define

$$\tau^x_{j+1/2} = (\mu^z_j)^\dagger \mu^z_{j+1}, \quad (\tau^z_{j-1/2})^\dagger \tau^z_{j+1/2} = \mu^x_j \nu^x_j. \tag{18}$$

Here, $\tau^z_{j+1/2}$ can be expressed as string operator: $\tau^z_{j+1/2} = \prod_{j' \le j} \mu^x_{j'} \nu^x_{j'}$. We have commutation relation: $\tau^z_{j+1/2} \tau^x_{j+1/2} = i \tau^x_{j+1/2} \tau^z_{j+1/2}$. $\tau^x_{j+1/2}$ can be interpreted as measuring $g$-domain wall at bond $(j, j+1)$, and $\tau^z_{j+1/2}$ is the $g$-domain wall creation operator.

Using the mapping defined in Eq. (18), we are able to work out the symmetry action on $g$ domain wall:

$$\begin{aligned}
g &: \tau^z_{j+1/2} \to \tau^z_{j+1/2}, \\
h &: \tau^z_{j+1/2} \to (-)^j \tau^z_{j+1/2}, \\
\sigma &: \tau^z_{j+1/2} \to (\tau^z_{-j-1/2})^\dagger, \\
T_{2x} &: \tau^z_{j+1/2} \to \tau^z_{j+2+1/2}.
\end{aligned} \tag{19}$$

Besides, there is an additional $\widetilde{Z^g_4}$ "$g$-domain wall conservation" symmetry, which is generated by $\widetilde{g} = \prod_j \tau^x_{j+1/2}$, and acts as

$$\widetilde{g} : \tau^z_{j+1/2} \to -i \tau^z_{j+1/2}. \tag{20}$$

We point out that any local operator should transform trivially under $\widetilde{g}$.

We are interested in symmetric phases, which are obtained by condensation phases of $g$-domain walls while preserving all other symmetries. In the following, we will show that naive condensation pattern of domain walls always break global symmetries.

To see this, we notice that when acting on domain walls, $h$ and $\sigma$ have non-trivial commutation relation:

$$h\sigma \circ \tau^z = -\sigma h \circ \tau^z = \widetilde{g}^2 \sigma h \circ \tau^z. \tag{21}$$

We claim that this non-trivial commutation relation forbids a symmetric phase by condensing $\tau^z$ variables. Actually $\tau^z$ condensed phases either break $Z^h_4$ symmetry or break $\sigma$ symmetry.

For example, let us consider the uniform condensation pattern of $\tau^z$: $\langle \tau^z_{j+1/2} \rangle \neq 0$ which is independent of $j$. Since domain walls are non-local variables, to determine the symmetry breaking pattern, we should consider representation of local observables formed by $\tau^z$'s, which can be expressed as $(\tau^z_{j+1/2})^\dagger \tau^z_{k+1/2} + h.c..$ Under $Z^h_4$ transformation, local operators such as $(\tau^z_{j-1/2})^\dagger \tau^z_{j+1/2} + h.c.$ obtains minus sign, and thus carries double $Z^h_4$ charge. So, this uniform condensation phase breaks $Z^h_4$ symmetry down to $Z_2$.

On the other hand, we may only condense even bond $\tau^z$: $\langle \tau^z_{2j+1/2} \rangle \neq 0$, $\langle \tau^z_{2j-1/2} \rangle = 0$. In this case, $Z^h_4$ is preserved, since all local operators with nonzero expectation value, which are $\tau^z_{2j+1/2} \tau^z_{2k+1/2}$, is a $Z^h_4$ singlet. However, this condensation pattern breaks $\sigma$ symmetry, since $\sigma$ maps even bond $\tau^z$ to odd bond $\tau^z$.

To get a fully symmetric phase, the key step is to condense the bound state of $g$ domain wall and $h$-charge $\mu^x$, $\nu^x$. In particular, the composite of domain wall and $h$-charge can be chosen as

$$\tilde{\tau}^z_{2j-1/2} \equiv \tau^z_{2j-1/2}(\nu^x_{2j})^\dagger = \mu^x_{2j} \tau^z_{2j+1/2}. \tag{22}$$

One can easily verify the following symmetry transformation rules for the composite $\tilde{\tau}^z$:

$$g : \tilde{\tau}^z_{2j-1/2} \to \tilde{\tau}^z_{2j-1/2},$$
$$h : \tilde{\tau}^z_{2j-1/2} \to (-)^j \mathrm{i}\, \tilde{\tau}^z_{2j-1/2},$$
$$\sigma : \tilde{\tau}^z_{2j-1/2} \to (\tilde{\tau}^z_{-2j+1/2})^\dagger,$$
$$T_{2x} : \tilde{\tau}^z_{2j-1/2} \to \tilde{\tau}^z_{2j+2-1/2}. \tag{23}$$

One can easily verify that $[h, \sigma] = 0$ when acting on the composite object $\tilde{\tau}^z_{2j-1/2/}$ of a $g$-domain wall and $h$ gauge charge, and therefore when it condenses $\langle \tilde{\tau}^z_{2j-1/2} \rangle \neq 0$, all local observables with non-zero expectation values are singlets under global symmetry action, which makes the condensing phase symmetric. Furthermore, according to Ref. [32, 37] and discussion in Appendix D.1, condensation of bound states of domain walls and symmetry charges leads to non-trivial SPT phases.

From domain wall condensation picture, one also obtain the possible phases for other $\eta_0$. For example, when $\eta_0 = 1$, action of $h$ on domain walls becomes

$$h : \tau^z_{j+1/2} \to \mathrm{i}^j \tau^z_{j+1/2}. \tag{24}$$

Thus, commutator between $h$ and $\sigma$ reads

$$h\sigma \circ \tau^z = \mathrm{i}\, \sigma h \circ \tau^z = \tilde{g}^3 \sigma h \circ \tau^z. \tag{25}$$

In this case, domain walls transform projectively under symmetry actions, and thus condensing domain walls always breaks symmetry. On the other hand, for the case with $\eta_0 = 0$, we have $h\sigma \circ \tau^z = \sigma h \circ \tau^z$. One can safely condense "bare" domain wall $\tau^z$ without breaking any symmetry, which gives the trivial SPT phase.

We mention that in fact that $g$-domain walls should be viewed as $\widetilde{Z^g_4}$ gauge charges, and the SPT phase is obtained by condensing bound states of gauge charges and symmetry charges. This point of view can be easily generalized to higher dimensions, and serves as a useful tool to construct generic SPT-LSM systems.

## 2.4 Group cohomology calculation

As shown in Ref. [33, 34], the classification of bosonic crystalline SPT phases with global symmetry $SG$ is the same as the classification of bosonic SPT phases with internal symmetry group $SG$ (i.e the same abstract group structure), as long as orientation reversing spatial symmetries are treated as antiunitary onsite symmetries. This is known as the "crystalline equivalence principle".

1D bosonic onsite SPT phases is classified by the second group cohomology $H^2[SG, U(1)_{\mathcal{T}}]$ [16, 38]. According to the "crystalline equivalence principle", SPT phases protected by onsite $Z^g_4 \times Z^h_4$ and reflection symmetry $Z^\sigma_2$ are classified by $H^2[Z^g_4 \times Z^h_4 \times Z^\sigma_2, U(1)_\sigma]$, where $\sigma$ act non-trivially on $U(1)$ coefficient[1]. This cohomological group can be calculated using Künneth formula [9, 19, 39] as

$$
\begin{aligned}
H^2[Z^g_4 \times Z^h_4 \times Z^\sigma_2, U(1)_\sigma] &= H^3[Z^g_4 \times Z^h_4 \times Z^\sigma_2, \mathbb{Z}_\sigma] \\
&= \prod_{p=0}^3 H^{3-p}[Z^\sigma_2, (H^p[Z^g_4 \times Z^h_4, \mathbb{Z}])_\sigma] \\
&= H^3[Z^\sigma_2, \mathbb{Z}_\sigma] \times H^0[Z^\sigma_2, (Z_4)_\sigma] = Z_2 \times Z_2, \tag{26}
\end{aligned}
$$

---

[1]We ignore translation symmetry $T_{2x}$ here, since the SPT phase we considered is still non-trivial even if we break $T_{2x}$ symmetry

where $M_\sigma$ denotes an Abelian group $M$ equipped with the non-trivial action of $\sigma$. And in the first line, we use $H^n[SG, U(1)] = H^{n+1}[SG, Z]$ for $n > 0$ and compact $SG$.

Generators of these two $Z_2$, labeled as $\nu_1 = 1$ and $\nu_2 = 1$ respectively, give two root SPT phases. $\nu_1 = 1$ corresponds to reflection protected Haldane phases in 1+1D, while $\nu_2 = 1$ is the SPT phase protected by $Z_4^g \times Z_4^h$ with $\eta = 2$ projective representation as edge states.

Clearly, the SPT phase we found in the 1D SPT-LSM system is beyond this classification. The reason is that the classification for crystalline SPT phases developed in Ref. [33, 34] actually makes a hidden assumption: global (i.e. onsite) symmetries act linearly on the local Hilbert space. When the local Hilbert space forms a projective representation of the symmetry group, one needs to develop a new framework to classify possible SPT phases. We will answer this question in the next few sections.

## 2.5 Real-space construction

Here, we present an algebraic calculation to capture the non-trivial SPT phases in this section. The input data for this calculation includes the lattice structure, the global symmetry group and projective representation of local Hilbert space. And the output data give us possible SPT phases supported by this system.

We use the idea of real space construction [20, 30, 40] to obtain possible SPT phases. To proceed, we assume that the correlation length $\xi$ is much smaller than the size of unit cell $a$, and thus it is meaningful to talk about decoration of gapped phases within a unit cell. In this kind of systems, the local Hilbert space is identified as effective low-energy degree of freedom in a unit cell. Although this assumption might not be true for most systems, we expect the classification of SPT phases remains to be true even if the assumption fails.

We then focus on a given bond, say bond $[0, 1]$, and decorate it with some SPT phase. The invariant symmetry group for bond $[0, 1]$ is $Z_4^g \times Z_4^h$, while lattice symmetry $\sigma$ and $T_{2x}$ maps bond $[0, 1]$ to other bonds. So the decoration of SPT phases on bond $[0, 1]$ is characterized by $\eta_{[0,1]} \in H^2[Z_4^g \times Z_4^h, U(1)] = Z_4$, where $\eta_{[0,1]} \in \{0, 1, 2, 3\}$. And edge states of this phase transform projectively under $Z_4^g \times Z_4^h$: the projective representation of left edge state is labeled by $\eta_{[0,1]}$ while that of right edge state is labeled by $-\eta_{[0,1]}$.

Decorations on other bonds are related with decoration on bond $[0, 1]$ by lattice symmetries. And the SPT phase of this system is constructed by decorations on all bonds, which is determined by decoration on bond $[0, 1]$.

In particular, decorations on bond $[-1, 0]$, labeled as $\eta_{[-1,0]}$, is related to $\eta_{[0,1]}$ by the following relation

$$\eta_{[-1,0]} = \sigma \circ \eta_{[0,1]} = -\eta_{[0,1]} \bmod 4, \tag{27}$$

where the minus sign comes from non-trivial action of $\sigma$.

Bond $[0, 1]$ and bond $[-1, 0]$ share a common edge, which is site 0. The projective representation of local Hilbert space at site 0 is determined by right edge of $[-1, 0]$ and left edge of $[0, 1]$ as

$$\eta_0 = -\eta_{[-1,0]} + \eta_{[0,1]} = 2\eta_{[0,1]} \quad \bmod 4. \tag{28}$$

For our system, $\eta_0 = 2$, and thus $\eta_{[0,1]} = -\eta_{[-1,0]} = \pm 1$. Decorations on other bonds are obtained by action of $T_{2x}$:

$$\eta_{[2n,2n+1]} = T_{2x}^n \circ \eta_{[0,1]} = \eta_{[0,1]}(= -\eta_{[-2n-1,2n]}). \tag{29}$$

Thus, the gapped SPT phase on this system can be constructed by the bond decoration, as shown in Fig. 1.

It is straightforward to check that projective representation at site $n$ is

$$\eta_n = -\eta_{[n-1,n]} + \eta_{[n,n+1]} = 2,\qquad(30)$$

which is consistent with the input data.

We mention that, by adding SPT phases obtained in Eq. (26) to the above decoration, we still have a valid decoration for the 1D spin chain system. In fact, phase labeled by $\eta_{[0,1]} = 1$ and $\eta_{[0,1]} = -1$ are related by the root phase of the second $Z_2$ in Eq. (26). Therefore, we still get four phases consistent with Eq. (26), but now the classification should be understood as a torsor over $\mathbb{Z}_2 \times \mathbb{Z}_2$. Here torsor emphasizes the fact that none of the phase can be regarded as the "identity" since they are all non-trivial (in the group cohomology classification the trivial product state is the identity ), but the difference of any two phases is a proper element of the group. This is in fact a common feature for SPT-LSM systems.

## 3  General framework for SPT-LSM systems – real space constructions

In this section, we present a general framework for SPT-LSM systems: given the global symmetry group and its action on local Hilbert space for a given system, we are able to identify whether this system is an SPT-LSM system or not. And for SPT-LSM systems, we can classify and construct all possible SPT phases. This framework is based on the real space construction [20, 27, 28, 30], which is a high-dimensional generalization of method used in Section (2.5).

The real space construction has a layered structure, which is related to the recently emerging concept of "high-order SPT phases" [41–47]. A $d$-dimensional SPT phase is called "$n$th order" if the codimension of protected boundary states equals $n$, while boundary states in lower codimensions can be gapped preserving symmetries. SPT phases protected by onsite symmetry, which host $d-1$ dimensional gapless edge states, are named as first order SPT in this language. Second order SPT phases in $d = 3$ supports "hinge states" on surfaces. From the point of view for real space construction, an $n$th order SPT ground state can be deformed into an assemble of block states with dimensions less than $d-n+1$. For example, $d$th order SPT phases, which host degenerate states at high symmetry points on the boundary, can be constructed using coupled 1D SPT phases.

Accordingly, an SPT-LSM system is called $n$th order if it allows $n$th order SPT phase but disallows $(n+1)$th order SPT phase (the $(d+1)$th order SPT phase is identified as the trivial symmetric phase). Notice that it is possible that systems with non-trivial projective representations do not allow any symmetric SRE phases, and this kind of systems are "conventional" LSM systems, which require long-range entangled phases as gapped symmetric ground states.

This section is divided to two parts. In the first part, according to action of global symmetries, we give a classification of local Hilbert space structures for a given system and symmetry group. In the second part, we start by presenting a physical picture for real space constructions, and based on this picture, we give an algorithm to classify/construct symmetric SRE phases for a given SPT-LSM system. A more mathematical treatment based on exact solvable models and spectral sequence of equivariant cohomology is presented in Appendix B and C.

### 3.1  The global symmetry group and local Hilbert spaces

Naively, one may think $SG$ contains enough input data for the purpose of classification of SPT phases. It is indeed true if $SG$ is onsite symmetry group. In this case, various mathematical tools are proposed to classify SPT phases, including group cohomology [9], cobordism

theory [48], generalized cohomological theory [49, 50], etc. An introduction to group cohomology and classification/construction of bosonic SPT phases protected by onsite symmetries is presented in Appendix A.

When $SG$ contains lattice symmetries, the classification of SPT phases (or topological crystalline phases) is enriched, and becomes more complicated. There are basically two approaches to classify/construct topological crystalline phases. On one hand, it was argued that one should treat spatial symmetries and onsite symmetries on the same footing, and the classification of SPT phases is given by group cohomology $H^{d+1}[SG, U(1)_{P\mathcal{T}}]$, with time reversal and orientation reversing lattice symmetries act non-trivially on $U(1)$ coefficient [33, 34].[2] On the other hand, a more physical way to understand SPT phases protected by both spatial and onsite symmetries is to construct these SPT phases by real space block states [27, 31, 40], which provides construction of SPT phases by decorating high-symmetry points, lines and planes with lower dimensional strong SPT phases protected by onsite symmetries. In recent works, it has been shown that this real-space construction is in one-to-one correspondence to classes in $H^{d+1}[SG, U(1)_{P\mathcal{T}}]$ [28, 30].

It is worth mentioning that when deriving the above classification result, one actually takes a hidden assumption: local Hilbert spaces are linear representations of their little groups. In general, symmetries can act projectively on local Hilbert spaces. For example, consider 1D spin chains with $SO(3)$ spin rotation symmetries. While $SO(3)$ group acts linearly on the spin-1 chain, it acts projectively on the spin-1/2 chain. In the following, for brevity, we will use the words local Hilbert spaces and spins interchangeably. And Hilbert spaces which transform as linear/projective representations are also referred as integer/fractional spins.

In the absence of lattice symmetry, for fractional spin systems, we can always group several fractional spins to form integer spins. With this coarse-graining procedure, the classification results for onsite symmetry SPT phases are the same for integer and fractional spin systems. However when we include lattice symmetries, the coarse-graining procedure may break lattice symmetry, and is hence disallowed. Therefore, in general, fractional and integer spin systems need to be treated separately and the resulting classifications can be very different. We have already seen the 1D example in Section 2. Here, we consider a more well-known example of translational symmetric spin chains.

There are two kinds of translational symmetric spin chains, with an integer or half-integer spin per unit cell. They share the same symmetry group: $SO(3) \times \mathbb{Z}$ where $\mathbb{Z}$ counts for the translation group. Yet possible symmetric phases in these two systems are completely different, as first pointed out by Haldane [7]. For integer spin chains, there are two symmetric gapped phases, one trivial symmetric phase and the other is the Haldane phase. For half-integer spin chains, the famous LSM theorem forbids any gapped symmetric phases, and the ground state must be either gapless or breaks translational symmetry by forming valence bond solid order.

From the above discussion, we learn that phases realized on fractional spin systems are in general quite different from those realized on integer spin systems. Moreover, there may be more than one type of fractional spin systems, and the classifications of phases on different fractional spin systems may be distinct from each other. Therefore to classify SPT phases on fractional spin systems, the first step is to classify/characterize different fractional spins for a given symmetry group $SG$.

Let us consider an arbitrary lattice system, where spins live on site: for site $i$, the corresponding local Hilbert space is labeled as $\mathcal{H}_i$. We define $SG_i$ as the little group of $\mathcal{H}_i$, which is the maximal subgroup of $SG$ mapping $\mathcal{H}_i$ to itself. Onsite symmetry group, labeled as $SG_{\text{onsite}}$, is a normal subgroup of $SG_i$. In general, $\mathcal{H}_i$ forms a projective representation of $SG_i$, which is classified by the second group cohomology $H^2[SG_i, U(1)_{\mathcal{T}}]$, where $U(1)_{\mathcal{T}}$ denotes the non-

---

[2]There are crystalline SPT phases beyond group cohomological classification [27, 51]. We will not focus on those phases here.

trivial action of antiunitary action on $U(1)$ coefficient (see Appendix A.1 for details).

If $\mathcal{H}_i$ is mapped to $\mathcal{H}_j$ by lattice symmetry action $g_0$, projective representation of these two Hilbert spaces are related. First, $SG_i$ and $SG_j$ are isomorphic to each other by the following outer automorphism map:

$$SG_j = g_0 \cdot SG_i \cdot g_0^{-1}. \tag{31}$$

To figure out how these two projective representations are related, let us write down lattice symmetry $g_0$ action on $\mathcal{H}_i$ explicitly:

$$g_0 |\phi_a\rangle_i = \sum_b [V(g_0)]_{ab} |\phi_b\rangle_j. \tag{32}$$

Here, $\{|\phi_a\rangle_i | a = 1, \ldots, dim(\mathcal{H}_i)\}$ is an orthonormal basis of $\mathcal{H}_i$ and $V(g_0)$ is some unitary matrix. We label $U_{i/j}$ as the projective representation of $SG_{i/j}$. Then, we have

$$g_0 g_i g_0^{-1} |\phi_a\rangle_j = [U_j(g_0 g_i g_0^{-1})]_{ab} |\phi_b\rangle_j [V(g_0)^{-1} \cdot {}^{g_0}U_i(g_i) \cdot V(g_0)]_{ab} |\phi_b\rangle_j, \tag{33}$$

where $g_i \in SG_i$ and $g_0 g_i g_0^{-1} \in SG_j$. ${}^{g_0}U_i(g_i)$ equals to $U_i(g_i)/[U_i(g_i)]^*$ if $g_0$ is an unitary/antiunitary symmetry

Therefore for $g_{i_1}, g_{i_2} \in SG_i$, according to the definition of projective representation, we have

$$U_i(g_{i_1}) \cdot U_i(g_{i_2}) = \eta_i(g_{i_1}, g_{i_2}) U_i(g_{i_1} g_{i_2}), \tag{34}$$

where $\eta_i(g_1, g_2)$ is an $U(1)$ phase which satisfies two cocycle condition. Thus, according to Eq. 33, we have

$$\eta_j(g_0 g_{i_1} g_0^{-1}, g_0 g_{i_2} g_0^{-1}) = [V(g_0)]^{-1} \cdot {}^{g_0}\eta_i(g_{i_1}, g_{i_2}) \cdot V(g_0) = {}^{g_0}\eta_i(g_{i_1}, g_{i_2}), \tag{35}$$

where ${}^{g_0}\eta_i = \eta_i/\eta_i^*$ if $g_0$ is a lattice transformation together with a unitary/antiunitary action.

We divide local Hilbert space to several groups according to lattice symmetries: $H_i$ and $H_j$ are in the same group if and only if they are related by lattice symmetry. We pick up one site in each group, and denote this representative set as $S$. Then, the projective representation for this system is classified by

$$\bigoplus_{i \in S} H^2[SG_i, U(1)_{\mathcal{T}}]. \tag{36}$$

And the projective representation of local Hilbert space within one group can be determined by Eq. (35).

From the above discussion, we are able to obtain classification of fractional spins for a given lattice. However, for a given symmetry group, there exist infinite many lattices, which give rise to infinite classes of fractional spins. For example, for translational symmetric systems, the number of sites within one unit cell can be any positive integer. Projective representations on sites within one unit cell can be chosen independently, leading to many different classes of fractional spins. This unphysical situation can be fixed by grouping all sites within one unit cell together, and consider the total projective representation after fusing these spins. Notice that we are not allowed to group sites beyond one unit cell, as it breaks translational symmetry. In general, two fractional spin systems are considered equivalent, if they belong to the same class after some symmetric grouping procedure.

Another way to formulate this procedure is by allowing symmetrically moving and fusing local Hilbert space. For the translational symmetric system, we can always move all spins within one unit cell to a single site and fuse them to a single spin. For systems with point group symmetry, by performing some symmetric movement, all local Hilbert space can be moved to high symmetry submanifolds (usually to be high symmetry points), known as Wyckoff positions of a space group. - And we will focus on such systems in the following discussion.

## 3.2 Real space constructions on SPT-LSM systems

In this part, we provide a recipe to classify and construct possible SPT phases for SPT-LSM systems. A more detailed and more mathematically rigorous treatment can be found in Appendix B and Appendix C.

### 3.2.1 Lattices and cells

We now give some basic definition about lattices and cells. Consider a $d$-dimensional spin system defined on lattice $Y$, where spins (can be either integer or fractional) live on sites of $Y$. We call sites/links/... of $Y$ as 0-cells/1-cells/.... And we define $Y_n$ as the set formed by $n$-cells of $Y$. The recipe presented in this section works for a special kind of lattice $Y$ satisfying the following condition. For an arbitrary cell $\Delta \in Y_n$ and its little group $SG_\Delta \subset SG$ which maps $\Delta$ to itself, we require any $g \in SG_\Delta$ has a pointwise action on $\Delta$. In other words, $SG_\Delta$ acts as an internal symmetry group locally on $\Delta$. In addition, we also assign orientation for cells of $Y$, which is required to be invariant under symmetry action. As an example, we present cell decomposition for 2D lattice with wallpaper symmetry group $P2$ (generated by translation $T_{x,y}$ and 180°-rotation $C_2$) in Fig. 2.

We notice that most lattices do not satisfy the above condition. For example, let us consider square lattice with translational symmetry $T_{x,y}$ and $C_4$ rotation at plaquette center. $C_4$ maps plaquette to itself, but the action is not pointwise. Yet, we are able to construct a new lattice which satisfies the pointwise-action condition by adding sites at every plaquette center, and connecting plaquette centers and original sites by adding new links. For an arbitrary lattice $Y$, we can always construct a new lattice $Y'$ from $Y$ by adding new cells, such that $Y'$ satisfies the pointwise-action condition.

A new spin system on $Y'$ is constructed by adding integer spins on the new sites of $Y'$ while keep the spins on sites of $Y$ untouched. Although the real-space construction algorithm only works for the new spin system on $Y'$, we expect to get the same classification result for the

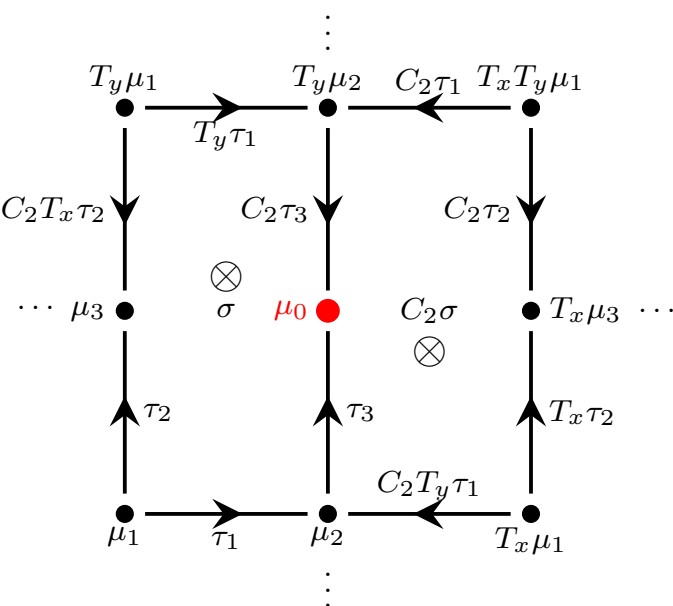

Figure 2: The cell decomposition for 2D wall paper group. The rotation center is marked in red. Here, we use $\sigma$ to label 2-cells, $\tau$ for 1-cell and $\mu$ for 0-cell. Cell $g\alpha$ is obtained from $\alpha$ by acting symmetry $g$. Their orientation are marked by arrows.

original spin system on $Y$.

We mention that the mathematical formulation of cells and their boundaries is presented in Appendix B.1.

### 3.2.2 Physical picture for real-space construction

Now, let us describe the physical picture behind real-space construction method. This construction method works for gapped states satisfying conditions that the correlation length $\xi$ is much smaller than the unit cell spacing $a$: $\xi \ll a$. So, lattice $Y$ mentioned in the last part can be viewed as "effective lattice", where cells of $Y$ ($\sim a$) are formed by many microscopic cells with lattice constant $l \sim \xi$. And spins on sites of $Y$ are treated as effective degrees of freedom, which are obtained by renormalization of microscopic spins. Under this assumption, it is legitimate to talk about decoration of gapped symmetric phases on an individual cell of $Y$. And gapped phases on the whole systems are then smoothly connected to some decoration of gapped phases on every cell of $Y$.

We focus on lattice $Y$ which satisfies the pointwise-action condition mentioned in the last part: for cell $\Delta \in Y_n$, the global symmetry for spin system on $\Delta$ are identified as $SG_\Delta$, which acts as onsite symmetry. We then decorate each cell with phases respecting onsite symmetry identified as the little group of the cell.

In the presence of lattice symmetry, decorations on cells related by lattice symmetry $g$ should be consistent with $g$ action. Let us label decoration on cell $\Delta$ as $\phi_\Delta$. For decoration on $g(\Delta)$, where $g \in SG$ is a lattice transformation, we can formally express the decoration as $\phi_{g(\Delta)} = g \circ \phi_\Delta$. (Meaning of $\phi$ and $g$ action on $\phi$ will be elaborated in Section 3.2.3.) Thus, for systems with lattice symmetry, it is enough to focus on decorations on a maximal subset of lattice symmetry independent cells, since decorations on cells beyond the subset can be generated by lattice symmetry action. This subset contains at least one element of $n$-cells, for any $0 \le n \le d$. Decoration pattern on this subset also gives us information about "orders" of SPT phase. If we put trivial SPT phase on all $n$-cells for any $n > d_0$ cells, and decorate non-trivial SPT phase on $d_0$ cells, we then expect to have $(d - d_0 + 1)$th order SPT phases.

One may think that by decorating each cell with some gapped SPT phases consistent with lattice symmetries, one obtains an SPT phase on the whole system, and different decorations give different SPT phases. Actually, the relation between cell-decoration and SPT phases of the whole system is more complicated and far from one-to-one correspondence. We would like to stress the following three issues.

First, as we will see, some decorations lead to gapless or symmetry breaking modes on interfaces between different cells, and fail to give a gapped symmetric phase for the whole system. It is necessary to figure out the consistent conditions for a valid decoration which leads to gapped symmetric phases.

Second, two seemingly different decorations may just differ by "a trivial decoration", which belong to the same phase. It is thus important to construct all possible "trivial decorations", and mod out those trivial decorations from all possible legitimate decorations.

Last but not the least, as we see in the 1D example presented in Section 2, fractional spin systems may give a distinct classification of SPT phases from integer spin systems. For a system with LSM anomaly, all symmetric SRE ground states are forbidden. For a system with SPT-LSM anomaly, the only allowed symmetric SRE ground states are SPT phases (but not the trivial state with no boundary excitations). Using this real-space construction formulation, can we identify whether a fractional spin system has "LSM anomaly" or "SPT-LSM anomaly"? For those SPT-LSM systems, can we systematically classify/construct SPT phases?

In the following, let us try to provide solutions to these three issues based on physical argument. More mathematical treatment is given in the next part, with details provided in Appendix B and C.

For the first issue, let us give a concrete decoration leading to gapless/symmetry breaking modes. In a $d$-dimensional lattice $Y$, let us consider decorations on two neighbouring $d$-cells $\Delta_1^d$ and $\Delta_2^d$ which intersect at a $(d-1)$-cell $\Delta^{d-1}$. If they are decorated by two distinct SPT phases, there will be gapless or symmetry breaking modes at $\Delta^{d-1}$. More generally, decoration on $n$-cells $\Delta_1^n, \cdots, \Delta_s^n$ which intersect at $(n-1)$-cell $\Delta^{n-1}$ will always lead to some boundary modes at $\Delta^{n-1}$. When "the summation" of decorations on those $n$-cells is a trivial $n$-dimensional SPT phase, we are able to gap out the boundary modes on $\Delta^{n-1}$ by adding some symmetric mass terms. Instead, if the summation of decorations gives non-trivial SPT phase, we are unable to symmetrically gap out the boundary modes at $\Delta^{n-1}$.

Yet there are more subtle cases. In general, even if decorations on $n$-cells avoid gapless modes on all $(n-1)$-cells, it is still possible that gapless modes appear at $(n-2)$ or even lower dimensional cells. In the next part, we present a mathematical tool to compute the boundary modes at interfaces, by which we are able to write down consistent equations for gapped SPT phases on the whole system.

The second issue is relatively easy to resolve. We simply define a trivial decoration as decorating every cell with some trivial SPT states. Notice that nearby cells in general are decorated with different trivial SPT states (which are all adiabatically connected to vacuum), leading to boundary modes at the interfaces. Yet these modes can be gapped out by adding symmetric mass terms on the interface. Thus, strictly speaking, for each cell, trivial decorations contains two elements: the trivial decorations and the symmetric mass term.

The issue about fractional spins are closely related to the issue of gapless boundary modes. A crucial observation is that fractional spins can be identified as gapless boundary modes on 0-cells. Thus, SPT phases on fractional spin systems corresponding to those decorations that are gappable in all $n > 0$ cells, but are gapless in 0-cells, with the gapless mode characterized by a particular class of projective representation. As we will show in the next part, inequivalent fractional spin systems have different classification of SPT phases, and these different classes have no common element. In other words, given a SPT decoration, it can never be realized in two inequivalent fractional spin systems.

Fractional spin systems can be divided to different categories according to the pattern of decorations. We first notice that a fractional spin system can never realize the trivial SPT phase, and thus must have either SPT-LSM anomaly or LSM anomaly. A fractional spin system is named as $d_0$th order SPT-LSM system, if the highest order SPT phases supported by this system are $d_0$th order SPT phases, which are obtained by non-trivial decorations in $(d-d_0+1)$-cells and trivial decoration on all higher dimensional cells. Notice that one can also realize different $n$th order SPT phases ($n \le d_0$) on a $d_0$th order SPT-LSM system by stacking an $n$th order SPT phase supported by integer spin systems.

There are fractional spin systems which can never realize any decorations with SPT phases. Such systems actually belong to the conventional LSM systems, where the symmetric phases realized in these systems must be long-range entangled.

### 3.2.3 Algorithm for real-space construction

In this part, we provide a well-defined algorithm to compute possible SPT phases on SPT-LSM systems. Derivation and detailed explanation of this algorithm is given in Appendix B and C.

As mentioned in the last part, we can label the decoration on $n$-cell $\Delta \in Y_n$ as $\phi_\Delta$, for any $0 \le n \le d$. Mathematically, $\phi_\Delta$ is a (homogeneous) $(n+1)$-cochain, which maps $n+2$ group elements to a $U(1)$ phase: $\phi_\Delta(g_0, \cdots, g_{n+1}) \in U(1)$, for $g_i \in SG$. $\phi_\Delta$ is required to satisfy the homogeneous condition for any $h \in SG_\Delta$:

$$\phi_\Delta(hg_0, \cdots, hg_{n+1}) = \rho_T(h) \cdot \phi_\Delta(g_0, \cdots, g_{n+1}), \tag{37}$$

where $\rho_T(h) = \pm 1$ if $h$ is unitary/antiunitary action.

Furthermore, for lattice symmetry $g \in SG$, decoration on $g(\Delta)$ are related to $\phi_\Delta$ by $g$ action, which is defined as

$$\phi_{g(\Delta)}(g_0, \cdots, g_{n+1}) = \rho_T(g)\phi_\Delta(g^{-1}g_0, \cdots, g^{-1}g_{n+1}). \tag{38}$$

Notice that Eq. (38) includes Eq. (37) as a special case.

Physically, the contribution of $\phi_\Delta$ comes from two parts: the first part is the decoration of SPT phases on $\Delta$ protected by $SG_\Delta$, and the second part is the symmetric "mass term" gapping out boundary modes of SPT phases decorated on nearby higher dimensional cells.

This physical picture is manifested in the relation between decorations on $\Delta \in Y_n$ and its nearby $(n+1)$-cells. For $1 \leq n \leq d$, the relation reads

$$d\phi_\Delta(g_0, \cdots, g_{n+2}) = \sum_{\substack{\Delta' \in Y_{n+1} \\ \Delta \text{ is part of } \partial\Delta'}} \pm\phi_{\Delta'}(g_0, \cdots, g_{n+2}), \tag{39}$$

where $d\phi_\Delta$ is the group coboundary operator defined as

$$d\phi_\Delta(g_0, \cdots, g_{n+2}) = \sum_{k=0}^{n+1}(-1)^k \phi_\Delta(g_0, \cdots, \hat{g}_k, \cdots, g_{n+2}), \tag{40}$$

and $\partial\Delta'$ denotes boundary $n$-cells of $\Delta'$. $\pm$ sign in Eq. (39) denotes that direction of $\Delta$ is consistent/inconsistent with direction of $\partial\Delta'$. We also define $Y_{d+1}$ as an empty set, which contains no cell. So for $\Delta \in Y_d$, Eq. (39) becomes a group cocycle condition:

$$d\phi_\Delta(g_0, \cdots, g_{d+2}) = 0. \tag{41}$$

For a legitimate decoration $\phi$, it satisfies Eq. (39) for any cells $\Delta$.

However, this consistent condition only applies for decorations on integer spin systems. For fractional spin systems, one should modify Eq. (39) for $\Delta \in Y_0$ as following. Fractional spins on an arbitrary 0-cell $\Delta \in Y_0$ is characterized by $[\nu_\Delta] \in H^2[SG_\Delta, U(1)_\mathcal{T}]$, where $\nu_\Delta$ also satisfies the homogeneous condition in Eq. (38) in order to be consistent with lattice symmetry actions. Here, $[.]$ means equivalent class by modding out coboundary elements. The gapping out condition for $\Delta \in Y_0$ should be modified as

$$\nu_\Delta(g_0, g_1, g_2) = \sum_{\substack{\Delta' \in Y_1 \\ \Delta \text{ is part of } \partial\Delta'}} \pm\phi_{\Delta'}(g_0, g_1, g_2), \tag{42}$$

where the equation holds up to coboundary. Any decoration $\phi_\bullet$ satisfying Eq. (38), (39), and (42) gives an SPT state for fractional spins labeled by $\nu_\bullet$.

We mention that two different decorations $\phi_\bullet^1$ and $\phi_\bullet^2$ may only differ by a "trivial decoration", and thus belong to the same SPT phase. Here, the trivial decoration is constructed in the following way. We start by decorating all cells with trivial phases consistent with lattice symmetry action in Eq. (38). The trivial decorations on an arbitrary $n$-cell $\Delta$ are characterized by a $n+1$ coboundary $d\varphi_\Delta$, where $\varphi_\Delta$ is a homogeneous $n$-cochain satisfying

$$\varphi_{g(\Delta)}(g_0, \cdots, g_n) = \rho_\mathcal{T}(g)\varphi_\Delta(g^{-1}g_0, \cdots, g^{-1}g_n), \tag{43}$$

for any $g, g_i \in SG$.

Nearby cells in general are decorated by different coboundaries, and thus will leave boundary modes on the interface. Yet these boundary modes can always be gapped by symmetric

mass terms on the interfaces. So, the final decoration on an arbitrary $n$-cell $\Delta$ again contains two parts: the mass term part and the trivial decoration part. Mathematically, trivial decoration on $\Delta \in Y_n$ reads

$$\phi_\Delta^0(g_0, \cdots, g_{n+1}) = \mathrm{d}\varphi_\Delta(g_0, \cdots, g_{n+1}) + \sum_{\substack{\Delta' \in Y_{n+1}\Delta \\ \text{is part of } \partial\Delta'}} \pm \varphi_{\Delta'}(g_0, \cdots, g_{n+1}), \qquad (44)$$

where the second line are symmetric mass terms.

As a consistent check, in Appendix B.3, we show that $\phi_\bullet^0$ satisfies Eq. (39) and Eq. (42) with $\nu$ to be a coboundary, and thus $\phi_\bullet^0$ indeed gives an SPT phase supported by integer spins. Any two decorations differ by such $\phi_\bullet^0$ should be treated as the same phase.

Solutions of Eq. (38), (39), and (42) can be solved in a iterative method. We consider solution with non-trivial decoration on $d_0$-cells and no decoration on higher dimensional $n$-cells ($\phi_{\Delta^n} = 0$ for any $\Delta^n \in Y_n$ with $n > d_0$). Such a solution gives a $(d - d_0 + 1)$th order SPT state.

In this case, Eq. (39) for $n = d_0$ becomes group cocycle condition

$$\mathrm{d}\phi_\Delta(g_0, \cdots, g_{d_0+2}) = 0. \qquad (45)$$

These equations constraint the decoration on $d_0$-cells to be cocycles (SPT states). However, there is no constraint on nearby cells if they are not related by any lattice symmetry.

We then consider equations for an arbitrary $(d_0 - 1)$-cell $\Delta^{d_0-1}$, which reads

$$\mathrm{d}\phi_{\Delta^{d_0-1}}(g_0, \cdots, g_{d_0+1}) = \pm \phi_{\Delta_1^{d_0}}(g_0, \cdots, g_{d_0+1}) \pm \phi_{\Delta_2^{d_0}}(g_0, \cdots, g_{d_0+1}), \qquad (46)$$

where $\Delta^{d_0-1} = \Delta_1^{d_0} \bigcap \Delta_2^{d_0}$, and $\pm$ sign depends on relative orientations. This equation puts constraints on possible decorations of neighbouring $d_0$-cells: SPT states decorated on $\Delta_1^{d_0}$ and $\Delta_2^{d_0}$ at most differ by a coboundary.

By examining equations on $n$-cells with $1 < n < d_0$, we exclude those decorations on $d_0$-cells that result in gapless mode on $n$-cells. And finally, decorations on nearby $d_0$-cells meeting at site $\Delta^0$ should give fractional spins $\nu_{\Delta^0}$.

It is easy to see that solutions for different $\nu$'s have no overlap. Furthermore, if $\phi_\bullet^1$ is a solution for fractional spins $\nu_1$ and $\phi_\bullet^2$ is a solution for $\nu_2$, $\phi_\bullet^1 + \phi_\bullet^2$ will then be a solution of $\nu_1 + \nu_2$. In particular, once we know a single decoration for fractional spins $\nu$, all other SPT decorations can be obtained by adding solutions for integer spins.

# 4 Examples for SPT-LSM systems in higher dimension

In this section, we provide examples of SPT-LSM systems in various dimensions for both strong and higher order SPT phases. We also propose a way to construct SPT-LSM systems realizing a given SPT phase from conventional LSM systems, based on the gauge charge condensation mechanism. For some examples, we give entanglement-spectrum based arguments to identify the nature of those enforced SPT phases.

The outline of this section is as following. We first provide a two-dimensional second order SPT-LSM system on honeycomb lattice. We argue that second order SPT phases on this system support degenerate boundary modes on symmetric samples with odd number of sites. While $d$th order SPT-LSM systems in $d$ dimension are somewhat trivial examples, we move to more non-trivial cases by proposing a procedure to construct SPT-LSM systems supporting strong SPT phases based on gauge charge condensation mechanism. Using the general procedure, we are able to identify several interesting examples in both 2D and 3D, including systems with "magnetic inversion" or "monopole translations".

## 4.1 Half-integer spins on honeycomb lattice – an example of 2nd order SPT-LSM systems

Let us consider possible gapped symmetric phases on the spin-1/2 honeycomb lattice.

It is well known that a square lattice with spin-1/2 per site satisfies a conventional LSM theorem. This is the consequence of an odd number of spin-1/2's in a unit cell. In fact, spin-1/2 at $C_4$ rotation center is enough to guarantee LSM anomaly.

In contrast, for the spin-1/2 system on honeycomb lattice, the total spin quantum number in a unit cell is integer, and therefore there is no LSM-type obstruction to realize a SRE symmetric ground state. Indeed, one can construct four classes of "featureless insulators" in this systems [52, 53], which are all symmetric gapped phases with trivial bulk excitations.

Here, instead of spin-1/2's, we present a construction of featureless insulators for spin-3/2's, which is more straightforward. As shown in Fig. 3, we decompose spin-3/2's to three spin-1/2's, and put them to point to three link directions respectively, and then make a singlet on a link from two spin-1/2's at two ends of the link. By choosing the sign of singlets carefully, we are able to construct featureless states respecting all lattice symmetries. This construction can be viewed as a 2D generalization of the AKLT construction in a 1D spin-1 chain. By using tensor networks, this fix-point wavefunction construction can be generalized to generic variational wavefunctions and also to other half-integer spin systems [53].

When the fix-point wavefunction is put on an open system with $C_3$ symmetry and an *odd number of sites*, it exhibits corner states with three free spin-1/2's related by $C_3$ symmetry. These $C_3$ symmetric corner states on odd-number-site samples are robust against symmetric perturbations, and are present for any of the four featureless insulators on this system. In this sense, these featureless insulators are identified as second order SPT phases.

To see the robustness of corner states, we present the following argument. Since the total spin is half-integral on odd number of sites, there are at least two-fold degeneracy in the ground state manifold protected by the $SO(3)$ symmetry. This degeneracy either comes from bulk states or edge states. If the bulk states are degenerate, it means that bulk excitations carry half-integer spin, which indicates that these excitations must be anyons (as local excitations have

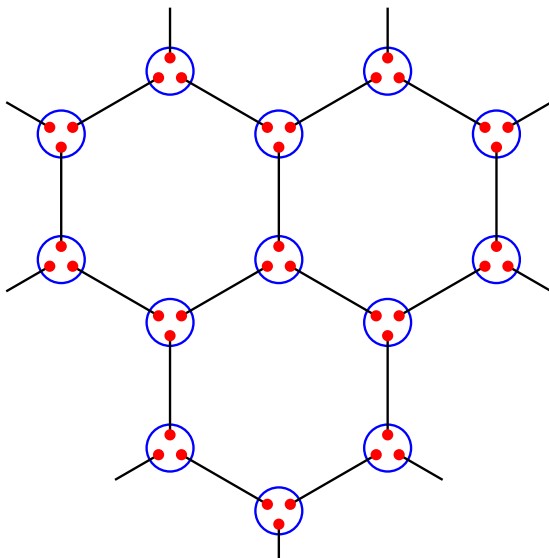

Figure 3: AKLT construction for spin-$\frac{3}{2}$ system on honeycomb lattice. A red point denotes one spin-$\frac{1}{2}$, and a blue circle on honeycomb site fuse three spin-$\frac{1}{2}$'s to one spin-$\frac{3}{2}$. A bond connecting two spin-$\frac{1}{2}$'s projects them to a spin singlet.

integer spins, e.g. magnons). So, for featureless insulators without bulk anyons, degeneracy must come from free spin-1/2's on edge states. To preserve $C_3$ symmetry, free spin-1/2's should appear in triples on the edge.

Note that the above argument applies to all honeycomb systems with half-integer spin each site, and thus, if we get a symmetric SRE phase in such systems, there must be edge/corner modes, which is a characterization of 1st/2nd order SPT phases.

We mention that different from conventional second order SPT phases, for featureless insulators on samples with *even number of sites*, all edge states can be gapped out.

## 4.2 General procedure to SPT-LSM systems

In this part, we describe a general procedure helping us to search for SPT-LSM systems, which enforces more interesting SPT phases. For these systems, decoration on 1-cell (coupled Haldane chain construction) is not able to absorb sites' fractional spins. Instead, decoration of higher dimensional (onsite) SPT phases are required in these systems.

To proceed, we first mention that many onsite SPT phases can be constructed by condensing bound states of gauge charges and symmetry charges of some symmetric gauge theory [33]. We review this condensation mechanism for one, two, and three dimensional SPT phases in Appendix D. To find the SPT-LSM system for a given SPT phase, our first step is to identify a symmetric gauge theory, where the desired SPT phase can be obtained by condensing gauge charge. There are various choice of gauge charge condensation. For example, by condensing "bare gauge charges" (labeled as $b_g$) which are singlets under global symmetries, one obtains a trivial symmetric phase.

To exclude this possibility, we would like to construct systems prohibiting condensation of $b_g$. This can be achieved by introducing additional lattice symmetries and constructing a conventional LSM system. And local Hilbert spaces of this system transform projectively under symmetries. One example is the 2D spin-1/2 system with spin rotation and translational symmetry. Excitation of the $Z_2$ gauge theory on this system (also known as $Z_2$ spin liquid) has non-trivial symmetry properties: spinon carry half-integer spin while vison pick up a minus sign under $T_x T_y T_x^{-1} T_y^{-1}$. Thus, condensing either quasiparticles leads to spontaneously symmetry breaking. Different from the usual convention, we always identify gauge charges as quasiparticles transforming linearly under onsite symmetry, which are vison in this system. In the LSM system, gauge charge $b_g$ transform projectively under lattice symmetries, and condensing them leads to spontaneous symmetry breaking. The searching for such conventional LSM system is relatively easier than for the SPT-LSM system, as we have more intuition (such as parton construction) for constructing symmetric gauge theories.

Despite being a singlet of onsite symmetry $s$, the bound state $b_g b_s$ of gauge charge $b_g$ and symmetry charge (labeled as $b_s$) transforms projectively under lattice symmetries in such system, and the condensation of $b_g b_s$ gives a mixture of onsite SPT and lattice SSB phase. To obtain a fully symmetric phase, one way is to modify lattice symmetry by entangling it with onsite symmetries, such that $b_s$ transforms oppositely from $b_g$ under the modified lattice symmetry. Example for such lattice symmetries includes magnetic translation group. We mention that one should be very careful for the modification of lattice symmetries: it is not guaranteed that the symmetric gauge theory on the original system will survive for the modified system. We provide an counterexample in Appendix E, where it is impossible to construct the symmetric gauge theory with the modified lattice symmetries.

This construction method ensure the system we obtained is the desired SPT-LSM system. However, one may able to realize more than one type of SPT phases on such system. In particular, the SPT-LSM system may support higher order SPT phase, in addition to the 1st-order strong SPT phases. We provide an example in Section 4.4, where one can realize both 1st-order and 2nd-order SPT phases in a 3D SPT-LSM system.

Calculation based on real space construction method is provided in Appendix C.

## 4.3   2D SPT-LSM system with magnetic inversion symmetry

In this part, we construct SPT-LSM systems which enforce a 2D strong SPT phase protected by onsite symmetry $Z_2^s \times Z_2^{\mathcal{T}} = \{1, s\} \times \{1, \mathcal{T}\}$.

   We first provide the classification of SPT phases with $Z_2^s \times Z_2^{\mathcal{T}}$. Using group cohomology, these phases are classified by $H^3[Z_2^s \times Z_2^{\mathcal{T}}, U(1)_{\mathcal{T}}] = Z_2^2$. The first $Z_2$ root phase, labeled as $\nu_s = 1$, is the well-known Levin-Gu SPT phase protected by $Z_2^s$ symmetry [54], while the second $Z_2$ root phase, labeled as $\nu_{s\mathcal{T}} = 1$, comes from interplay between $Z_2^s$ and $Z_2^{\mathcal{T}}$. In particular, $\nu_{s\mathcal{T}} = 1$ phase has a decorated domain wall picture: domain walls of $Z_2^s$ are decorated with $Z_2^{\mathcal{T}}$ Haldane chains [32]. A $Z_2^s$ domain wall can terminate on a $Z_2^s$ symmetry flux, which then carries a Kramers doublet.

   Our goal is to construct SPT-LSM systems with $\nu_{s\mathcal{T}} = 1$ ($\nu_s$ can be either 0 or 1). We mention that an example based on magnetic translation group is presented in Ref. [25]. Here, we give a new example based on inversion symmetry.

   The global symmetry group for the system is $Z_4^{\widetilde{\mathcal{I}}} \times Z_2^{\mathcal{T}}$, where $Z_4^{\widetilde{\mathcal{I}}}$ is generated with "magnetic inversion" $\widetilde{\mathcal{I}}$ with $\widetilde{\mathcal{I}}^2 = s$ and $\widetilde{\mathcal{I}}^4 = 1$. At inversion center, the local Hilbert space transforms as a Kramers doublet. As we will show in the following, this system is an SPT-LSM system, where a symmetric SRE phase in this system must be an SPT phase with $\nu_{s\mathcal{T}} = 1$.

### 4.3.1   $\nu_{s\mathcal{T}} = 1$ SPT phases from gauge charge condensation

In this part, we follow the general procedure in Section 4.2 to "derive" the SPT-LSM system.

1. The first step is to identify a symmetric gauge theory, such that condensing its gauge charge leads to $\nu_{s\mathcal{T}} = 1$ phase.

   We start from a $Z_2$ gauge theory (toric code topological order) with global symmetry $Z_2^s \times Z_2^{\mathcal{T}}$, with gauge flux $m$ transforming as a Kramers doublet. Gauge charge $e$ here carries linear representation. Symmetry action on these anyons can be expressed as

$$\mathcal{T}^2 \circ m = -m, \quad s^2 \circ m = m;$$
$$\mathcal{T}^2 \circ e = s^2 \circ e = e. \tag{47}$$

   $\nu_{s\mathcal{T}} = 1$ phase is obtained by condensing bound state of $e$ and a $Z_2^s$ charge excitation $\mathcal{R}_s$ [33], while condensing $e$ leads to trivial SPT phase. Details of this condensation mechanism can be found in Appendix D.3.

2. To design an SPT-LSM system, we should prohibit the condensation of bare gauge charge $e$. As we showed in Section 4.2, one way is to add lattice symmetry and start from a "conventional" LSM system.

   In this example, we add inversion symmetry $\mathcal{I}$, and we present a simple cell decomposition for systems with symmetry $\mathcal{I}$ in Fig. 4.

   The LSM condition for this system can be satisfied by putting a Kramers double at inversion center. All $Z_2$ symmetric gauge theories realized in this system is "anomalous": both $e$ and $m$ transform projectively under total symmetry group. One particular choice of symmetry action is:

$$\mathcal{I}^2 \circ e = -e, \quad \mathcal{T}^2 \circ m = -m. \tag{48}$$

   Due to the non-trivial symmetry action on both $e$ and $m$, condensing either quasiparticle leads to spontaneously symmetry breaking phase. In particular, by condensing bound

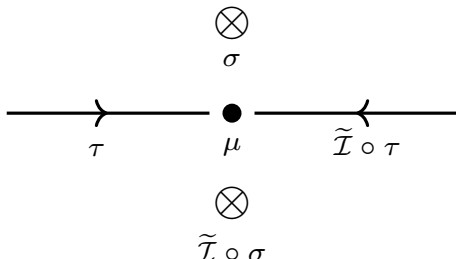

Figure 4: Cell decomposition of $Y = \mathbb{R}^2$ respecting inversion symmetry around $\mu$. Cells are grouped according to their dimension as $Y_2 = \{\sigma, \widetilde{\mathcal{I}} \circ \sigma, \dots\}$, $Y_1 = \{\tau, \widetilde{\mathcal{I}} \circ \tau, \dots\}$, and $Y_0 = \{\mu, \dots\}$, where $\dots$ denote cells that are not drawn here. 1-cells $\tau$ and $\widetilde{\mathcal{I}} \circ \tau$ point towards $\mu$, while directions of 2-cells $\sigma$ and $\widetilde{\mathcal{I}} \circ \sigma$ are pointing into the paper.

state of $e$ and $Z_2^s$ charge $\mathcal{R}_s$, we get mixture phase with $\nu_{s\mathcal{T}} = 1$ SPT and inversion symmetry breaking.

3. In order to obtain a fully symmetric SPT phase, we modify inversion to "magnetic inversion" $\widetilde{\mathcal{I}}$ with $\widetilde{\mathcal{I}}^2 = s$. In other words, we have $\widetilde{\mathcal{I}}^2 \circ R_s = -R_s$ as well as $\widetilde{\mathcal{I}}^2 \circ e = -e$. Then, bound state of $e$ and $\mathcal{R}_s$ transform trivially under the modified symmetry group: $\widetilde{\mathcal{I}}^2 \circ (eR_s) = eR_s$. By condensing $eR_s$, we obtain the $\nu_{s\mathcal{T}} = 1$ phase without breaking any lattice symmetry.

We point out that SPT phases obtained in this system is not unique, which is related to the fact that symmetry action on the $Z_2$ gauge theory is not uniquely determined in the original LSM system. For example, we could start from a different symmetric gauge theory with additional non-trivial $Z_2^s$ action $s^2 \circ m = -m$. Condensing $eR_s$ in this case leads to phase $\nu_s = \nu_{s\mathcal{T}} = 1$.

### 4.3.2 Entanglement argument

In this part, we give an entanglement-based argument to prove that any gapped symmetric ground state must be a SPT phase with $\nu_{s\mathcal{T}} = 1$.

Without loss of generality, we consider a square lattice model, where Ising charges (neutral under time reversal $\mathcal{T}$) live on each lattice site $\mathbf{r} = (x, y) \in \mathbb{Z}^2$. Besides, there is a Ising-neutral spin-1/2, which is a projective representation of time reversal symmetry $\mathcal{T}$, living on each plaquette center $(x + \frac{1}{2}, y + \frac{1}{2})$. The magnetic inversion symmetry $\tilde{\mathcal{I}}$ is implemented as

$$\tilde{\mathcal{I}} \equiv \left[\prod_{\mathbf{r}} (s_{\mathbf{r}})^y\right] \cdot \mathcal{I}, \tag{49}$$
$$(x, y) \xrightarrow{\mathcal{I}} (1 - x, 1 - y),$$

where we have chosen the inversion center to be each plaquette center. It is straightforward to check that $\tilde{\mathcal{I}}^2 = \prod_{\mathbf{r}} s_{\mathbf{r}} = s$ is indeed satisfied, where $s_{\mathbf{r}}$ denotes $Z_2$ spin rotation on each site $\mathbf{r}$.

One can always embed the $Z_2$ Ising symmetry $s$ in a $U(1)$ group, for example by choosing

$$s = e^{i\pi\hat{Q}}, \quad s^2 = e^{2\pi i\hat{Q}} = 1, \tag{50}$$
$$U(1) \equiv \{e^{i\phi\hat{Q}} | 0 \le \phi < 2\pi\}.$$

The magnetic inversion symmetry $\tilde{\mathcal{I}}$ implements the following constraint on the $U(1)$ vector potential

$$\vec{A}(1 - x, 1 - y) = -\vec{A}(x, y) + (0, \pi), \tag{51}$$

which has the following solution in the Landau gauge

$$\vec{A}(x, y) = (0, \pi x).$$ (52)

This implies the presence of a $\pi$ flux (i.e. an Ising symmetry flux) in each plaquette, in addition to the spin-1/2 at the plaquette center.

When put on a cylinder with infinite length $L_x \longrightarrow +\infty$ and an *odd* circumference $L_y =$ odd, the boundary condition along $\hat{y}$ direction oscillates between periodic and anti-periodic boundary conditions between different columns $x =$ even and $x =$ odd. Below we present an argument based on entanglement spectrum properties of a SRE quasi-1D cylinder with time reversal symmetry [18], which dictates that any symmetric SRE ground state must be a strong SPT phase with $\nu_{s\mathcal{T}} = 1$ in a way similar to Ref. [24, 25, 55].

We consider the entanglement spectra of the $L_y =$ odd cylinder at two different cuts $\bar{x} = \epsilon \in (0, 1/2)$ and $\bar{x} = 1 - \epsilon$, which are related by inversion symmetry $\tilde{\mathcal{I}}$. We write the Schmidt decompositions of the symmetric SRE ground state $|\psi\rangle$ w.r.t. the two cuts as

$$|\psi\rangle = \sum_{\lambda_\epsilon} \lambda_\epsilon |L_{\lambda,\epsilon}\rangle \otimes |R_{\lambda,\epsilon}\rangle = \sum_{\lambda_{1-\epsilon}} \lambda_{1-\epsilon} |L_{\lambda,1-\epsilon}\rangle \otimes |R_{\lambda,1-\epsilon}\rangle.$$

Since there is an odd number of Kramers doublet between the two cuts, the degeneracy of Schmidt eigenstates at the two cuts must differ by 2-fold due to Kramers degeneracy [18]. Without loss of generality, we assume Schmidt eigenstates (e.g. $|L_{\lambda,\epsilon}\rangle$) at cut $\epsilon$ are Kramers singlets (non-degenerate) and those at cut $1-\epsilon$ (e.g. $|R_{\lambda,1-\epsilon}\rangle$) are Kramers doublets (two-fold degenerate). However due to magnetic inversion symmetry $\tilde{\mathcal{I}}$, under pure spatial inversion operation which maps the spatial region of $|L_{\lambda,\epsilon}\rangle$ to the region of $|R_{\lambda,1-\epsilon}\rangle$, the only change to the many-body Hamiltonian is the twisting of boundary condition along $L_y$ direction by the onsite $Z_2$ symmetry $s$. This indicates that twisting boundary condition by $s$ for any symmetric SRE state must also change the entanglement spectrum by a Kramers degeneracy. This is a defining property for 2D SPT phase with $\nu_{s\mathcal{T}} = 1$, where a Kramers doublet of $\mathcal{T}$ symmetry is bound to each flux of Ising symmetry $s$ [32]. Therefore we have shown that this is indeed a 1st-order SPT-LSM system, where each symmetric SRE ground state must be a 1st-order (i.e. strong) 2D SPT phase with $\nu_{s\mathcal{T}} = 1$.

### 4.3.3 Model Hamiltonian

We now briefly describes a model that realizes such a SPT-LSM system. The model is in fact identical to one studied in for a SPT-LSM theorem with magnetic translation symmetry. We will only describe the setup and sketch the Hamiltonian, referring the details to Ref. [25].

Consider a spin-1/2 triangular lattice, and the dual honeycomb lattice. On the dual lattice we place Ising spins on each site. The system has SO(3) spin rotation symmetry acting on spin-1/2's on the triangular lattice, and $Z_2$ symmetry on the Ising spins on the dual lattice, generated by $\prod_p \sigma_p^x$.

We also define a "magnetic" site inversion symmetry on the triangular lattice. First we define the coordinate system. We label the honeycomb sites on one sublattice by $m, n$ so its coordinate is $\mathbf{r} = m\mathbf{a} + n\mathbf{b}$, where $\mathbf{a} = (1, 0), \mathbf{b} = (\frac{1}{2}, \frac{\sqrt{3}}{2})$. The other sublattice is $m\mathbf{a} + n\mathbf{b} + \frac{\mathbf{a} + \mathbf{b}}{3}$. Denote $\mathcal{I}$ the "normal" inversion that only operates on the spatial coordinates,

$$\tilde{\mathcal{I}} = \prod_{\mathbf{r}} [\sigma_{\mathbf{r}}^x]^n \cdot \mathcal{I}.$$ (53)

We now briefly describe the Hamiltonian, which is in fact identical to the one given in Ref. [25]. There are three kinds of terms in the Hamiltonian:

$$H = H_{\text{Ising}} + H_{\text{binding}} + H_A,$$ (54)

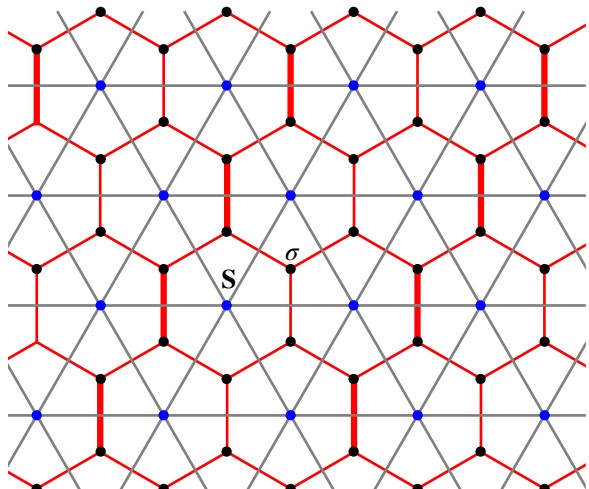

Figure 5: Illustration of the 2D Hamiltonian, where spin-1/2's form a triangular lattice, and Ising spins reside on the dual honeycomb lattice. Thicken bonds denote frustrated Ising couplings.

where $H_{\text{Ising}}$ takes the following form:

$$H_{\text{Ising}} = -K \sum_{\langle pq \rangle} s_{pq} \sigma_p \sigma_q \,. \tag{55}$$

The signs $s_{pq}$ are chosen such that around each hexagonal plaquette the product of $s$'s is equal to $-1$, i.e. a $\pi$ flux lattice. One choice is depicted in Fig. 5. Thus the Ising couplings are frustrated. Next $H_{\text{binding}}$ couples the Ising spins and the spin-1/2's:

$$H_{\text{binding}} = -\lambda \sum_e (1 - s_{pq} \sigma_p \sigma_q) P_e \,, \tag{56}$$

where $e$ sums over nearest-neighbour edges of the triangular lattice, and $P_e$ projects the two spins connected by $e$ to a spin singlet. $p$ and $q$ denote the two plaquettes adjacent to $e$. $H_A$ gives dynamics to the spin singlets and its form is quite complicated, so we refer the readers to for details. Due to the frustrated Ising couplings, the (honeycomb plaquette-centered) inversion symmetry must be magnetic, given in Eq. (53). As shown in Ref. [25], in the limit $\lambda \to \infty$, the model realizes precisely the SPT phases expected from the SPT-LSM theorem.

## 4.4 3D SPT-LSM system with magnetic inversion

Now we consider a 3D SPT-LSM system with $Z_2^s \times Z_2^{\mathcal{T}}$ global symmetry and magnetic inversion. The 3D inversion again satisfies $\widetilde{\mathcal{I}}^2 = s$. Notice that the 3D inversion is an orientation-reversing operation. One simple cell decomposition is presented in Fig. 6.

Interestingly, now we can construct at least two completely different kinds of SPT phases in this system:

1. A strong SPT phase protected by $Z_2^s \times Z_2^{\mathcal{T}}$.

2. A 2nd order SPT phase: restricted to an arbitrary plane passing through the inversion center, we find exactly the same 2D SPT-LSM system discussed in the previous section. Thus one can form the 2D SPT phase with $\nu_{s\mathcal{T}} = 1$ on this plane.

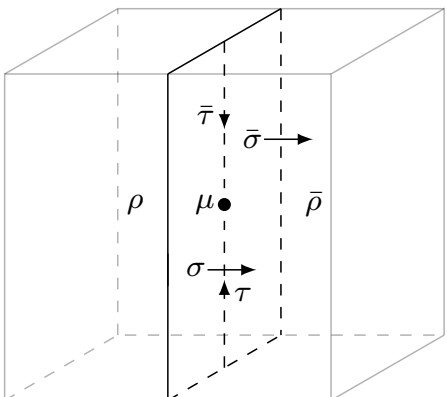

Figure 6: Cell decomposition for a 3D lattice respecting inversion symmetry with inversion center $\mu$. Cell $\bar{s}$ and cell $s$ are related by (magnetic) inversion symmetry $\widetilde{I}$: $\bar{s} = \widetilde{I} \circ s$. Orientations for 1-cells and 2-cells are denoted by arrows, while 3-cells $\rho$ and $\bar{\rho} = \widetilde{\mathcal{I}} \circ \rho$ have opposite orientations.

Let us discuss in more detail how to realize the first option, i.e. a strong SPT phase. We start from a LSM theorem, with a Kramers doublet at the 3D inversion center and the inversion $\mathcal{I}^2 = 1$. In such a system, one may realize a U(1) spin liquid. We assume that under the time reversal and the inversion, the gauge fields transform as

$$\begin{aligned} \mathcal{T} &: \mathbf{E} \to -\mathbf{E}, \mathbf{B} \to \mathbf{B}, \\ \mathcal{I} &: \mathbf{E} \to \mathbf{E}, \mathbf{B} \to -\mathbf{B}. \end{aligned} \tag{57}$$

Notice that the Gauss law $\rho = \nabla \cdot \mathbf{E}$ implies that the gauge charge density $\rho$ changes sign under $\mathcal{I}$. Therefore, $\mathcal{T}^2$ is well-defined for magnetic monopoles and $\mathcal{I}^2$ is well-defined for electric charges. We set $\mathcal{I}^2 = -1$ on electric charge and $\mathcal{T}^2 = -1$ on monopoles, which realizes the LSM anomaly.

Now we condense the bound state of an electric charge with a $Z_2^s$ charge $\mathcal{R}_s$, which transforms as $\tilde{\mathcal{I}}^2 = s$. This object transforms trivially under all symmetries and thus the condensation leads to a SPT phase.

In order to understand the nature of the SPT phase, it is convenient to first Higgs the gauge symmetry down to $Z_2^g$. This can be done by condensation of a pair of electric charges. After the condensation, monopoles are confined and there emerges $\pi$ flux loops. Two identical flux loops fuse into a $2\pi$ flux loop, whose end points correspond to (now confined) unit magnetic monopoles. Recall that these monopoles are Kramers doublets. In other words, we can think of a $2\pi$ flux loop as carrying a Haldane chain. Therefore, a $\pi$ flux loop must carry a "half" Haldane chain. This phenomenon can be regarded as a generalization of the familiar fractional charges of anyons in 2D, while integer charges are identified as 0D SPT states.

Now we further condense the bound state of a $Z_2^g$ gauge charge with a $Z_2^s$ charge $\mathcal{R}_s$. This step is essentially the same as before. The condensation now implies that we may identify the $Z_2^s$ symmetry flux loop with the $Z_2^g$ flux loop, which carries a half Haldane chain. This corresponds to the following term in the Künneth decomposition of $H^4[Z_2^\mathcal{T} \times Z_2^s, U(1)]$:

$$H^3[Z_2^\mathcal{T}, H^1[Z_2^s, U(1)]] = Z_2, \tag{58}$$

where the non-trivial element in $H^1[Z_2^s, U(1)] = Z_2$ characterizes the $Z_2^s$ symmetry flux loop, and $H^3[Z_2^\mathcal{T}, Z_2]$ describes decoration of half-Haldane chain.

We also perform a spectral sequence calculation and confirm the results in Appendix C.3.

## 4.5   3D SPT-LSM with monopole translation

In this part, we propose an SPT-LSM system enforcing strong SPT phases in 3+1D protected by onsite symmetry group $U_s(1) \times Z_2^{\mathcal{T}}$. Physically, this case corresponds to a time reversal invariant spin system in which the $z$ component of spin is conserved. Cohomological group calculation gives $Z_2^3$ classification:

$$H^4\big[U_s(1) \times Z_2^{\mathcal{T}}, U_{\mathcal{T}}(1)\big] = Z_2^3 \,. \tag{59}$$

And there is another $Z_2$ class beyond group cohomology classification [48, 56].

There are two ways to characterize these phases: either by studying the surface state [56, 57] or by by coupling $U_s(1)$ charge to external compact electromagnetic field, and studying properties of external monopole excitations. Properties of these SPT phases, as well as approaches to obtain them from monopole condensation are reviewed in Appendix D.4.

Here, we focus on one $Z_2$ root phase, whose external monopole transforms as a Kramers doublet under time reversal. To design an SPT-LSM system enforcing such a phase, we follow the procedure presented in Section 4.2. We first identify a possible route to obtain this SPT phase by condensing quasiparticles of a symmetric gauge theory. One starts from a compact $U_g(1)$ quantum spin liquid (QSL) with global symmetry $U_s(1) \times Z_2^{\mathcal{T}}$. Excitations of this spin liquid are gauge charges, monopoles as well as photons. For the purpose here, we can safely ignore photons, and focus on symmetry properties of gauge charges and monopoles. We mention that monopoles and gauge charges are dual to each other: monopoles can be viewed as gauge charges of a dual $\widetilde{U_g}(1)$ gauge field.

The $U_g(1)$ gauge field can be killed by condensing gauge charges or monopoles or their bound states. Here, we focus on phases obtained from monopole condensation.

We assume the following symmetry properties of this $U_g(1)$ QSL: gauge charge, labeled as $b_g$, is a Kramers doublet, while the monopole $M_g$ is transformed into its antiparticle $M_g^\dagger$ under time reversal. Both of them transform trivially under $U_s(1)$. We claim that condensing bound state of monopole $M_g$ and a unit $U_s(1)$ charge leads to the SPT phase with Kramers doublet monopole of $U_s(1)$. Notice that monopole $M_g$ of $U_g(1)$ gauge field and the external monopole of $U_s(1)$ are two different objects, and should not be confused with each other. The detailed argument for this condensation mechanism is presented in Appendix D.4.3.

More precisely, under onsite symmetry $U_s(1) \times Z_2^{\mathcal{T}}$, gauge charge $b_g$ and monopole $M_g$ transform as

$$
\begin{aligned}
U_s(\theta) &: b_g \to b_g \,, & M_g &\to M_g \,; \\
\mathcal{T} &: b_g \to \mathrm{i}\,\sigma^y \cdot b_g \,, & M_g &\to M_g^\dagger \,, & \mathrm{i} \to -\mathrm{i} \,,
\end{aligned}
\tag{60}
$$

where $b_g = (b_{g\uparrow}, b_{g\downarrow})^{\mathrm{t}}$ is a two component bosonic operator with spin index.

Notice that there is an important distinction between onsite unitary and anti-unitary symmetries. Under onsite unitary symmetry action, both $b_g$ and $M_g$ should either remain in the same topological sector or both transform to their antiparticles. However for onsite anti-unitary action, in order to preserve commutation relation between electric field $\vec{E}_g$ and vector potential $\vec{A}_g$, if $b_g$ transforms to its antiparticle $b_g^\dagger$, $M_g$ must remain in the same topological sector, and vice versa.

This $U_g(1)$ spin liquid can be realized in a spin system, where local Hilbert space contains one qubit (with basis $|\uparrow\rangle$ and $|\downarrow\rangle$) and one qutrit (with basis $|0\rangle$ and $|\pm 1\rangle$). The qubit is a Kramers doublet, but carry no $U_s(1)$ charge. Meanwhile the qutrit is a Kramers singlet, and carries $U_s(1)$ charge. By introducing $\vec{S}$ as spin-1/2 operator for qubit and $B^\pm$ as raising/lowering operator for qutrit, the symmetry action reads

$$
\begin{aligned}
U_s(\theta) &: \vec{S} \to \vec{S} \,, & B^\pm &\to \mathrm{e}^{\pm \mathrm{i}\theta} B^\pm \,, \\
\mathcal{T} &: \vec{S} \to -\vec{S} \,, & B^+ &\leftrightarrow B^- \,, & \mathrm{i} \to -\mathrm{i} \,.
\end{aligned}
\tag{61}
$$

Then, $U_g(1)$ QSL can be constructed using parton formulation. $U_g(1)$ gauge charge $b_g$ are identified as partons for spin operator

$$\vec{S} \sim \frac{1}{2} b_g^\dagger \cdot \vec{\sigma} \cdot b_g. \tag{62}$$

Then, under global symmetry, $b_g$ transforms in the same way as shown in Eq. (60). The desired $U_g(1)$ spin liquid phase is obtained by putting $b_g$ on a trivial gapped Mott insulator, and thus monopoles transform linearly under global symmetry. The SPT phase with Kramers doublet $U_s(1)$ external monopoles is obtained by condensing the bound state of $M_g$ and $U_s(1)$ charge $S^+$.

Having identified the symmetric gauge theory, the next step to figure out a 3+1D LSM system to support this $U_g(1)$ gauge theory with the same onsite symmetry action defined in Eq. (60). Here, let us start from a cubic lattice system with one qubit living on each lattice site with onsite symmetry defined in Eq. (61). We also impose translational symmetry as

$$T_i : \vec{S}(j) \rightarrow \vec{S}(j + \hat{e}_i). \tag{63}$$

(We will add qutrit later for this construction.)

This system has LSM anomaly due to a single Kramers doublet per unit cell, and disallows symmetric SRE phase. We are able to construct $U_g(1)$ QSL phase with the same onsite symmetry properties defined in Eq. (60) by parton construction. Physical Hilbert space is identified as one boson per site. Due to this restriction, one effectively introduces $U_g(1)$ gauge field, and $b_g$ is identified as gauge charge, while $M_g$ lives on the cubic center (or dual lattice site). One can choose mean field ansatz for $b_g$ with onsite chemical potential and nearest neighbouring pairing terms. This ansatz is invariant under a global $U_g(1)$ transformation with

$$U_g(\phi) : b_g(j) \rightarrow e^{\pm i\phi} b_g(j). \tag{64}$$

Here, we choose $+\phi$ for even sites and $-\phi$ for odd sites where for site $j = (j_x, j_y, j_z)$, even/odd lattice site means $j_x + j_y + j_z$ is an even/odd number. This action is actually a gauge transformation, and is named as invariant gauge group (IGG), which determines low-energy gauge dynamics [58]. Notice that nearest neighbouring hopping breaks $U_g(1)$ IGG to $Z_2$, and is identified as Higgs terms.

How do gauge charges and monopoles transform under lattice symmetries? We notice that translations have non-trivial action on IGG: $T_\alpha U_g(\phi) T_\alpha^{-1} = U_g(-\phi)$. In other words, translations act as charge conjugation on gauge charges. (Remember that $b_g(j)$ at even/odd $j$ carries positive/negative gauge charge.)

To preserve commutation relation between $\vec{E}$ and $\vec{A}$, $T_\alpha$ should also map $M_g(j)$ to its antiparticle $M_g^\dagger(j + \hat{e}_\alpha)$ up to a phase factor. Every site lives a single $b_g$, which is interpreted as background gauge charge: there is one positive gauge charge on each even site, and one negative gauge charge on each odd site. Due to the background gauge charge distribution, magnetic monopole $M_g$ would acquire non-trivial Berry phase when hopping around a closed loop. A specific hopping ansatz for $M_g$ to characterize "odd number of gauge charges per unit cell" is given in Ref. [59]. Here, we present the ansatz in Fig. 7.

According to this mean field ansatz, we extract translation action on $M_g$ as

$$\begin{aligned}
T_x &: M_g(j) \rightarrow M_g^\dagger(j + \hat{x}), \\
T_y &: M_g(j) \rightarrow M_g^\dagger(j + \hat{y}), \\
T_z &: M_g(j) \rightarrow i^{(x+y)^2 + 2x} M_g^\dagger(j + \hat{z}).
\end{aligned} \tag{65}$$

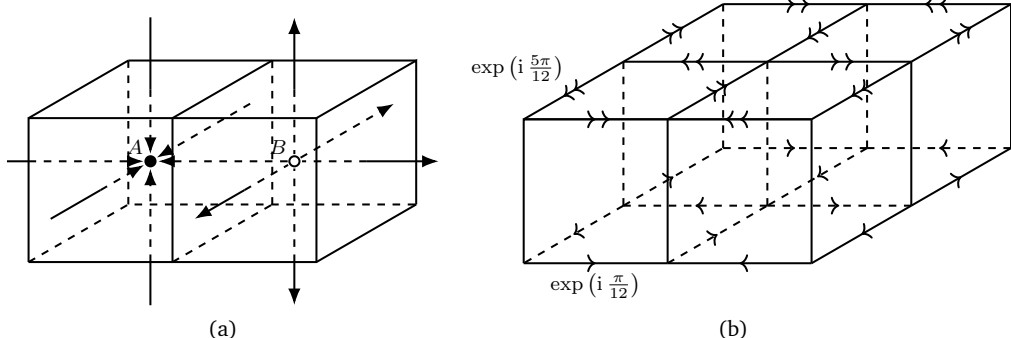

Figure 7: (a) Monopole on dual cubic lattice with $\pm 1$ background gauge charge per unit cell. $A/B$ denotes gauge charge $-1/+1$ as the drain/source of electric line. (b) Hopping ansatz for monopole with the above non-trivial gauge charge background. Monopole picks up phase factor $\exp(\pm i\pi/12)$ when hops along the same/opposite direction of single arrow, while picks up $\exp(\pm i 5\pi/12)$. Thus, monopole will pick up $\pm\frac{\pi}{3}$ Berry phase when travelling around a plaquette.

As shown in Ref. [59], while condensation of $b_g$ leads to magnetic ordered phases, condensation of monopole always breaks translational symmetry, and patterns of the resulting VBS orders depend on details of condensation.

Now, let us add one qutrit at every site. Qutrits carry $U_s(1)$ charge, as shown in Eq. (60). As discussed before, condensation of the bound state of $M_g$ and $B^+$ leads to the SPT phase with Kramers doublet external $U_s(1)$ monopole. However this condensation breaks translational symmetry assuming translation act trivially on qutrits, and leads to mixture of VBS and SPT phase. To avoid lattice symmetry breaking and obtain fully symmetric phase, the last step is to carefully design translation actions on qutrits, such that hopping ansatz for qutrits is the complex conjugate of hopping ansatz for $M_g$. Thus, the bound state of $M_g$ and $B^+$ will hop in a zero flux background. By condensing the bound state (or design correlated hopping of $M_g$ and $B^+$) at $\Gamma$ point, we get a symmetric SRE phase, with the desired SPT index.

Let us now identify the modified translation symmetries for this $B^+$ hopping ansatz. Since background flux for $B^+$ would be opposite to flux for $M_g$, translation symmetry action on spin operators becomes

$$
\begin{aligned}
T_x &: \ B^+(j) \to B^-(j+\hat{x}); \\
T_y &: \ B^+(j) \to B^-(j+\hat{y}); \\
T_z &: \ B^+(j) \to (-i)^{(x+y)^2+2x} B^-(j+\hat{z}).
\end{aligned}
\tag{66}
$$

We coin the above modified translation operators as "monopole translation operators".

One may worry if it is possible to obtain trivial symmetric SRE phase by condensing other bound states of gauge charges and monopoles (dyons). A dyon can be labeled by a 2D integer vector $(e, m)$, where $e$ denotes the electric charge and $m$ is the magnetic charge. In this QSL, charge $(1, 0)$ and monopole $(0, 1)$ are boson, so when $e \cdot m$ is even/odd, dyon $(e, m)$ is a boson/fermion [60].

Let us consider condensation of bosonic dyons. When $\gcd(e, m) = n > 1$, condensing dyon leads to discrete $Z_n$ gauge theory. So, we require the condensed dyon satisfying $\gcd(e, m) = 1$.

Under $\mathcal{T}$ action, $(e, m) \to (e, -m)$. When $e$ and $m$ are both nonzero, the condensed phase would break $\mathcal{T}$ symmetry. We are only left with options with electric charge $(\pm 1, 0)$ and magnetic monopole $(0, \pm 1)$. However, condensing charge $(\pm 1, 0)$ would break $U_s(1)$ symmetry according to Eq. (60). So, to obtain symmetric SRE phase from this $U_g(1)$ QSL, the only

choice is to condense monopole/anti-monopole. In order to preserve monopole translation symmetry after condensation, we should condense the bound state of monopole and $U_s(1)$ charge $B^+$, and the resulting phase is nothing but the desired SPT phase.

Now, we are able to identify the global symmetry group by commutation relations of generators, which are

$$
\begin{aligned}
&\mathcal{T}^2 = 1 \,; \\
&\mathcal{T} U_s(\theta) \mathcal{T}^{-1} = U_s(\theta), \quad T_\alpha \mathcal{T} T_\alpha^{-1} = \mathcal{T} \,; \\
&T_\alpha U_s(\theta) T_\alpha^{-1} = U_s(-\theta), \quad \alpha = x, y, z \,; \\
&T_x T_y = T_y T_x \,; \quad T_y T_z = U_s\!\left(\frac{\pi}{2}\right) T_z T_y \,; \\
&T_z T_x = U_s\!\left(\frac{\pi}{2}\right) T_x T_z \,.
\end{aligned}
\tag{67}
$$

We point out a subtlety in the above definition. Let us define $\omega(\alpha, \beta) = T_\alpha T_\beta T_\alpha^{-1} T_\beta^{-1}$, where $\omega(\alpha, \beta) \in U_s(1)$. $\omega(\alpha, \beta)$ is not an invariant quantity: by redefining generator $T_{\alpha/\beta} \to \varphi_{\alpha/\beta} T_{\alpha/\beta}$, with $\varphi_{\alpha/\beta} \in U_s(1)$, $\omega(\alpha, \beta)$ changes to $\omega(\alpha, \beta) \cdot \varphi_\alpha^2 \varphi_\beta^{-2}$. Instead, $\omega(x, y) \cdot \omega(y, z) \cdot \omega(z, x)$ is an invariant quantity. In this case, this quantity equals to $U_s(\pi)$, and it is natural to interpret it as an odd number of background external monopoles in one unit cell.

We have shown that for system with symmetry group defined in Eq. (67) and a single Kramers doublet per unit cell, we are able to construct strong bosonic SPT phases in 3+1D with external Kramers doublet monopoles. One may wonder if it is possible to have higher order SPT phases in this system. The answer is no. As shown in Section 3.2, high order SPT phases can be constructed by layering lower dimensional (1D or 2D) SPT in a way preserving lattice symmetries. These 1D/2D SPT phases are classified by the second/third group cohomology of symmetry $U_s(1) \times Z_2^{\mathcal{T}}$, which are

$$
\begin{aligned}
H^2[U_s(1) \times Z_2^{\mathcal{T}}, U_{\mathcal{T}}(1)] &= Z_2^2 \,, \\
H^3[U_s(1) \times Z_2^{\mathcal{T}}, U_{\mathcal{T}}(1)] &= Z_1 \,.
\end{aligned}
\tag{68}
$$

We conclude 2nd order SPT phases are not possible due to the vanishing $H^3$. It is easy to check 3rd order SPT phases are also not possible in this system: to preserve translation symmetry of cubic lattice, we should always decorate some 1D SPT on even number of links with the same end point. Due to the $Z_2$ nature of the classification, the end point should support linear representation of $U_s(1) \times Z_2^{\mathcal{T}}$, which contradicts with the fact that there is one Kramers doublet per site.

We also perform a spectral-sequence calculation to confirm this result. To simplify the calculation, we replace the $U_s(1)$ symmetry by a $Z_8$ symmetry. This replacement can be understood physically as breaking the $U_s(1)$ symmetry to $Z_8$. The strong SPT state in $H^4[U_s(1) \times Z_2^{\mathcal{T}}, U_{\mathcal{T}}(1)]$ remains non-trivial, and becomes a strong SPT state in $H^4[Z_8 \times Z_2^{\mathcal{T}}, U_{\mathcal{T}}(1)]$. For this simplified symmetry group, which is discrete, the spectral sequence reviewed in Appendices B and C is computed using the free resolution constructed in Ref. [61]. The calculation reveals that there is a non-trivial $d_3$ map on the third page, pointing from the aforementioned strong-SPT class in $H^4[Z_8 \times Z_2^{\mathcal{T}}, U_{\mathcal{T}}(1)]$ to the anomaly in $H^2[Z_2^{\mathcal{T}}, U_{\mathcal{T}}(1)] \subset H^2[Z_8 \times Z_2^{\mathcal{T}}, U_{\mathcal{T}}(1)]$, representing the anomaly of one Kramer's doublet per magnetic unit cell. This $d_3$ map proves an (1st-order) SPT-LSM Theorem relating the anomaly to the 3D strong SPT class.

# 5 Conclusion and future directions

In this paper, we present a general theoretical framework for LSM-type theorems for bosonic SPT phases through a real-space construction, and also describe a general approach to con-

struct new SPT-LSM theorems from known results of more conventional LSM theorems. Our main results are summarized as below:

1. Topological crystalline phases can be constructed by symmetrically decorating SPT phases on all cells. $n$th order SPT phases are constructed by non-trivial decorations on $(d-m+1)$-cells and trivial decorations on all higher dimensional cells.

2. For a given symmetry group, systems are classified according to patterns of fractional spins (projective representations of local Hilbert spaces). For a given pattern, only certain symmetric decorations of SPT phases are allowed, and can be calculated using algorithm provided in Section 3.2. Different patterns of fractional spins support different decorations of SPT phases. We point out that there exist certain fractional spin patterns, where *no such decorations are allowed*. These patterns actually give the conventional LSM systems, and no SRE phases are allowed in such systems (see more discussion on Appendix C).

3. On one hand, many SPT phases can be obtained by condensing topological excitations from symmetric gauge theories. On the other hand, symmetric gauge theories on conventional LSM systems are anomalous, in the sense one can never obtain symmetric SRE phases from condensing topological excitations. By making use of these two facts, we design a way to obtain a large class of SPT-LSM systems from conventional LSM systems by properly modifying lattice symmetries for various dimensions.

The real-space construction method presented here are quite general, and may be applied to many other contexts. A natural future direction is to generalize the real-space construction to fermionic symmetry protected topological phases, and classify possible SPT-LSM theorems. Some partial results along this line have been obtained for rotational symmetries [62].

As pointed in Ref. [28], the idea of real space construction can also be used in classifying symmetry enriched topological (SET) phases. While for the case of SPT phases, real-space construction fits perfectly in the mathematical framework of equivariant cohomology and spectral sequence, it is unclear how to implement an algorithm to construct/classify SET phases, especially for the case where symmetry operations permute anyons.

We may also apply this idea to coupled wire construction. While coupled wire methods usually breaks lattice rotational symmetries, it seems possible to have a more symmetric construction based on the real space construction of these LSM-SPT systems. This approach may potentially leads to a better understanding of symmetry implementations on gapless systems.

From a practical point of view, while this paper focuses on possible phases given the global symmetries and fractional spins, it is desirable to find a microscopic Hamiltonian to realize these SPT phases. For example, in Section 4.5, we found LSM theorems for SPT phases in 3D, through dyon condensation in a parent $U(1)$ spin liquid. An important question is to construct a realistic Hamiltonian, similar to the 2D model, to realize this scenario, and make connections with candidate materials for $U(1)$ spin liquids. And in order to simulate these spin models, it is important to construct variational wavefunctions for the SPT phases in the SPT-LSM system. While tensor network constructions for SPT phases on systems with integer spins are obtained in Ref [33], where cohomology data can be extracted from tensor equations, it is unclear how to relate tensor equations to spectral sequence on fractional spin systems. We will leave it as a future project.

# Acknowledgement

Shenghan thanks Lesik Motrunich, Xie Chen, Xu Yang, and Peng Ye for helpful discussions on 3D SPT phase and monopole condensation. This work is supported by the Institute for

Quantum Information and Matter, an NSF Physics Frontiers Center, with support of the Gordon and Betty Moore Foundation (SJ), NSF under award number DMR-1653769 (YML) and DMR-1846109 (MC). MC is also supported by Alfred P. Sloan Research Fellowship. YQ acknowledges support from Minstry of Science and Technology of China under grant numbers 2015CB921700, and from National Science Foundation of China under grant number 11874115. This work was performed in part at Aspen Center for Physics, which is supported by National Science Foundation grant PHY-1607611. MC and YML also thanks hospitality of CMSA program "Topological Aspects of Condensed Matter" at Harvard University, where a part of this work was performed.

*Note added:* We would like to draw the readers' attention to a related work by Dominic Else and Ryan Thorngren, to appear in the same arXiv posting.

# A    Group cohomology and bosonic SPT phases protected by onsite symmetry

In this part, we briefly review group cohomology theory, and its application to the (partial) classification and construction of bosonic onsite SPT phases. Bosonic SPT phases involving lattice symmetries will be discussed in Appendix B.

## A.1    Mathematical definition of group cohomology

There are many equivalent definitions of group cohomology. In this paper, we mainly use definition based on the homogeneous cochains. A $n$-cochain $\phi$ for group $G$ with coefficient $M$ in is a function that maps $(n+1)$-tuple $(g_0, g_1, \cdots, g_n)$ of elements of $G$, to an abelian group $M$:

$$\phi(g_0, \cdots, g_n) \in M, \tag{69}$$

and we require this function to be invariant under $G$-action: $g \circ \phi = \phi$, $\forall g \in G$.

The definition of $G$-action on $\phi$ is based on $G$-action on $(g_0, g_1, ..., g_n)$, where

$$g \cdot (g_0, g_1, \cdots, g_n) = (g g_0, g g_1, \cdots, g g_n), \quad \forall g \in G \tag{70}$$

as well as $G$-action on $M$, labeled as $\rho$, which is required to be compatible with group operation of $M$:

$$\rho(g)(m_1 + m_2) = \rho(g)m_1 + \rho(g)m_2, \ \forall g \in G, \ \ m_1, m_2 \in M. \tag{71}$$

Then, $G$-action is defined "diagonally" on $\phi$:

$$(g \circ \phi)(g_0, g_1, \cdots, g_{d+1}) \equiv \rho(g)\phi(g^{-1}g_0, g^{-1}g_1, \cdots, g^{-1}g_n). \tag{72}$$

Thus, invariance of homogeneous n-cochain $\phi$ under $\rho(g)$ action is expressed as

$$\rho(g)\phi(g_0, g_1, \cdots, g_n) = \phi(g g_0, g g_1, \cdots, g g_n). \tag{73}$$

For most cases considered in this paper, $M$ is chosen to be $U(1)$. Since our convention for abelian group $M$ is addition instead of group multiplication, $U(1)$ group elements are treated as phase angles modulo $2\pi$.

Action of $G$ on $M$ is usually given by three $\mathbb{Z}_2$ gradings of the symmetry group. First, we use $\rho_{\mathcal{T}}(g) = \pm 1$ to denote whether $g$ is antiunitary operation (e.g. time reversal): $\rho_{\mathcal{T}}(g) = 1 \, (-1)$ if $g$ is unitary (antiunitary). Second, $\rho_P(g) = \pm 1$ to denote whether $g$ reverses the spatial

orientation: a proper transformation, including a translation, a rotation and a skew rotation, has $\rho_P(g) = 1$; an improper transformation, including a mirror-reflection, a 3D inversion and a glide reflection, has $\rho_P(g) = -1$. Finally, we use $\rho_{P\mathcal{T}}$ to denote $\rho_{P\mathcal{T}}(g) = \rho_P(g)\rho_{\mathcal{T}}(g)$. We also use $M_{\mathcal{T}}$, $M_P$ and $M_{P\mathcal{T}}$ to denote coefficient with the corresponding symmetry actions: $g \in G$ acts as a unitary (antiunitary) operator on coefficients in $M_X$ if $\rho_X(g) = \pm 1$, respectively.

All n-cochains form an abelian group, which is equipped with trivial $G$-action, labeled as $C^n[G, M_X]$. We now define a coboundary map $\mathrm{d}^n$ from $n$-cochain $C^n[G, M_X]$ to $(n+1)$-cochain $C^{n+1}[G, M_X]$ as

$$\mathrm{d}^n \phi(g_0, \cdots, g_{n+1}) = \sum_{k=0}^{p+1} (-1)^k \phi(g_0, \cdots, \hat{g}_k, \cdots, g_{n+1}), \tag{74}$$

where $\hat{g}_k$ means the element $g_k$ is skipped. The superscript $n$ in $\mathrm{d}^n$ denotes the cochain space it acts upon, and we often omit it when it can be determined from the context.

The coboundary operator satisfies the condition

$$\mathrm{d}^n \mathrm{d}^{n-1} = 0, \tag{75}$$

which can be verified straightforwardly. Linked by $\mathrm{d}^n$, the cochain spaces form a cochain complex:

$$\cdots \to C^{n-1}[G, M_X] \xrightarrow{\mathrm{d}^{n-1}} C^n[G, M_X] \xrightarrow{\mathrm{d}^n} C^{n+1}[G, M_X] \to \cdots, \tag{76}$$

where we set $C^n[G, M_X] = 0$ for $n < 0$.

We define n-cocycle $Z^n[G, M_X] \equiv \ker \mathrm{d}^n$ and n-coboundary $B^n[G, M_x] \equiv \mathrm{imag}\, \mathrm{d}^{n-1}$. According to Eq. (75), $B^n[G, M_X] \subseteq Z^n[G, M_X] \subseteq C^n[G, M_X]$. The group cohomology of $G$ is defined as a subquotient abelian group of $C^n[G, M_X]$:

$$H^n[G, M_X] = Z^n[G, M_X]/B^n[G, M_X]. \tag{77}$$

## A.2 Bosonic SPT phase from group cohomology

In this part, we use group cohomology to construct fix point wavefunction for bosonic SPT phase. We focus on onsite symmetry group $SG_0$ with finite number of elements.

Let us start with a $d$-dimensional lattice with a triangularization and a branching structure. The vertices of the lattice is organized to $d$-dimensional simplices (lines in 1D, triangles in 2D and tetrahedral in 3D). The branching structure is a set of orientations on all links between vertices, satisfying the condition that the links do not form any oriented loop. The branching structure can be obtained by first labelling all vertices with ordered numbers and then choose the link orientation from the vertex labeled by a smaller number to the vertex labeled by a larger number.

We then build a physical system in this triangulated lattice. The Hilbert space for this system is formed by local Hilbert spaces on each vertex, spanned by basis vector $|g_i\rangle$ for every $g_i \in SG_0$. And symmetry action is defined as $g|g_i\rangle = |gg_i\rangle$.

We focus on the case where the underlying manifold is closed, i.e. it has no boundary. Given $\phi \in C^{d+1}[SG_0, U(1)_{\mathcal{T}}]$, we construct a physical wavefunction by equal weight superposition of all configurations, and the phase factor for each phase is given by summation of contributions from all $d$-simplex, as shown in the following:

$$|\Psi[\phi]\rangle = \sum_{\{g_i\}} \prod_{\Delta_{i_0 \cdots i_d}} \exp\left[is(i_0, \cdots, i_d)\phi(1, g_{i_0}, \cdots, g_{i_d})\right] |g_1, g_2, \cdots, g_N\rangle. \tag{78}$$

Here, $N$ is number of vertices, and $i_0 < i_1 < \cdots < i_d$ labels ordered vertices of some $d$-simplex. The product runs over all $d$-dimensional simplices in the lattice. $s(i_0, \cdots, i_d) = \pm 1$ denotes whether the orientation of the simplex is the same or opposite to an overall orientation of the underlying manifold. The orientation of simplex $\Delta_{i_0 \dots i_d}$ is determined by its branching structure: a $d$-dimensional local coordinate system is determined as

$$\{\vec{e}_1, \cdots, \vec{e}_d\} \equiv \{\overrightarrow{i_0 i_1}, \cdots, \overrightarrow{i_{d-1} i_d}\}, \tag{79}$$

and this local coordinate $\{\vec{e}_i\}$ defines the orientation of the simplex.

Under $g_0 \in SG_0$ action, we have

$$g_0|\Psi[\phi]\rangle = \sum_{\{g_i\}} \prod_{\Delta_{i_0 \dots i_d}} \exp\left[i s(i_0, \cdots, i_d) \phi(g_0, g_{i_0}, \cdots, g_{i_d})\right] |g_1, g_2, \cdots, g_N\rangle, \tag{80}$$

which can be derived from Eq. (73). Notice that the absence of $\rho_T(g_0)$ here is due to an additional complex conjugate action on i when $g_0$ is antiunitary.

It is easy to see that a generic cochain $\phi$ breaks symmetry. Yet if $\phi \in Z^{d+1}[SG_0, U(1)_{\mathcal{T}}]$ which satisfies $d\phi = 0$, $|\Psi[\phi]\rangle$ is invariant under $SG_0$ [9].

We mention that one is able to construct an exact solvable Hamiltonian with $|\Psi[\phi]\rangle$ serving as ground state wavefunction, and is uniquely determined on any closed manifold.

One may wonder if there is one-to-one correspondence between cocycles and SPT phases. The answer is negative: different cocycles may describe the same SPT phase. In particular, two cocycles differ by a coboundary $\varphi \in B^{d+1}[SG_0, U(1)_{\mathcal{T}}]$ characterize the same SPT phases. To see this, we can put different cocycles on nearby $d$-simplex. If these two cocycles describe the same SPT phases, we should be able to gap out the boundary modes at the interphase of these two simplices. In the next part, we will show that this statement is true iff two cocycles differ by a coboundary.

## A.3 Boundary between two SPT phases

Let us consider systems containing two SPT phases and study the boundary state between these two phases. These two SPT phases are generated by $(d+1)$-cocycles $\phi_1$ and $\phi_2$, and they live in region $B_1$ and $B_2$ respectively, where the whole manifold is formed by $B_1 \cup B_2$. We assume $B_1$ and $B_2$ has the same orientation as the underlying manifold. The interface, labeled as $\partial B_1 = B_1 \cap B_2$, is composed by $(d-1)$-simplices, and its orientation is induced by $B_1$. A generic boundary state can be generated by attaching some $d$-cochain $\varphi$ to $\partial B_1$.

Wavefunction for the whole system contains contributions from $\phi_1$, $\phi_2$, and $\varphi$, which can be written as

$$|\Psi[\phi_1, \phi_2, \varphi]\rangle = \sum_{\{g_i\}} \Phi_{bulk}^{\phi_1}(\{g_i | i \in B_1\}) \Phi_{bulk}^{\phi_2}(\{g_i | i \in B_2\}) \Phi_{bdry}^{\varphi}(\{g_i | i \in \partial B_1\}) |g_1, g_2, \cdots, g_N\rangle. \tag{81}$$

The bulk wavefunction reads

$$\Phi_{bulk}^{\phi_t}(\{g_i | i \in B_t\}) = \prod_{\Delta_{i_0 \dots i_d} \in B_t} \exp\left[i s(i_0, \cdots, i_d) \phi_t(1, g_{i_0}, \cdots, g_{i_d})\right], \quad t = 1, 2, \tag{82}$$

where $s(i_0, \cdots, i_d) = \pm 1$ when the orientation of simplex $\Delta_{i_0 \dots i_d}$ has the same/opposite orientation of $B_t$.

The boundary wavefunction is

$$\Phi_{bdry}^{\varphi}(\{g_i | i \in \partial B_1\}) = \prod_{\Delta_{i_0 \dots i_{d-1}} \in \partial B_1} \exp\left[i s(i_0, \cdots, i_{d-1}) \varphi(1, g_{i_0}, \cdots, g_{i_{d-1}})\right]. \tag{83}$$

Here, $s(i_0, \cdots, i_{d-1}) = \pm 1$ when orientation of $(d-1)$-simplex $\Delta_{i_0 \dots i_{d-1}}$ is consistent/inconsistent with the induced orientation of boundary $\partial B_1$.

For simplicity, let us consider the case where the underlying manifold is a $d$-sphere, and its triangulation is given by faces of a single $(d+1)$-simplex, which contains $d+2$ number of $d$-simplices. SPT phase generated by $\phi_1$ sits on simplex $\Delta_{12\dots d+1}$, while SPT phase generated by $\phi_2$ occupies other $d$-simplices. Using the cocycle condition, we are able to simplify the bulk wavefunction amplitude for state $|g_1, \cdots, g_{d+2}\rangle$ as

$$\Phi_{bulk}^{\phi_1} \Phi_{bulk}^{\phi_2}(g_1, \cdots, g_{d+2}) = \exp\left[i\left(\phi_1(1, g_1, \cdots, g_{d+1}) - \phi_2(1, g_1, \cdots, g_{d+1})\right)\right] . \tag{84}$$

While the boundary wavefunction can be simplified as

$$\Phi_{bdry}^{\varphi}(g_1, \cdots, g_{d+2}) = \exp\left[i(d\varphi(1, g_1, \cdots, g_{d+1}) - \varphi(g_1, \cdots, g_{d+1}))\right] . \tag{85}$$

If $\phi_1$ and $\phi_2$ generate the same SPT phases, we should be able to gap out boundary modes at the interface $B_1 \cap B_2$. In other words, there exists a $(d-1)$-cochain $\varphi$ such that wavefunction $|\Psi[\phi_1, \phi_2, \varphi]\rangle$ is invariant under $SG_0$ for the case where $\phi_1$ and $\phi_2$ belong to the same class. By equating $g|\Psi\rangle$ and $|\Psi\rangle$ and plug in Eq. (84) and Eq. (85), we obtain the following condition for symmetric gapped boundary modes

$$\phi_1 - \phi_2 = d\varphi . \tag{86}$$

Here, we also use the fact that $\varphi(g_1, \cdots, g_d)$ is symmetric under $SG_0$ action from homogeneous cochain condition in Eq. (73).

In other words, the symmetric gapping condition means that $\phi_1$ and $\phi_2$ can only differ by a coboundary: they are in the same cohomology class. The above calculation can be generalized to an arbitrary $d$-dimensional triangulated manifold. One can show that once Eq. (86) is satisfied, the wavefunction is invariant under $SG_0$.

Notice that when $\phi_1$ and $\phi_2$ belong to different cohomology class, Eq. (86) has no solution, and the boundary state must be either gapless or break symmetry, and thus wavefunctions generated by $\phi_1$ and $\phi_2$ belong to different SPT phases. So, we reach the conclusion that cohomology group $H^{d+1}[SG_0, U(1)_{\mathcal{T}}]$ gives a classification of SPT phases.

# B  Classification and construction for bosonic topological crystalline phases by equivariant cohomology

Topological crystalline phases are defined as SPT phases involving spatial symmetries. In this appendix, we introduce the mathematical framework named as equivariant cohomology to construct and classify bosonic topological crystalline phases [28–30].

The outline of this appendix is as following. We first carefully define systems supporting fixed point wavefunction for bosonic topological crystalline phases. We then construct a large class of symmetric wavefunctions using simplex-dependent phase factors. We will see that equivariant cohomology naturally pops up when we imposing symmetry constraint on the wavefunction, which has a physical interpretation as real-space construction. To get classification from those fixed point wavefunction, we should be able to identify when two wavefunctions are in the same phase. So, we discuss the meaning of trivial topological crystallines wavefunctions in this settings, and any two wavefunctions differ by a trivial wavefunction should be treated as in the same class. Finally, we introduce more formal mathematical languages for the above constructions.

### B.1 Defining the lattice system

#### B.1.1 Triangulated lattice and boundary operator

We consider a triangulated $d$-dimensional lattice $Y$ with branching structure. We define the set $Y_p$ as

$$Y_p = \{p - \text{dimensional simplices belongs to } Y\} . \tag{87}$$

For example, $Y_0$ is the collection of all sites (vertices), and $Y_1$ is the collection of all links. For $p < 0$ or $p > d$, $Y_p$ is defined as empty set.

We use $\{1, \cdots, N\}$ to label sites (vertices) in this system, and $N$ is the total number of sites. This labelling naturally induce a branching structure: the orientation of a given link is fixed as from site labeled by a small number to site labeled by a large number. An element of $Y_p$ consists of $p + 1$ vertices. For $p$-simplex $\Delta^p$ with vertices $\{i_0, \cdots, i_p\}$ ($i_0 < \cdots < i_p$), we label it as $[i_0 \cdots i_p]$. Orientation of $[i_0 \cdots i_p]$ is determined by local coordinate induced by branching structure, as shown in Eq. (79). We use $-[i_0 \cdots i_p]$ to denote the simplex which reverses orientation from the simplex $[i_0 \cdots i_p]$.

A free abelian group, labeled as $C_p(Y)$, is generated by elements of $Y_p$. It is defined as

$$C_p(Y) = \left\{ \sum_{\Delta^p \in Y_p} a_\Delta \, |\Delta^p\rangle \; \middle| \; a_\bullet \in \mathbb{Z} \right\}, \tag{88}$$

where summation of simplex is understood as a formal sum. Here we use Dirac's bra-ket notation to label elements of this group.

For later convenience, we also define the dual bra space $\widetilde{C}_p(Y)$ as free abelian group generated by basis $\langle \Delta^p |$ for each $\Delta^p \in Y_p$. Here, the dual basis $\langle \Delta^p |$ is determined by its inner product to all ket states:

$$\langle \Delta_1^p | \Delta_2^p \rangle = \delta_{\Delta_1^p, \Delta_2^p} . \tag{89}$$

For $p > d$ or $p < 0$, $C_p(Y)$ ($\widetilde{C}_p(Y)$) is identified as the trivial group with only identity element.

Let us also introduce boundary operators, which is defined as

$$\begin{aligned} \partial^p : \; & C_p(Y) \to C_{p-1}(Y) \\ & [i_0 \ldots i_p] \mapsto \sum_k (-1)^k [i_0 \ldots \hat{i}_k \ldots i_p]. \end{aligned} \tag{90}$$

The geometric meaning for $\partial^p$ is clear: for a given $p$-simplex, it picks out all its boundary $(p-1)$-simplex. The additional $(-)^k$ in the above expression make sure that orientation of the boundary $(p-1)$-simplex is consistent with orientation of $p$-simplex $[i_0 \ldots i_p]$. It is then straightforward to check that $\partial^{p-1} \partial^p = 0$.

Action of boundary operator $\partial$ on the bra space follows Eq. (90):

$$\begin{aligned} \partial^p : \; & \widetilde{C}_{p-1}(Y) \to \widetilde{C}_p(Y) \\ & \langle \Delta^{p-1} | \mapsto \sum_{\Delta^p} \langle \Delta^{p-1} | \partial^p | \Delta^p \rangle \langle \Delta^p | . \end{aligned} \tag{91}$$

Namely, $|\Delta^{p-1}\rangle$ gives summation of all $p$-simplices that intersect at $\langle \Delta^{p-1} |$. In the following, we also use the left/right action of boundary operator to distinguish its action on ket/bra state: $\partial \Delta^p \equiv \partial^p |\Delta^p\rangle$ and $\Delta^{p-1} \partial \equiv \langle \Delta^{p-1} | \partial^p$.

### B.1.2 Global symmetry and local Hilbert space

Now, let us assign global symmetry group $SG$ and local Hilbert spaces on this triangulated lattice system. We focus on the case where $SG$ is a discrete group, which includes both onsite and lattice symmetry.

For an arbitrary $p$-simplex $\Delta^p \in Y_p$, we define $SG_{\Delta^p}$ as the subgroup that maps $\Delta^p$ to itself while preserving orientation of $\Delta^p$. Notice that internal symmetry, labeled as $SG_0$, is always a normal subgroup of $SG_{\Delta^p}$.

The triangulation as well as the branching structure is chosen to be invariant under $SG$ action. Namely, for any $g \in SG$ and $\Delta_1^p \in Y_p$, there exists $\Delta_2^p \in Y_p$, such that $\Delta_2^p = g(\Delta_1^p)$, without additional minus sign. Furthermore, we choose triangulation such that $SG_{\Delta^p}$ to be a pointwise action on $\Delta^p$. That is to say, $SG_{\Delta^p}$ acts as an onsite symmetry group locally on $\Delta^p$. Under this choice, for $\Delta^p \in Y_p$ and $\Delta^{p-1} \in Y_{p-1}$, if $\langle \Delta^{p-1} | \partial \Delta^p \rangle \neq 0$, then $SG_{\Delta^p} \subseteq SG_{\Delta^{p-1}}$. And for any $d$-simplex $\Delta^d$, we have $SG_{\Delta^d} = SG_0$.

We then define local Hilbert spaces on this triangulated lattice. Local Hilbert spaces live on sites, and dimension of each local Hilbert space equals $|SG|$, defined as the number of elements in $SG$. For local Hilbert space living on site $[i]$, its basis vector is labeled by group elements as $|g\rangle_{[i]}$ with $g \in SG$.

Action of symmetry $g \in SG$ is defined as

$$g|g_i\rangle_{[i]} = |g g_i\rangle_{g([i])}, \quad \forall g_i \in SG. \tag{92}$$

Notice that lattice symmetry both acts on internal degree of freedom as well as moves the site.

### B.2 Fixed point wavefunctions for topological crystalline phases

Let us now construct quantum states on this lattice system. We focus on a special class of wavefunctions, which are equal weight amplitude superposition of basis states $|g_1, g_2, \cdots, g_N\rangle$, $\forall g_i \in SG$. Here, $|g_1, g_2, \cdots, g_N\rangle$ is a shorthand for $|g_1\rangle_{[1]} \otimes \cdots \otimes |g_N\rangle_{[N]}$. Intuitively, this kind of wavefunctions, when respecting symmetry, can be interpreted as condensation of all possible domain wall configurations. And we expect they belong to symmetric phases rather than spontaneously symmetry breaking phases (namely, they are not cat states).

Since all configurations has the same weight amplitude, different quantum phases are distinguished by phase factors of different configurations. We will focus on the case where phase factors for a given configuration is determined by local quantum state of every simplex. Namely, the phase factor for configuration $|g_1, g_2, \cdots, g_N\rangle$ can be factorized to phase factors from every simplex.

In this case, the most generic wavefunction reads

$$|\Psi[\phi]\rangle = \sum_{\{g_i\}} \prod_{p=0}^{d} \prod_{[i_0 \cdots i_p] \in Y_p} \exp\left[ i\, \phi_{[i_0 \cdots i_p]}(1, g_{i_0}, \cdots, g_{i_p}) \right] |g_1, g_2, \cdots, g_N\rangle. \tag{93}$$

Here, $\phi$ is a mapping from a $p$-simplex $[i_0 \cdots i_p]$ and a $(p+2)$-tuple of group elements $(g_0, g_{i_0}, \cdots, g_{i_p})$ to a phase factor $\phi_{[i_0 \cdots i_p]}(g_0, g_{i_0}, \cdots, g_{i_p}) \in [0, 2\pi)$ for any $0 \leq p \leq d$.

In Eq. (93), the first argument of $\phi_{[i_0 \cdots i_p]}$ is fixed to be identity, which seems to be redundant and can be moved away. Yet as we will see later, this argument will be useful when we impose symmetry constraint.

Now, let us consider symmetry action on $|\Psi[\phi]\rangle$. Under $g \in SG$ action, we have

$$g|\Psi[\phi]\rangle = \sum_{\{g_i\}}\prod_{p=0}^{d}\prod_{[i_0\cdots i_p]}\exp\Big[\mathrm{i}\,\phi_{[i_0\cdots i_p]}(1, g_{i_0}, \cdots, g_{i_p})\Big]|gg_1\rangle_{g([1])}\otimes\cdots\otimes|gg_N\rangle_{g([N])}$$

$$= \sum_{\{g_i\}}\prod_{p=0}^{d}\prod_{[i_0\ldots i_p]}\exp\Big[\rho_{\mathcal{T}}(g)\mathrm{i}\,\phi_{g^{-1}([i_0\ldots i_p])}(1, g^{-1}g_{i_0}, \cdots, g^{-1}g_{i_p})\Big]|g_1, \cdots, g_N\rangle$$

$$= \sum_{\{g_i\}}\prod_{p=0}^{d}\prod_{[i_0\cdots i_p]}\exp\Big[\mathrm{i}\,\phi_{[i_0\cdots i_p]}(g, g_{i_0}, \cdots, g_{i_p})\Big]|g_1, \cdots, g_N\rangle. \tag{94}$$

The last line follows from the definition below:

$$\phi_{[i_0\cdots i_p]}(g, g_{i_0}, \cdots, g_{i_p}) \triangleq \rho_{\mathcal{T}}(g)\phi_{g^{-1}([i_0\cdots i_p])}(1, g^{-1}g_{i_0}, \cdots, g^{-1}g_{i_p}). \tag{95}$$

This definition can be expressed as the homogeneous condition for $\phi$, which reads

$$\rho_{\mathcal{T}}(g)\phi_{[i_0\cdots i_p]}(g_0, g_1, \cdots, g_{p+1}) = \phi_{g([i_0\cdots i_p])}(gg_0, gg_1, \cdots, gg_{p+1}). \tag{96}$$

We call such $\phi$ equivariant cochain. We will discuss it in detail in Appendix B.4.

We require wavefunction defined in Eq. (94) to be invariant under $g \in SG$ action (up to a $U(1)$ phase factor): $g|\Psi[\phi]\rangle = \exp[\mathrm{i}\,\alpha(g)]|\Psi[\phi]\rangle$. Here, $\alpha$ form a 1D representation for $SG$.

Symmetric condition puts following constraints on $\phi$:

$$\sum_{p=0}^{d}\sum_{[i_0\cdots i_p]\in Y_p}\phi_{[i_0\cdots i_p]}(g-1, g_{i_0}, \ldots, g_{i_p}) = \alpha(g), \tag{97}$$

for an arbitrary configuration $\{g_1, \ldots, g_N\}$. Here we define

$$\phi(\ldots, g \pm h, \ldots) \equiv \phi(\ldots, g, \ldots) \pm \phi(\ldots, h, \ldots), \tag{98}$$

where $g \pm h$ is understood as formal summation/subtraction, and should not be confused with multiplication operation of group elements $gh$.

In order to solve Eq. (97), we define two operators acting on $\phi$ to simplify the equation. The first operator is labeled as $\mathrm{d}_\wedge$, which is the analog of the coboundary operator for group cochain defined in Eq. (74). Action of $\mathrm{d}_\wedge$ on $\phi$ reads

$$(\mathrm{d}_\wedge\phi_{[i_0\ldots i_p]})(g_0, \cdots, g_{p+2}) \equiv \sum_{k=0}^{p+2}(-1)^k\phi_{[i_0\ldots i_p]}(g_0, \cdots, \hat{g}_k, \cdots, g_{p+2}). \tag{99}$$

And it is straightforward to verify that $d_\wedge^2 = 0$.

The second operator, labeled as $\mathrm{d}_>$, is induced by boundary operator defined in Eq. (91). For an arbitrary $p$-simplex $\Delta^p$, we define $(\mathrm{d}_>\phi)_{\Delta^p}$ as

$$(\mathrm{d}_>\phi)_{\Delta^p} \equiv \phi_{\Delta^p\partial}, \tag{100}$$

where $\Delta^p\partial = \langle\Delta^p|\partial^{p+1} \in \widetilde{C}_{p+1}(Y)$. Here, the right side of Eq. (100) follows the following definition: for any $\sum_i a_i\Delta_i^p \in \widetilde{C}_p(Y)$, we define

$$\phi_{\sum_i a_i\Delta_i^p} = \sum_i a_i\phi_{\Delta_i^p}, \quad a_i \in \mathbb{Z}. \tag{101}$$

The physical meaning of $d_>$ can be interpreted as following: it produce boundary modes at the $p$-dimensional interface of several $(p+1)$-simplices.

By inserting definition of boundary operator in Eq. (90, 91), we obtain the explicit expression for Eq. (100) as

$$(d_> \phi)_{[i_0 \dots i_p]}(g_0, \cdots, g_{p+2}) \equiv \sum_{k=0}^{p+1} \sum_{\substack{i_{k-1}<j<i_k \\ [\dots i_{k-1},j,i_k \dots] \in Y_{p+1}}} (-1)^k \phi_{[\dots i_{k-1},j,i_k \dots]}(g_0, \cdots, g_{p+2}). \quad (102)$$

Notice that for a $d$-simplex $\Delta^d$, we have $(d_> \phi)_{\Delta^d} = 0$. It is easy to verify that $d_>$ is also satisfies coboundary condition: we have $d_>^2 = 0$, which is induced by $\partial^2 = 0$.

By using these two new operators defined in Eq. (99) and Eq. (102), we are able to simplify Eq. (97) as following

$$\sum_p \sum_{[i_0 \cdots i_p]} \phi_{[i_0 \cdots i_p]}(g-1, g_{i_0}, \dots, g_{i_p})$$

$$= \sum_p \sum_{[i_0 \dots]} \left\{ d_\wedge \phi_{[i_0 \cdots i_p]}(1, g, g_{i_0}, \cdots, g_{i_p}) - \sum_{k=0}^p (-1)^k \phi(1, g, g_{i_0}, \cdots, \hat{g}_{i_k}, \cdots, g_{i_p}) \right\}$$

$$= \sum_{p=0}^{d-1} \sum_{[i_0 \dots]} \left\{ \left( (d_\wedge - d_>)\phi \right)_{[i_0 \dots i_p]}(1, g, g_{i_0}, \cdots, g_{i_p}) \right\} - \sum_i \phi_{[i]}(1, g)$$

$$= \alpha(g) \bmod 2\pi \quad (103)$$

for any $|g_1, g_2, \cdots, g_N\rangle$ state.

The above equation is satisfied if we require

$$\left( (d_\wedge - d_>)\phi \right)_{\Delta^p}(1, g, g_0, \cdots, g_p) = f_{\Delta^p}(1, g), \quad \forall \Delta^p \in Y_p, \forall g_i \in SG, \ 0 \le p \le d, \quad (104)$$

where $f_{\Delta^p}$ is a function depending only on the first two arguments of $\phi_{\Delta^p}$. Notice that $f$ also satisfies homogeneous condition in Eq. (105): $\rho_{\mathcal{T}}(g)f_{\Delta^p}(g_1, g_2) = f_{g(\Delta^p)}(gg_1, gg_2)$.

Here, we focus on the simple case where $f = 0$. In this case, we obtain

$$(d_\wedge - d_>)\phi = 0. \quad (105)$$

Mathematically, it is equivalent to say that $\phi$ belongs an equivariant cocycle $Z_{SG}^{d+1}[X; U(1)_{PT}]$, where $X$ is the dual lattice of $Y$. We will discuss this in full details in Appendix B.4.

Quantum number of $g$ action can be easily extracted as

$$\exp[i\,\alpha(g)] = \exp\left[ i \sum_{i=1}^N \phi_{[i]}(1, g) \right]. \quad (106)$$

As a consistency check, let us prove that phase factor $\alpha$ forms a 1D representation of $SG$. For action of $g_2 \cdot g_1$, $\alpha$ should satisfy the equation $\alpha(g_2) + \rho_{\mathcal{T}}(g_2)\alpha(g_1) = \alpha(g_1 g_2)$. To see this, we insert Eq. (106)

$$\alpha(g_2) + \rho_{\mathcal{T}}(g_2)\alpha(g_1) = \sum_i \left( \phi_{[i]}(1, g_2) + \rho_{\mathcal{T}}(g_2)\phi_{[i]}(1, g_2) \right)$$

$$= \sum_i \left( \phi_{[i]}(1, g_2) + \phi_{g_2([i])}(g_2, g_2 g_1) \right)$$

$$= \sum_i \left( d_\wedge \phi_{[i]}(1, g_2, g_2 g_1) + \phi_{[i]}(1, g_2 g_1) \right)$$

$$= \sum_i (d_> \phi)_{[i]}(1, g_2, g_2 g_1) + \alpha(g_2 g_1)$$

$$= \alpha(g_2 g_1), \quad (107)$$

where we use Eq. (96) to obtain the third line, and use Eq. (105) to obtain the fifth line. And the last equation is due to

$$\sum_{[i]} (\mathrm{d}_{>}\phi)_{[i]} = \phi_{\sum_i [i] \partial} = 0 \,, \tag{108}$$

according to the definition of boundary operator $\partial$.

Now, let us give physical interpretation for Eq. (105). We claim that it is related to the real space construction of SPT phases. To see this, we first consider Eq. (105) for a $d$-simplex $\Delta^d$. In this case, the equivariant cocycle condition becomes

$$\mathrm{d}_{\wedge}\phi_{\Delta^d} = 0 \,. \tag{109}$$

Readers may notice that the above equation looks similar to the group cocycle condition.

However, there is an important difference from group cocycle. Notice that the homogeneous condition defined in Eq. (96) is different from the usual homogeneous condition for group cochain defined in Eq. (73). In particular, when restrict on $p$-simplex $\Delta^p$ and its little group $SG_{\Delta^p}$, Eq. (96) becomes

$$\rho_{\mathcal{T}}(g)\phi_{\Delta^p}(g_0, g_1, \cdots) = \phi_{\Delta^p}(gg_0, gg_1, \cdots), \quad \forall g \in SG_{\Delta^p} \,, \tag{110}$$

where $g_0, g_1, \cdots$ take value in $SG$, while $g$ take value in $SG_{\Delta^p}$. Here, $\phi_{\Delta^p}$ satisfying Eq. (110) is called $SG$-valued $SG_{\Delta^p}$ group $(p+1)$-cochain (labeled as $C_{SG}^{p+1}[SG_{\Delta^p}, U(1)_{\mathcal{T}}]$).

Even with this difference, we can still interpret Eq. (109) as decorating $d$-simplex $\Delta^d$ by a $d$-dimensional SPT phase protected by $SG_{\Delta^d}$. And the homogeneous condition in Eq. (96) relates decorations on all symmetry related $d$-simplices. Roughly speaking, one should decorate "the same SPT phases" on lattice symmetry related $d$-simplices [3]. However, Eq. (109) does not put any constraint on the decoration of $d$-simplices that are not related by any symmetries.

We then move to other constraint imposed by Eq. (105). On an arbitrary $p$-simplex $\Delta^p$ with $p < d$, the equivariant cocycle condition reads

$$[(\mathrm{d}_{\wedge} - \mathrm{d}_{>})\phi]_{\Delta^p} = \mathrm{d}_{\wedge}\phi_{\Delta^p} - \phi_{\Delta^p \partial} = 0 \,. \tag{111}$$

This constraint can be interpreted as "no-open-edge" condition, as we will explained.

Remember that $\Delta^p$ is "the common edge" of several $(p+1)$-simplices, which we label as $\Delta_i^{p+1}$, with $i = 1, 2, \cdots, n$. According to definition of boundary operator $\partial$, we have $\Delta^p \partial = \sum_{i=1}^{n} (\pm)\Delta_i^{p+1}$, where $\pm$ depends on orientation. Then, the second term $\phi_{\Delta^p \partial}$, is interpreted as the gapless edge mode on $\Delta^p$, generated by bulk wavefunctions $\phi_{\Delta_i^{p+1}}$. Thus, symmetric condition Eq. (111) simply means that this edge mode can be gapped out by a symmetric mass term on $\Delta^p$, which is expressed as $\mathrm{d}_{\wedge}\phi_{\Delta^p}$.

We point out that Eq. (111) puts constraint both on symmetric mass term $\mathrm{d}_{\wedge}\phi_{\Delta^p}$ as well as bulk wavefunctions $\phi_{\Delta_i^{p+1}}$. For example, when $p = d-1$, Eq. (111) tells us that if $\Delta_1^d$ and $\Delta_2^d$ share a common $(d-1)$-dimensional boundary, then the decoration on these two cells should belong to the same SPT phase: they differ at most by a coboundary.

There is an especially interesting case, where $\phi_{\Delta^{p'}}$ vanishes for all $\Delta^{p'}$ with $p' > p$. In this case, the equivariant cocycle condition for $\Delta^p$ simply becomes

$$\mathrm{d}_{\wedge}\phi_{\Delta^p} = 0 \,. \tag{112}$$

Then, $\phi_{\Delta^p}$ should be an $SG$-valued $SG_{\Delta^p}$ $p+1$-cocycle. Namely, decoration on $\Delta^p$ is interpreted as $p$-dimensional SPT phases, which would give $(p-1)$-dimensional edge modes. This phenomena is actually the defining feature for $(d+1-p)$th order SPT phases.

---

[3]For symmetry related simplex $\Delta_1$ and $\Delta_2 = g(\Delta_1)$, $SG_{\Delta_1}$ and $SG_{\Delta_2}$ are in general different. Yet they are isomorphic to each other by relation $SG_{\Delta_2} = g \cdot SG_{\Delta_1} \cdot g^{-1}$. And their decoration are related by Eq. (96), which can be interpreted as decorating the same SPT phases.

## B.3 Mod out equivalent classes

In the last part, we show that for any $\phi$ satisfying Eq. (105) (known as equivariant cocycle condition), we are able to construct a symmetric fixed point wavefunction, which has physical interpretation related to real space construction. Distinct $\phi$'s in general give distinct wavefunction. To classify bosonic topological crystalline phases, we should be able to identify in which case different $\phi$'s actually represent the same phases. In other words, we need to figure out definition of equivariant coboundary, and then the classification is given by cocycle mod out coboundary.

Physically, states generated by equivariant coboundary is adiabatic connecting to vacuum, which can be formulated using the "bubble equivalence" picture [27,40]. We start by decorating trivial SPT state on every simplex. For $p$-simplex $\Delta^p$ with $0 \leq p \leq d$, the decoration can be written as $d_\wedge \varphi_{\Delta^p}$, where $\varphi_{\Delta^p}$ is a $(p-1)$-cochain satisfying

$$\rho_{\mathcal{T}}(g)\varphi_{\Delta^p}(g_0, g_1, \cdots, g_p) = \varphi_{g(\Delta^p)}(gg_0, gg_1, \cdots, gg_p). \tag{113}$$

When restricting on $p$-simplex and its little group $SG_{\Delta^p}$, $\varphi_{\Delta^p}$ is an $SG$-valued $SG_{\Delta^p}$ group $p$-cochain: $\varphi_{\Delta^p} \in C^p_{SG}[SG_{\Delta^p}, U(1)_{\mathcal{T}}]$.

Decoration of trivial SPT phases on neighbouring cells will produce gapless edge modes at their interfaces. Formally, for a $p$-simplex $\Delta^p$ ($p < d$), which is the common edge of some $(p+1)$-d simplices, the edge mode is written as $(d_> d_\wedge \varphi)_{\Delta^p} = (d_\wedge \varphi)_{\Delta^p \partial}$. It is easy to see that this edge mode can be gapped out by symmetric mass terms.

Thus, wavefunction at $\Delta^p$, labeled as $\phi^0_{\Delta^p}$, contains two contributions: trivial SPT decorated at $\Delta^p$ as well as symmetric mass terms. Mathematically, we have

$$\phi^0 = (d_\wedge + d_>)\varphi. \tag{114}$$

And the fixed point wavefunction $|\Psi(\phi^0)\rangle$ is generated using Eq. (93). We claim that $\phi^0$ defined in the above equations gives equivariant coboundary.

As a consistency check, we will show wavefunction $|\Psi(\phi^0)\rangle$ is symmetric under $SG$. To see that, we check the symmetric condition in Eq. (105):

$$\begin{aligned} (d_\wedge - d_>)\phi^0 &= (d_\wedge - d_>)(d_\wedge + d_>)\varphi \\ &= d^2_\wedge \varphi - d^2_> \varphi + (d_\wedge d_> - d_> d_\wedge)\varphi = 0. \end{aligned} \tag{115}$$

Here, to obtain the last line, we use coboundary condition $d^2_\wedge = d^2_> = 0$ as well as identity $d_\wedge d_> = d_> d_\wedge$.

## B.4 Mathematical formulation

In this part, we provide a more mathematical formulation for the classification and construction of bosonic topological crystalline states. As we will see, the construction of symmetric wavefunctions discussed in the last two parts naturally fits to the framework of equivariant cohomology [34,63]. We also provide an algorithm, known as spectral sequence method, to solve equivariant cohomology.

### B.4.1 Dual lattice formulation

In order to be more consistent with the convention in mathematical literature, we construct dual lattice $X$ for the $d$-dimensional lattice $Y$. When $Y$ is a triangulated space, $X$ becomes a trivalent lattice. The collection of $p$-cells in $X$ is labeled as $X_p$, and we use $\bar{\Delta}^p$ to label the element in $X_p$. By definition, there is a one-to-one correspondence between $\Delta^p_i \in Y_p$ and $\bar{\Delta}^{d-p}_i \in X_{d-p}$.

Orientation of $\bar{\Delta}^{d-p}$ is induced by orientation of $\Delta^p$ in the following way. Remember that orientation of a manifold is determined by chirality of local coordinates of this manifold. We denote the local coordinate for $\Delta^p$ as $\{\vec{e}_1, \cdots, \vec{e}_p\}$. For triangulated lattice, this local coordinate is induced by its branching structure, as shown in Eq. (79). Then, the local coordinate $\{\vec{e}'_1, \cdots, \vec{e}'_{d-p}\}$ for the dual cell $\bar{\Delta}^{d-p}$ is chosen such that the combination $\{\vec{e}_1, \cdots, \vec{e}_p, \vec{e}'_1, \cdots, \vec{e}'_{d-p}\}$ matches orientation of the underlying $d$-dimensional manifold.

Then, action of symmetry $g \in SG$ on $X$ is induced by action of $g$ on $Y$. By construction, for $\Delta_1^p \in Y_p$ and $g \in SG$, we can find $\Delta_2^p = g(\Delta_1^p)$ for some $\Delta_2^p \in Y_p$. Correspondingly, for $\bar{\Delta}_{1,2}^{d-p} \in X_{d-p}$, we have $\bar{\Delta}_2^{d-p} = \rho_P(g) g(\bar{\Delta}_1^{d-p})$, where $\rho_P(g) = \pm 1$ for orientation preserving (reversing) symmetry $g$.

We mention that although we focus on the case where $Y$ is a triangulated space (and $X$ is trivalent), equivariant cohomology is defined in more general context. For example, Ref. [28, 30] consider an $SG$-symmetric cellular decomposition of the underlying manifold, which includes triangulated space as a special case.

### B.4.2  Double cochain complex and equivariant cohomology

In the dual lattice $X$, local Hilbert spaces are associated with elements of $X_d$, with basis states $|g\rangle$ labeled by $g \in SG$. Following similar procedure in Appendix B.2, fixed point wavefunctions are generated by function $\phi$: given any $\bar{\Delta}^p \in X_p$ and quantum state $|g_1, \ldots, g_{d+1-p}\rangle$ associated with $\bar{\Delta}^p$, $\phi_{\bar{\Delta}^p}(1, g_1, \ldots, g_{d+1-p})$ provides a $U(1)$ phase factor.

We consider those $\phi$'s belong to equivariant $(d+1)$-cochains defined in Eq. (96), . Collection of equivariant $(d+1)$-cochains forms an Abelian group, labeled as $C_{eqv}^{d+1}$, where the group multiplication rule is given by

$$(\phi^1 + \phi^2)_{\bar{\Delta}^{d-p}}(g_0, \cdots, g_{p+1}) \equiv \phi_{\bar{\Delta}^{d-p}}^1(g_0, \cdots, g_{p+1}) + \phi_{\bar{\Delta}^{d-p}}^2(g_0, \cdots, g_{p+1}), \tag{116}$$

for any $0 \le p \le d$.

It is convenient to decompose $\phi$ in the following way:

$$\phi = \bigoplus_{p=0}^{d} \phi^{p+1,d-p}, \tag{117}$$

with

$$\phi^{p,q} : SG^{p+1} \times X_q \to U(1)$$
$$\left((g_0, \cdots, g_p), \bar{\Delta}^q\right) \mapsto \phi_{\bar{\Delta}^q}^{p,q}(g_0, \ldots, g_p). \tag{118}$$

Clearly, collection of $\phi^{p,q}$ for fixed $p$ and $q$ also forms an Abelian group $C^{p,q}$ induced by $U(1)$. Thus, we have

$$C_{eqv}^{p+q} = \bigoplus_{p,q \in \mathbb{Z}} C^{p,q}. \tag{119}$$

And $C^{p,q}$ is set to be zero (group with only identity element) when $p < 0$ or $q < 0$.

Now, let us study the structure of $C^{p,q}$ in more detail. We claim that equipped with coboundary operator $d_\wedge$ and $d_>$, $C^{p,q}$ becomes a double cochain complex. Namely, by fixing $q$ and focusing on function acting on $(p+1)$-tuple of group elements, we obtain a group cochain complex $C^{\bullet,q}$ induced by $d_\wedge$, while by fixing $p$ and focusing on function acting on $X_q$, we obtain another cochain complex $C^{p,\bullet}$ induced by $d_>$.

In the following, let us study these two cases separately.

1. First, we consider the case with fixed $q$ and varying $p$.

Function acting on $(p+1)$-tuples of group elements induced by $\phi^{p,q} \in C^{p,q}$ is defined as

$$\phi_{\#}^{p,q} : (g_0, \cdots, g_p) \mapsto \phi_{\#}^{p,q}(g_0, \cdots, g_p) \in \bigoplus_{j=1}^{N_q} U(1), \tag{120}$$

where $N_q$ is number of elements in $X_q$. $\phi_{\#}(g_0, \cdots, g_{p+1})$ is a $q$-cell dependent phase factors, where $U(1)$ phase on a $q$-cell $\bar{\Delta}^q$ is given as $\phi_{\bar{\Delta}^q}(g_0, \cdots, g_{p+1})$.

Coboundary map increasing $p$ is identified as $\mathrm{d}_\wedge$ in Eq. (99), whose definition reads

$$\mathrm{d}_\wedge^p : \ C^{p,q} \to C^{p+1,q},$$

$$\phi_{\#}^{p,q}(g_0, \cdots, g_p) \mapsto \sum_{k=0}^{p+1} (-1)^k \phi_{\#}^{p,q}(g_0, \cdots, \hat{g}_k, \cdots, g_{p+1}). \tag{121}$$

As discussed before, this operator satisfies the condition $\mathrm{d}_\wedge^{p+1} \mathrm{d}_\wedge^p = 0$, which makes $C^{\bullet,q}$ a cochain complex linked by $d_\wedge$ for fixed $q$. In fact, it is a group cochain complex $C^\bullet[SG, M_q]$, where the coefficient $M_q$ for this group cochain complex is identified as the $q$-cell dependent phase factors:

$$M_q = \bigoplus_{j=1}^{N_q} U(1). \tag{122}$$

For a complete characterization of this group cochain, we should figure out the group action on $M_q$, which should be consistent with the homogeneous condition in Eq. (96). Let us define the action of $g \in SG$ on $M_q$ as following: for $f_\# \in M_q$, and $\bar{\Delta}^q \in X_q$, $g$ action reads

$$g : f_{\bar{\Delta}^q} \mapsto \rho_{P\mathcal{T}}(g) f_{g^{-1}(\bar{\Delta}^q)}, \tag{123}$$

where $f_{-\bar{\Delta}^q} \equiv -f_{\bar{\Delta}^q}$.

By definition, group cochain $\phi^{p,q}$ should be invariant under the diagonal group actions on $M_q$ and $(p+1)$-tuples of group elements:

$$\begin{aligned}
\phi_{\bar{\Delta}^q}^{p,q}(g_0, \cdots, g_p) &= (g \circ \phi^{p,q})_{\bar{\Delta}^q}(g_0, \cdots, g_p) \\
&= \rho_{P\mathcal{T}}(g) \phi_{g^{-1}(\bar{\Delta}^q)}^{p,q}(g^{-1}g_0, \cdots, g^{-1}g_p).
\end{aligned} \tag{124}$$

In direct lattice, the above equation becomes

$$\phi_{\Delta^{d-q}}^{p,q}(g_0, \cdots, g_p) = \rho_{\mathcal{T}}(g) \phi_{g^{-1}(\Delta^{d-q})}^{p,q}(g^{-1}g_0, \cdots, g^{-1}g_q), \tag{125}$$

which is indeed consistent with homogeneous condition in Eq. (96). Here, the absence of $\rho_P(g)$ in Eq. (125) is due to the fact that under orientation reversing symmetry action, $\bar{\Delta}^q$ obtain an extra minus while $\Delta^{d-q}$ does not.

To further explore the structure of $C^\bullet[SG, M_q]$, let us consider a fixed $\bar{\Delta}^q \in X_q$. Coboundary operator for $\phi_{\bar{\Delta}^q}^{p,q}$ is induced by definition in Eq. (121), which makes $\phi_{\bar{\Delta}^q}^{p,q}$ a group $p$-cochain. Notice that although $\phi_{\bar{\Delta}^q}^{p,q}$ takes value in elements of $SG$, it only satisfies homogeneous condition for $g_s \in SG_{\Delta^{d-q}}$ action. It makes $\phi_{\bar{\Delta}^q}^{p,q}$ an $SG$-valued $SG_{\Delta^{d-q}}$-cochain, with the coefficient identified as $U(1)_{\mathcal{T}}$. And the collection of cochain $\phi_{\bar{\Delta}^q}^{p,q}$ forms an Abelian group $C_{SG}^p[SG_{\Delta^{d-q}}, U(1)_{\mathcal{T}}]$.

Once cochain $\phi^{p,q}_{\bar{\Delta}^q}$ is fixed, all lattice symmetry related cochains $\phi^{p,q}_{g(\bar{\Delta}^q)}$ can be generated by Eq. (124). We then define $\Sigma_q$ as a representative set for orbits $X_q/G$. Namely, $\Sigma_q$ is a maximal set of symmetry independent elements of $X_q$. According to the above discussion, $\phi^{p,q}_{\#}$ is determined by $\left\{ \phi^{p,q}_{\bar{\Delta}^q} \middle| \bar{\Delta}^q \in \Sigma_q \right\}$. In other words, cochain complex $C^\bullet[SG, M_q]$ is isomorphic to the following decomposition:

$$C^\bullet[SG, M_q] \simeq \bigoplus_{\bar{\Delta}^q \in \Sigma_q} C^\bullet_{SG}[SG_{\Delta^{d-q}}, U(1)_{\mathcal{T}}]. \tag{126}$$

Equipped with coboundary operator $d_\wedge$, we are able to define group cocycle $Z^\bullet[SG, M_q]$, group coboundary $B^\bullet[SG, M_q]$ as well as group cohomology $H^\bullet[SG, M_q] \equiv Z^\bullet[SG, M_q]/B^\bullet[SG, M_q]$. Following similar argument, $H^\bullet[SG, M_q]$ also has the following decomposition:

$$H^\bullet[SG, M_q] \simeq \bigoplus_{\bar{\Delta}^q \in \Sigma_q} H^\bullet_{SG}[SG_{\Delta^{d-q}}, U(1)_{\mathcal{T}}]. \tag{127}$$

2. To identify cochain complex with fixed $p$ and varying $q$, let us consider $C_q(X)$, which is the free abelian group generated by $X_q$:

$$C_q(X) = \left\{ \sum_{\bar{\Delta}^q \in X_q} a_{\bar{\Delta}^q} \, |\bar{\Delta}^q\rangle \,\middle|\, a_\bullet \in \mathbb{Z} \right\}. \tag{128}$$

We define boundary operator $\bar{\partial}^q$ acting on $C_q(X)$ as

$$\langle \bar{\Delta}^{q-1}|\bar{\partial}^q|\bar{\Delta}^q\rangle \equiv \langle \Delta^{d-q}|\partial^{d-q+1}|\Delta^{d-q+1}\rangle. \tag{129}$$

Superscripts of $\bar{\partial}^q$ are often omitted when it can be determined from the context. Apparently, we have $\bar{\partial}^{q-1} \circ \bar{\partial}^q = 0$

To see the cochain complex structure of $C^{p,\bullet}$, we first point out that $\phi^{p,q} \in C^{p,q}$ can also be viewed as function defined on $X_q$

$$\phi^{p,q} : \ \bar{\Delta}^q \mapsto \phi^{p,q}_{\bar{\Delta}^q}. \tag{130}$$

We then define coboundary operator on this function $d_> : C^{p,q} \to C^{p,q+1}$ as following:

$$\left( d_>^q \phi^{p,q} \right)_{\bar{\Delta}^{q+1}} = (-1)^{p+q-d} \phi^{p,q}_{\partial \bar{\Delta}^{q+1}}. \tag{131}$$

It is straightforward to verify that $d_>^{q+1} d_>^q = 0$, which makes $C^{p,\bullet}$ a cochain complex (but not a group cochain complex).

Here, the definition of $d_>$ here differs from Eq. (100) by a phase factor. Due to this phase factor, we deduce that $d_\wedge$ and $d_>$ anticommute with each other:

$$(d_\wedge^p d_>^q + d_>^q d_\wedge^p)\phi^{p,q} = 0. \tag{132}$$

In summary, $C^{p,q}$ can be viewed as a double cochain complex: by fixing $q$ and varying $p$, we obtain a group cochain complex $C^{\bullet,q}[SG, M_q]$ induced by $d_\wedge$, while for the other case with fixing $p$ and varying $q$, we obtain another cochain complex induced by $d_>$.

We then define the "total coboundary operator" $d^n$ acting on $C_{eqv}^n = \bigoplus_{p \in \mathbb{Z}} C^{p,n-q}$ as

$$d^n = \bigoplus_p (d_\wedge^p + d_>^{n-p}) : C_{eqv}^n \to C_{eqv}^{n+1}. \tag{133}$$

By equation $d_\wedge^2 = d_>^2 = 0$ and $d_\wedge d_> + d_> d_\wedge = 0$, it is easy to verify that

$$d^{n+1} d^n = 0. \tag{134}$$

So, $C_{eqv}^\bullet$ forms a cochain complex linked by the total coboundary operator d. The cohomology group for this cochain complex is named as equivariant cohomology, which is defined as

$$H_{SG}^n[X, U(1)_{P\mathcal{T}}] \equiv \ker d^n / \operatorname{imag} d^{n-1}. \tag{135}$$

As we show in the fixed point wavefunction construction, the equivariant cohomology group $H_{SG}^{d+1}[X, U(1)_{P\mathcal{T}}]$ classifies SPT phases protected by $SG$ for systems on $d$-dimensional lattice $X$. When $X$ is a lattice system defined on $\mathbb{R}^d$, $H_{SG}^{d+1}[X, U(1)_{P\mathcal{T}}] = H^{d+1}[SG, U(1)_{P\mathcal{T}}]$ [63]. This is consistent with the classification result of bosonic crystalline phases obtained in Ref. [33,34].

### B.4.3 Real space construction and spectral sequence

Equivariant cohomology group not only gives a classification of bosonic topological crystalline phases, but also provides real space constructions of these phases. In this part, we discuss in detail about real space constructions, which arise naturally when one tries to solve equivariant cohomology equations. Mathematically, real space constructions are closely related to the spectral sequence method.

To get a better understanding for the structure of equivariant cochain, we introduce a two dimensional network representation for double cochain complex $C^{\bullet,\bullet}$ as following:

$$
\begin{array}{ccccccccc}
& & \vdots & & \vdots & & \vdots & & \\
& & \uparrow & & \uparrow & & \uparrow & & \\
\cdots & \to & C^{p+1,q-1} & \xrightarrow{d_>} & C^{p+1,q} & \xrightarrow{d_>} & C^{p+1,q+1} & \to & \cdots \\
& & \uparrow {\scriptstyle d_\wedge} & & \uparrow {\scriptstyle d_\wedge} & & \uparrow {\scriptstyle d_\wedge} & & \\
\cdots & \to & C^{p,q-1} & \xrightarrow{d_>} & C^{p,q} & \xrightarrow{d_>} & C^{p,q+1} & \to & \cdots \\
& & \uparrow {\scriptstyle d_\wedge} & & \uparrow {\scriptstyle d_\wedge} & & \uparrow {\scriptstyle d_\wedge} & & \\
\cdots & \to & C^{p-1,q-1} & \xrightarrow{d_>} & C^{p-1,q} & \xrightarrow{d_>} & C^{p-1,q+1} & \to & \cdots \\
& & \uparrow & & \uparrow & & \uparrow & & \\
& & \vdots & & \vdots & & \vdots & &
\end{array}
\tag{136}
$$

This network is bounded from left and below: $C^{p,q} = 0$ for $p < 0$ or $q < 0$.

Then, the $n$th equivariant cochain $C_{eqv}^n = \bigoplus_{p \in \mathbb{Z}} C^{p,n-p}$, which is the direct sum of diagonal elements in the above network. And $\phi^{(n)} \in C_{eqv}^n$ is decomposed as $\phi^{(n)} = \bigoplus_p \phi^{p,n-p}$ with $\phi^{p,n-p} \in C^{p,n-p}$. Under this decomposition, the equivariant cocycle condition $d\phi^{(n)} = 0$ reads

$$d_\wedge \phi^{p,n-p} + d_> \phi^{p+1,n-p-1} = 0, \quad \forall p \in \mathbb{Z}. \tag{137}$$

And the equivariant coboundary condition $\phi^{(n)} = d\phi^{(n-1)}$ reads

$$\phi^{p,n-p} = d_\wedge \phi^{p-1,n-p} + d_> \phi^{p,n-p-1}, \quad \forall p \in \mathbb{Z}. \tag{138}$$

Since network in Eq. (136) is bounded by zeros from both left and below, for $p = n$, the cocycle condition becomes $d_\wedge \phi^{n,0} = 0$, and the coboundary condition reads $\phi^{n,0} = d_\wedge \phi^{n-1,0}$.

In the following, let us provide an algorithm, known as the spectral sequence method, to solve these equations to obtain $H_{SG}^{d+1}[X, U(1)_{P\mathcal{T}}]$.

The strategy is to solve different dimensional decoration separately. For a fix $d_0$ with $0 \leq d_0 \leq d$, we focus on a special case where $\phi^{d_0+1,d-d_0} \neq 0$ and $\phi^{p+1,d-p} = 0$ for any $p > d_0$. Then, the decomposition of $\phi^{(d+1)}$ becomes

$$\phi^{(d+1)} = \bigoplus_{0 \leq p \leq d_0} \phi^{p+1,d-p}. \tag{139}$$

The collection of $\phi^{(d+1)}$ satisfying the above equation are labeled as $S^{d_0+1,d-d_0}$, which is a subset of $C_{eqv}^{d+1}$. Physically, solutions for equivariant cocycle/coboundary equations within $S^{d_0+1,d-d_0}$ give $(d + 1 - d_0)$th order SPT phases.

Before moving on, let us comment on the simplest case with $p = -1$. Remember $X$ is the dual lattice for $d$ dimensional lattice $Y$, and thus we have $X_{d+1} = 0$, which gives $\phi^{0,d+1} = 0$. We can ignore $\phi^{0,d+1}$ and focus on $\phi^{p+1,d-p}$ with $p > 0$.

To obtain solutions for Eq. (137) and Eq. (138) within $S^{d_0+1,d-d_0}$, the key step is to solve constraints imposing on $\phi^{d_0+1,d-d_0}$. These constraints can be solved using spectral sequence method [30, 63], as we will explain in detail in the following.

**First page**

We first consider the cocycle/coboundary conditions for $p = d_0$, which read:

$$d_\wedge \phi^{d_0+1,q_0} = 0 \text{ (cocycle)},$$
$$\phi^{d_0+1,q_0} = d_\wedge \phi^{d_0,q_0} \text{ (coboundary)}, \tag{140}$$

where $q_0 = d - d_0$. Remember that $d_\wedge$ is a group coboundary operator, which acts on group cochain complex $C^\bullet[SG, M_{q_0}]$, with $M_{q_0}$ defined in Eq. (122). Solution of the above cocycle (coboundary) equation is actually group cocycle (coboundary) $Z^{d_0+1}[SG, M_{q_0}]$ ($B^{d_0+1}[SG, M_{q_0}]$). And we obtain group cohomology classification from these two equations, as

$$H^{d_0+1}[SG, M_q] \simeq \bigoplus_{\bar{\Delta}^q \in \Sigma_q} H_{SG}^{d_0+1}[SG_{\Delta^{d-q}}, U(1)_{\mathcal{T}}], \tag{141}$$

where the identity follows Eq. (127).

The physical meaning for $H^{d_0+1}[SG, M_q]$ is interpreted as following. We choose a representative set of $X_{q_0}/SG$ as $\Sigma_{q_0} \subseteq X_{q_0}$, where any two elements of $\Sigma_q$ cannot be related by lattice symmetry. We then decorated every $q_0$-cell in $\Sigma_q$ with some $d_0$ dimensional SPT phases. Decorations of SPT phases on $q_0$-cells beyond $\Sigma_q$ are generated by lattice symmetry, whose action is defined in Eq. (124).

$H^{d_0+1}[SG, M_q]$ is also known as first page of degree $(d_0 + 1, q_0)$, labeled as $E_1^{d_0+1,q_0}$.

For convenience, we also define $C^{p,q}$ as zeroth page, labeled as $E_0^{p,q}$. Then $d_\wedge$ can be viewed as coboundary operators defined on zeroth page, relabeled as $d_0$:

$$d_0^{p,q} : E_0^{p,q} \rightarrow E_0^{p+1,q}. \tag{142}$$

We define cocycle and coboundary for $d_0$ as

$$Z_1^{p,q} \equiv \ker d_0^{p,q} = Z^p[SG, M_q],$$
$$B_1^{p,q} \equiv \operatorname{imag} d_0^{p-1,q} = B^p[SG, M_q]. \tag{143}$$

Thus, first pages can be viewed as cohomology based on zeroth page and $d_0$ as

$$E_1^{p,q} = \ker d_0^{p,q}/\operatorname{imag} d_0^{p-1,q} = Z_1^{p,q}/B_1^{p,q}. \tag{144}$$

**Second pages**

Yet, not all elements in first pages give a consistent solution for Eq. (137) and Eq. (138).

For equivariant cocycle, we also require that elements in $Z_1^{d_0+1,q_0}$ also satisfy "the second level" cocycle condition, which reads

$$d_\wedge \phi^{d_0,q_0+1} + d_> \phi^{d_0+1,q_0} = 0. \tag{145}$$

Physically, this cocycle equation is interpreted as "no-open-edge" condition discussed after Eq. (111): when several $d_0$-dimensional SPT phases, which live on different $q_0$-cells of dual lattice, meet at the $(q_0+1)$-cell interface, we should be able to gap out those $(d_0-1)$-dimensional edge states. Only a subset of $\ker d_0^{d_0+1,q_0}$ satisfies Eq. (145), and is named as $Z_2^{d_0+1,q_0}$.

Similarly, coboundary condition in Eq. (140) does not give the most general equivalence relation. We express "the second level" coboundary condition as following:

$$\phi^{d_0+1,q_0} = d_> \phi^{d_0+1,q_0-1} + d_\wedge \phi^{d_0,q_0},$$
$$d_\wedge \phi^{d_0,q_0} = 0. \tag{146}$$

And $\phi^{d_0+1,q_0}$'s generated by "the second level" coboundary equation should also be treated as trivial decoration on $q_0$-cells of dual lattice. As discussed in Appendix B.3, the coboundary equation denotes "bubble equivalence": it says that those $(q_0+1)$th order SPT phases should be considered as trivial, when they can be constructed by decorating trivial $(d_0+1)$-dimensional SPT phases in $(q_0-1)$-cell on dual lattice. We define Abelian group $B_2^{d_0+1,q_0} \subseteq C^{d_0+1,q_0}$, whose elements give consistent solution for Eq. (146). By definition, we have $B_1^{d_0+1,q_0} \subseteq B_2^{d_0+1,q_0}$.

We define the second page $E_2^{d_0+1,q_0}$ as

$$E_2^{d_0+1,q_0} = Z_2^{d_0+1,q_0}/B_2^{d_0+1,q_0}. \tag{147}$$

The physical meaning of $E_2^{d_0+1,q_0}$ is clear: elements of $E_2^{d_0+1,q_0}$ are those $d_0$-dimensional SPT decorations that can be gapped on $Y_{d_0-1}$ yet cannot be trivialized by $(d_0+1)$-dimensional "trivial SPT bubbles".

**Higher pages**

$E_2^{d_0+1,q_0}$ is a better approximation for equivariant cohomology when comparing to $E_1^{d_0+1,q_0}$. And we are able to obtain even better approximations by adding "$r$th level" cocycle/coboundary conditions for $r \geq 2$. The solution is named as $r$th page $E_r^{d_0+1,q_0}$. When $r \to \infty$, we recover equivariant cohomology equation, and thus the classification of $(q_0+1)$th order SPT on $d$ dimension is given by $\infty$-page, labeled as $E_\infty^{d_0+1,q_0}$.

Let us works out $r$th pages for general $r \geq 0$. The cocycle conditions from the first level to the $r$th level are

$$d_\wedge \phi^{d_0+1,q_0} = 0,$$
$$d_> \phi^{d_0+1,q_0} + d_\wedge \phi^{d_0,q_0+1} = 0,$$
$$\cdots\cdots$$
$$d_> \phi^{d_0-r+3,q_0+r-2} + d_\wedge \phi^{d_0-r+2,q_0-r+1} = 0. \tag{148}$$

Those $\phi^{d_0+1,q_0}$ consistent with above equations forms an Abelian group, labeled as $Z_r^{d_0+1,q_0}$. And we also define $Z_0^{d_0+1,q_0} = C^{d_0+1,q_0}$. We then have $Z_0 \supseteq Z_1 \supseteq \cdots \supseteq Z_\infty$

The physical meaning of $Z_r^{d_0+1,q_0}$ is interpreted as following. Let us use elements in $Z_r^{d_0+1,q_0}$ to decorate $Y_{d_0}$, and leave $Y_{d'}$ undecorated for $d_1 > d_0$. We can then add symmetric mass terms

on $Y_{d_2}$ to avoid gapless modes for $d_2 \geq d_0 - r + 2$. However, we are not guaranteed to be able to add symmetric mass terms on $Y_{d_0-r+1}$ (or lower dimensions), and thus incapable to construct symmetric SRE state.

Similarly, the $r$th level coboundary equations reads

$$0 = \mathrm{d}_\wedge \phi^{d_0+r-1,q_0-r+1},$$
$$\cdots\cdots$$
$$0 = \mathrm{d}_> \phi^{d_0+2,q_0-2} + \mathrm{d}_\wedge \phi^{d_0+1,q_0-1},$$
$$\phi^{d_0+1,q_0} = \mathrm{d}_> \phi^{d_0+1,q_0-1} + \mathrm{d}_\wedge \phi^{d_0,q_0}. \tag{149}$$

Solutions of $\phi^{d_0+1,q_0}$ for these equations form an Abelian group named as $B_r^{d_0+1,q_0}$. We define $B_0^{d_0+1,q_0} = 0$. Then, we have $B_0 \subset B_1 \subset \cdots \subset B_\infty$.

Physically, by decorating $Y_{d_0}$ with elements in $B_r^{d_0+1,q_0}$, we are actually constructing trivial $(q_0 + 1)$th order SPT phases, which can be trivialized by "bubbles" on $Y_{d_0+r-1}$.

Since $C^{p,q} = 0$ when $p < 0$ or $q < 0$, calculation for $Z_r$ ($B_r$) will converge at certain $r_z$ ($r_b$), namely, for $r \geq r_z$ ($r_b$), we have $Z_r = Z_\infty$ ($B_r = B_\infty$). In consequence, we have $E_r = E_\infty$ for $r > \max(r_z, r_b)$, where $E_r \equiv Z_r / B_r$.

**Cohomology of pages**

We mention that the $(r + 1)$th page $E_{r+1}^{d_0+1,q_0}$ can be viewed as cohomology of $r$th pages equipped with coboundary operator $\mathrm{d}_r$, which is induced by "total coboundary operator" $\mathrm{d}$ of double complex.

To see this, let us define Abelian group $S^{p_0,n-p_0} \subset C_{eqv}^n$, where for $\phi^{(n)} \in S^{p_0,n-p_0}$ with decomposition $\phi^{(n)} = \bigoplus_p \phi^{p,n-p}$, we have $\phi^{p,n-p} = 0$ for $p > p_0$ and $\phi^{p_0,n-p_0} \neq 0$.

We then define abelian group $S_r^{d_0+1,q_0} \subset C_{eqv}^{d+1}$ as

$$S_r^{d_0+1,q_0} = \left\{ \phi^{(d+1)} \mid \phi^{(d+1)} \in S^{d_0+1,q_0}, \mathrm{d}\phi^{(d+1)} \in S^{d_0+2-r,q_0+r} \right\}. \tag{150}$$

By definition, we have $S_0^{p,n-p} = S^{p,n-p}$. Also, for $\phi^{(n)} \in S_\infty^{p,n-p}$, it is easy to check $\mathrm{d}\phi^{(n)} = 0$.

Compare Eq. (148) and Eq. (149) with Eq. (150), we conclude that the leading term for $\phi^{(d+1)} \in S_r^{d_0+1,q_0}$ is an element of $Z_r^{d_0+1,q_0}$, and elements of $B_r^{d_0+1,q_0}$ are leading terms for $\phi^{d+1} \in \mathrm{d}S_{r-1}^{d_0+r-1,q_0-r+1}$. Mathematically, the leading terms can be extracted by following expression:

$$Z_r^{d_0+1,q_0} = S_r^{d_0+1,q_0} / S_{r-1}^{d_0,q_0+1}, \qquad B_r^{d_0+1,q_0} = \mathrm{d}S_{r-1}^{d_0+r-1,q_0-r+1} / S_{r-1}^{d_0,q_0+1}. \tag{151}$$

And $r$th page can be written as

$$E_r^{d_0+1,q_0} = \frac{Z_r^{d_0+1,q_0}}{B_r^{d_0+1,q_0}} = \frac{S_r^{d_0+1,q_0}}{\mathrm{d}S_{r-1}^{d_0+r-1,q_0-r+1} + S_{r-1}^{d_0,q_0+1}}, \tag{152}$$

where the plus sign in denominator is understood as the abelian group multiplication operation.

Now, let us act equivariant coboundary operator on $S_r$. According to definition of $S_r$, we have

$$\mathrm{d}S_r^{d_0+1,q_0} \subseteq S_\infty^{d_0-r+2,q_0+r} \subseteq S_r^{d_0-r+2,q_0+r},$$
$$\mathrm{d}\left( \mathrm{d}S_{r-1}^{d_0+r-1,q_0-r+1} + S_{r-1}^{d_0,q_0+1} \right) = \mathrm{d}S_{r-1}^{d_0,q_0+1}. \tag{153}$$

According to Eq. (152), we have

$$E_r^{d_0-r+2,q_0+r} = \frac{S_r^{d_0-r+2,q_0+r}}{\mathrm{d}S_{r-1}^{d_0,q_0+1} + S_{r-1}^{d_0-r+1,q_0+r+1}} \,. \tag{154}$$

Thus, action of equivariant coboundary operator on $S_r$ naturally induce coboundary operator $\mathrm{d}_r$ on $r$th page defined as

$$\mathrm{d}_r^{d_0+1,q_0} : E_r^{d_0+1,q_0} \to E_r^{d_0-r+2,q_0+r} \,, \tag{155}$$

with coboundary condition $\mathrm{d}_r^{d_0+r,q_0-r}\, \mathrm{d}_r^{d_0+1,q_0} = 0$. A schematic representation of $\mathrm{d}_r$ is shown in Fig. 8.

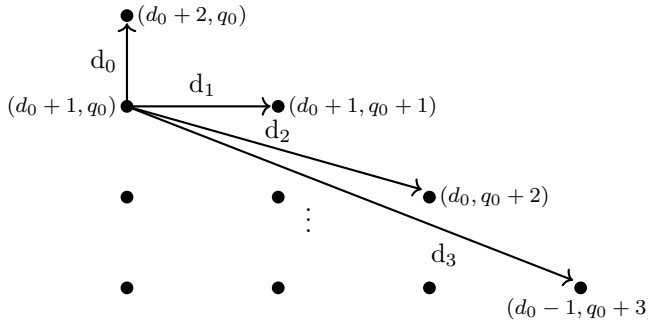

Figure 8: A pictorial representation of coboundary operators $\mathrm{d}_r$ mapping between $r$th pages.

Here, we claim without proof that

$$E_{r+1}^{d_0+1,q_0} = \ker \mathrm{d}_r^{d_0+1,q_0} / \mathrm{imag}\, \mathrm{d}_r^{d_0+r,q_0-r} \,. \tag{156}$$

**Equivariant cohomology and real space constructions from $E_\infty$**

As we shown before, $E_r^{d_0+1,q_0}$ eventually converges to $E_\infty^{d_0+1,q_0}$. By varying $d_0$ from 1 to $d$, we can work out all $E_\infty$. Physically, elements of $E_\infty^{d_0+1,q_0}$ are decorations on $Y_{d_0}$ for $(q_0+1)$th order SPT phases.

Now, let us relate real space construction of SPT phases with $E_\infty$. Let us consider an arbitrary $[\phi^{d_0+1,q_0}] \in E_\infty^{d_0+1,q_0}$, where $[.]$ denotes equivalent class respect to $B_\infty$. By inserting this $\phi^{d_0+1,q_0}$ to Eq. (137) and set all $\phi^{p,d+1-p} = 0$ for $p > d_0 + 1$, we are able to obtain one solution $\phi^{(d+1)}$ with decomposition $\phi^{(d+1)} = \bigoplus_{p>d_0} \phi^{p,d+1-p}$, which gives a real space construction for a $(q_0+1)$th order SPT phase.

One can then add any $\phi^{p,d+1-p}$ $(p \le d_0)$, which satisfies $[\phi^{p,d+1-p}] \in E_\infty^{p,d+1-p}$, to $\phi^{(d+1)}$ obtained above. This gives a distinct $(q_0+1)$th order SPT phases but with the same SPT decoration on $Y_{d_0}$.

One may naively think the equivariant cohomology $H_{SG}^{d+1}[X, U(1)_{P\mathcal{T}}]$ is given by $\bigoplus_p E_\infty^{p,d+1-p}$, however, in general it is not true. Actually, we have the following filtration

$$G^{d+1} \to G^d \to \cdots \to G^1 \to H_{SG}^{d+1}[X, U(1)_{P\mathcal{T}}], \tag{157}$$

where arrows here are inclusion maps, and $G^n$ satisfies $G^i/G^{i+1} = E_\infty^{i,d+1-i}$, and $G^{d+1} \equiv E_\infty^{d+1,0}$. In other words, a $d$-dimensional bosonic SPT phase are labeled by a list of numbers $\vec{\nu} = (\nu^{d+1}, \nu^d, \cdots, \nu^1)$, where $\nu^i \in E_\infty^{i,d+1-i}$.

If $G^i$ is a trivial extension of $G^{i+1}$ for all $i$: $G^i = G^{i+1} \times E_\infty^{i,d+1-i}$, we then have

$$H_{SG}^{d+1}[X, U(1)_{P\mathcal{T}}] = \bigoplus_p E_\infty^{p,d+1-p}. \tag{158}$$

However, there are many cases where $G_i$ is a non-trivial extension of $G^{i+1}$ for certain $i$. In these cases, Eq. (158) does not hold, and the summation rule of $\vec{\nu}$ does not follow the simple vector sum rule. The exact form of mapping $G_{i+1} \to G_i$ cannot be obtained by spectral sequence method.

## C  SPT-LSM systems from equivariant cohomology

In this appendix, based on equivariant cohomology, we discuss the classification and real space construction of SPT phases in SPT-LSM systems.

Consider systems defined on an oriented $d$-dimensional lattice $Y$ with cell decomposition, whose dual lattice is $X$. As in Appendix B, we use $Y_p/X_p$ ($0 \leq p \leq d$) to label the collection of $p$-dimensional cells in $Y/X$. The cell decomposition of $Y$ are chosen such that $Y_p$ is invariant under global symmetry $SG$ for any $p$. For $\Delta_p \in Y_p$, we label the corresponding cell in $X_{d-p}$ as $\bar{\Delta}^{d-p}$

Local Hilbert spaces live on $Y_0/X_d$. As discussed in Section 3.1, the local Hilbert space at site $i \in Y_0$, labeled as $\mathcal{H}_i$, forms a projective representation of its little group $SG_i$, which is classified by $H^2[SG_i, U(1)_{\mathcal{T}}]$. If site $i$ is transformed to site $j$ by symmetry operation $g_0 \in SG$, then $SG_j = g_0 \cdot SG_i \cdot g_0^{-1}$, and their projective representations are related by Eq. (35). Thus, as shown in Section 3.1, the "UV property" of systems on lattice $Y$ with global symmetry $SG$ are classified by Eq. (36). In dual lattice language, the classification reads

$$\bigoplus_{\bar{i} \in \Sigma_d} H^2[SG_i, U(1)_{\mathcal{T}}], \tag{159}$$

where $\Sigma_d$ denotes a representative set for orbits $X_d/SG$.

Comparing the above equation with Eq. (127) and the definition of first page, the "UV property" is actually characterized by $E_1^{2,d}$:

$$E_1^{2,d} = H^2[SG, M_d] \simeq \bigoplus_{\bar{i} \in \Sigma_d} H^2[SG_i, U(1)_{\mathcal{T}}]. \tag{160}$$

Physically, $H^2[SG_i, U(1)_{\mathcal{T}}]$ classify edge states of 1D SPT phase with symmetry $SG_i$. Thus, $E_1^{2,d} = H^2[SG, M_d]$ can also be interpreted as decoration of $X_d/Y_0$ with edge states of 1D SPT phase in a way preserving all lattice symmetries.

One may naively think that the non-trivial decoration on $X_d/Y_0$ can be viewed as edge states of $(d+1)$-dimensional $d$th order SPT phases. Due to LSM anomaly, the $d$-dimensional phase must be either spontaneously symmetry breaking phases, or symmetric long-range entangled phases. However, not all elements in $E_1^{2,d}$ gives non-trivial SPT phases in $H_{SG}^{d+2}[X, U(1)_{P\mathcal{T}}]$, according to our discussion on spectral sequence. Some elements in $E_1^{2,d}$ may be trivialized by $d_r$ defined in Eq. (155) for $r \geq 1$. In other words, some projective representation patterns may corresponds to boundary of a trivial $(d+1)$-dimensional SPT phases, and thus do not have LSM anomaly. In these cases, the projective representations can actually be absorbed by some $d$-dimensional SPT phases. In the following, for a given projective representation pattern, we give an algorithm to determine if it hosts LSM anomaly, and if not, which SPT phases can absorb this pattern.

We use $[\phi^{2,d}]_1 \in E_1^{2,d}$ to label the projective representation pattern, where $\phi^{2,d} \in C^{2,d}$ and $[.]_1$ denotes equivalent class respect to $B_1$ defined in Eq. (143). It is easy to see that $\phi_0^{(d+2)} \equiv \phi^{2,d}$ is an equivariant cocycle satisfying $\mathrm{d}\phi_0^{(d+2)} = 0$. Since $\phi^{(d+2)}$ is defined on a $d$-dimensional lattice $X$, it can be interpreted as the boundary state of a $(d+1)$-dimensional SPT phase.

The corresponding "bulk" $(d+1)$ dimensional SPT phase may be a trivial symmetric SPT phase. In other words, there may exist $\phi_0^{(d+1)}$, such that

$$\phi_0^{(d+2)} = \mathrm{d}\phi_0^{(d+1)}. \tag{161}$$

Assuming $\phi_0^{(d+1)}$ is one solution for this equation, $\phi_0^{(d+1)} + \phi_1^{(d+1)}$ is also a solution if and only if $\forall \phi_1^{(d+1)} \in Z_{SG}^{d+1}[X, U(1)_{P\mathcal{T}}]$. Notice that $\phi_0^{(d+1)} \notin Z_{SG}^{d+1}[X, U(1)_{P\mathcal{T}}]$, meaning SPT phases supported on systems with projective representations can never be realized on systems with local Hilbert spaces as linear representations.

If the above equation has no solution, then the "bulk" $(d+1)$ dimensional SPT phase is non-trivial. In this case, the system has LSM anomaly, which makes it impossible to support SPT phases.

For cases where there exist solutions to Eq. (161), and consider decomposition of an arbitrary solution $\phi_j^{(d+1)}$, which reads

$$\phi_j^{(d+1)} = \bigoplus_{p \leq d_j} \phi_j^{p+1,d-p}. \tag{162}$$

Here, $(d_j + 1, d - d_j)$ is the index for leading term: $\phi^{p+1,d-p} = 0$ for $p > d_j$. Then, $\phi_j^{(d+1)}$ can be interpreted as a $(d - d_j + 1)$th order SPT phase, obtained by symmetrically decorating $Y_{d_j}$ with $d_j$-dimensional SPT phase.

Among all solutions, there is a special type of solutions, whose leading term $\phi^{d_m+1,d-d_m}$ has the smallest index $d_m$. Then, for systems characterized by $[\phi^{2,d}]_1 \in E_1^{2,d}$, the symmetric SRE phase should at least be $(d - d_m + 1)$th order.

To solve Eq. (161) and find $d_m$, we start from decoration of $Y_1$ with 1D SPT: $\phi^{(d+1)} = \phi^{2,d-1} \oplus \phi^{1,d}$, where $[\phi^{2,d-1}]_1 \in E_1^{2,d-1}$. If we are not able to find solution in this form, we then try decoration of $Y_2$ with 2D SPT $\phi^{(d+1)} = \phi^{3,d-2} \oplus \phi^{2,d-1} \oplus \phi^{1,d}$, where $[\phi^{3,d-1}]_1 \in E_1^{2,d-1}$, and so on so forth.

In the following, we will present spectral sequence calculation for some examples from main text.

## C.1 Half-integer spins on honeycomb lattice

Consider a half-integer spin system on honeycomb lattice discussed in Section 4.1. The projective representations of this system are classified by

$$E_1^{2,d} \simeq H^2[SO(3) \times C_3, U(1)], \tag{163}$$

where $C_3$ is the three fold rotation around a honeycomb site, which leaves this site invariant. The system considered in the main text transforms linearly under $C_3$, and transforms projectively under $SO(3)$. As shown in the main text, half-integer spins can be trivialized by symmetrically decorating Haldane chains on links, which gives a second order SPT phase.

## C.2 2D SPT-LSM system with magnetic inversion

Let us consider the example in Section 4.3, where the global symmetry group is $Z_4^{\widetilde{\mathcal{I}}} \times Z_2^{\mathcal{T}}$. The generator of $Z_4^{\widetilde{\mathcal{I}}}$ is the "magnetic inversion" $\widetilde{\mathcal{I}}$, where $\widetilde{\mathcal{I}}^2$ gives an onsite $Z_2$ symmetry action, labeled as $s$. And we require the local spin at inversion center to be a Kramers doublet.

To proceed, it is enough to focus on a small portion of cell decomposition of direct lattice $Y = \mathbb{R}^2$ respecting inversion symmetry, as shown in Fig. 4. The (right) boundary mapping reads

$$
\begin{aligned}
\langle \mu | \partial &= -\langle \tau | - \langle \widetilde{\mathcal{I}} \circ \tau |\,; \\
\langle \tau | \partial &= -\langle \widetilde{\mathcal{I}} \circ \tau | \partial = -\langle \sigma | + \langle \widetilde{\mathcal{I}} \circ \sigma |\,; \\
\langle \sigma | \partial &= \langle \widetilde{\mathcal{I}} \circ \sigma | \partial = 0\,.
\end{aligned}
\tag{164}
$$

Local Hilbert spaces live on 0-cells. In this case, there is a Kramers doublet sitting at $\mu$. Namely, the local Hilbert space form a projective representation $U$, characterized by $[\nu] \in H^2[Z_4^{\widetilde{\mathcal{I}}} \times Z_2^{\mathcal{T}}, U(1)_{\mathcal{T}}]$.

$$
U(g_0) \cdot U(g_1) = \exp[\mathrm{i}\,\nu(1, g_0, g_1)]\, U(g_0 g_1)\,.
\tag{165}
$$

To see the explicit form of $\nu$, let us present symmetry $g$ as $g = \widetilde{\mathcal{I}}^{n_{\widetilde{\mathcal{I}}}(g)} \cdot \mathcal{T}^{n_{\mathcal{T}}(g)}$, where $n_{\widetilde{\mathcal{I}}} \in \{0, 1, 2, 3\}$ and $n_{\mathcal{T}} \in \{0, 1\}$. Then a particular cocycle $\nu$ reads

$$
\nu(g_0, g_1, g_2) = \pi \cdot [n_{\mathcal{T}}(g_0) - n_{\mathcal{T}}(g_1)] \cdot [n_{\mathcal{T}}(g_1) - n_{\mathcal{T}}(g_2)]\,.
\tag{166}
$$

First, it is not hard to see that decoration of 1-cells cannot trivialize the Kramers doublet at $\mu$. One can decorate 1-cells $\tau$ and $\widetilde{\mathcal{I}} \circ \tau$ with 1D SPT protected by $Z_2 \times Z_2^{\mathcal{T}} = \{1, \widetilde{\mathcal{I}}^2\} \times \{1, \mathcal{T}\}$. These two 1D SPT are related by symmetry $\widetilde{\mathcal{I}}$, and thus their edge states should host the same property under $\mathcal{T}$ action: it is not possible to have one Kramers singlet and one Kramers doublet. And when meet at $\mu$ (under $d_>$ operation), one always get Kramers singlet, which contradicts with our settings.

We then consider decoration of 2-cells: $[\phi^{3,0}]_1 \in E_1^{3,0} \simeq H^3[Z_2 \times Z_2^{\mathcal{T}}, U(1)_{\mathcal{T}}]$. A solution for $\phi_\sigma^{3,0}$ reads

$$
\phi_\sigma^{3,0}(g_0, g_1, g_2, g_3) = [n_s(g_0) - n_s(g_1)] \cdot \nu(g_1, g_2, g_3)\,,
\tag{167}
$$

where $g_i = \widetilde{\mathcal{I}}^{n_{\widetilde{\mathcal{I}}}(g_i)} \cdot \mathcal{T}^{n_{\mathcal{T}}(g_i)}$, and $n_s = [n_{\widetilde{\mathcal{I}}}/2] \in \{0, 1\}$, where $[.]$ means take integer part. And $\nu$ is defined in Eq. (166). We point out that $\phi_\sigma^{3,0}$ corresponds to the non-trivial element in $H^1\big[Z_2, H^2[Z_2^{\mathcal{T}}, U(1)_{\mathcal{T}}]\big] \subset H^3[Z_2 \times Z_2^{\mathcal{T}}, U(1)_{\mathcal{T}}]$, which can be interpreted as decorating $Z_2$ domain wall with Haldane chain protected by $\mathcal{T}$.

According to Eq. (96), we have

$$
\begin{aligned}
\phi_{\widetilde{\mathcal{I}} \circ \sigma}^{3,0}(g_0, g_1, g_2, g_3) &= \phi_\sigma^{3,0}\big(\widetilde{\mathcal{I}}^{-1} \circ (g_0, g_1, g_2, g_3)\big) \\
&= \big[n_s\big(\widetilde{\mathcal{I}}^{-1} g_0\big) - n_s\big(\widetilde{\mathcal{I}}^{-1} g_1\big)\big] \cdot \nu(g_1, g_2, g_3)\,.
\end{aligned}
\tag{168}
$$

Then, $\phi^{2,1}$ can be obtained by solving equation $d_\wedge \phi^{2,1} = d_> \phi^{3,0}$. Using Eq. (164), the equivariant coboundary equation on $\tau$ is

$$
d\phi_\tau^{2,1}(g_0, g_1, g_2, g_3) = \big[ -\phi_\sigma^{3,0} + \phi_{\widetilde{\mathcal{I}} \circ \sigma}^{3,0} \big](g_0, g_1, g_2, g_3)\,.
\tag{169}
$$

One solution for the above equation reads

$$
\phi_\tau^{2,1}(g_0, g_1, g_2) = \langle n_{\widetilde{\mathcal{I}}}(g_0) \rangle_2 \cdot \nu(g_0, g_1, g_2)\,,
\tag{170}
$$

where

$$
\langle n \rangle_\alpha = n \bmod \alpha\,.
\tag{171}
$$

And $\phi^{2,1}_{\widetilde{\mathcal{I}}\circ\tau}$ is obtained by $\widetilde{\mathcal{I}}$ action on $\phi^{2,1}_\tau$ as

$$\phi^{2,1}_{\widetilde{\mathcal{I}}\circ\tau}(g_0,g_1,g_2) = \langle n_{\widetilde{\mathcal{I}}}(\widetilde{\mathcal{I}}^{-1}g_0)\rangle_2 \cdot \nu(g_0,g_1,g_2). \tag{172}$$

It is easy to check that

$$\phi^{2,1}_\tau + \phi^{2,1}_{\widetilde{\mathcal{I}}\circ\tau} = \nu. \tag{173}$$

Namely, fractional spin at $\mu$ can be trivialized by the 2D SPT phase described above.

## C.3 3D SPT-LSM systems with magnetic inversion

Let us consider the example in Section 4.4. We start with a 3D lattice system, where the global symmetry group of this system is the same as the last part, which reads $Z_4^{\widetilde{\mathcal{I}}} \times Z_2^{\mathcal{T}}$. Cells around the inversion center is shown in Fig. 6, where we can read off the boundary mappings of these cells:

$$\begin{aligned}
\langle\mu|\partial &= -\langle\tau| - \langle\bar{\tau}|; \\
\langle\tau|\partial &= -\langle\bar{\tau}|\partial = \langle\sigma| - \langle\bar{\sigma}|; \\
\langle\sigma|\partial &= \langle\bar{\sigma}|\partial = \langle\rho| + \langle\bar{\rho}|; \\
\langle\rho|\partial &= \langle\bar{\rho}|\partial = 0.
\end{aligned} \tag{174}$$

We then put a Kramers doublet at inversion center, labeled by a 2-cocycle $\nu$ defined in Eq. (166). Following similar calculation in the last part, we conclude that to accommodate the Kramers doublet at inversion center, we at least require 2nd order SPT, which is constructed by decorating $\sigma$ and $\bar{\sigma}$ with a non-trivial 2D SPT protected by $Z_2 \times Z_2^{\mathcal{T}}$ defined in Eq. (167) and (168).

As shown in the main text, it is also possible to have a strong SPT phase to cancel the "SPT-LSM anomaly", which is constructed by decorating 3-cells with an SPT phase protected by $Z_2 \times Z_2^{\mathcal{T}}$ symmetry. To identify the decoration, we first calculate the group cohomology using Künneth formula as

$$\begin{aligned}
H^4\left[Z_2 \times Z_2^{\mathcal{T}}, U(1)_{\mathcal{T}}\right] &= H^4\left[Z_2^{\mathcal{T}}, U(1)_{\mathcal{T}}\right] \times H^3\left[Z_2^{\mathcal{T}}, H^1\left[Z_2, U(1)_{\mathcal{T}}\right]\right] \times H^1\left[Z_2^{\mathcal{T}}, H^3\left[Z_2, U(1)_{\mathcal{T}}\right]\right] \\
&= Z_2^3.
\end{aligned} \tag{175}$$

We claim that the 3-cell-decoration is characterized by the generator of $H^3\left[Z_2^{\mathcal{T}}, H^1\left[Z_2, U(1)_{\mathcal{T}}\right]\right]$ $= Z_2$.

To see this, we first express any $g \in Z_4^{\widetilde{\mathcal{I}}} \times Z_2^{\mathcal{T}}$ as $g = \widetilde{\mathcal{I}}^{n_{\widetilde{\mathcal{I}}}(g)}\mathcal{T}^{n_{\mathcal{T}}(g)}$, where $n_{\widetilde{\mathcal{I}}}(g) \in \{0,1,2,3\}$ and $n_{\mathcal{T}}(g) \in \{0,1\}$. We also define $n_s(g) = [n_{\widetilde{\mathcal{I}}}(g)/2] \in \{0,1\}$ for later use. We then decorate $\rho$ with cocycle $\phi_\rho$, which reads

$$\phi_\rho(g_0,g_1,g_2,g_3,g_4) = [n_s(g_0) - n_s(g_1)] \cdot \beta(g_1,g_2,g_3,g_4), \tag{176}$$

where $\beta(g_1,g_2,g_3,g_4) = \pi \cdot \prod_{i=1}^3 [n_{\mathcal{T}}(g_i) - n_{\mathcal{T}}(g_{i+1})]$. Hence, decoration on $\bar{\rho}$ can be obtained by action of $\widetilde{\mathcal{I}}$:

$$\begin{aligned}
\phi_{\bar{\rho}}(g_0,g_1,g_2,g_3,g_4) &= \phi_\rho(g_0,g_1,g_2,g_3,g_4) \\
&= \left[n_s\left(\widetilde{\mathcal{I}}^{-1}g_0\right) - n_s\left(\widetilde{\mathcal{I}}^{-1}g_1\right)\right] \cdot \beta(g_1,g_2,g_3,g_4).
\end{aligned} \tag{177}$$

Decoration on 2-cell $\sigma$ satisfies equation $\mathrm{d}\phi_\sigma = \phi_{\sigma\partial} = \phi_\rho + \phi_{\bar{\rho}}$. One solution reads

$$\phi_\sigma(g_0,g_1,g_2,g_3) = \langle n_{\widetilde{\mathcal{I}}}(g_0)\rangle_2 \cdot \beta(g_0,g_1,g_2,g_3). \tag{178}$$

And decoration on $\bar{\sigma}$ can be generated by $\widetilde{\mathcal{I}}$ as

$$\phi_{\bar{\sigma}}(g_0, g_1, g_2, g_3) = \langle n_{\widetilde{\mathcal{I}}}(\widetilde{\mathcal{I}}^{-1} g_0)\rangle_2 \cdot \beta(g_0, g_1, g_2, g_3). \tag{179}$$

Decorations on 1-cells $\tau$ is obtained by solving equation $\mathrm{d}\phi_\tau = \phi_{\tau\partial} = \phi_\sigma - \phi_{\bar{\sigma}}$, and one solution for $\phi_\tau$ reads

$$\phi_\tau(g_0, g_1, g_2) = \begin{cases} \pi/2 & \text{if } n_{\mathcal{T}}(g_0) = n_{\mathcal{T}}(g_2) = 0, \; n_{\mathcal{T}}(g_1) = 1 \\ -\pi/2 & \text{if } n_{\mathcal{T}}(g_0) = n_{\mathcal{T}}(g_2) = 1, \; n_{\mathcal{T}}(g_1) = 0 \; . \\ 0 & \text{otherwise} \end{cases} \tag{180}$$

We then obtain $\phi_{\bar{\tau}} = \widetilde{\mathcal{I}} \circ \phi_\tau = \phi_\tau$.

Finally, on site $\mu$, we have

$$\phi_{\mu\partial} = -\phi_\tau - \phi_{\bar{\tau}} = -2\phi_\tau = \nu, \tag{181}$$

where $\nu$ is defined in Eq. (166), which is a non-trivial cocycle labelling Kramers doublet.

To summarize, 3D spin system with global symmetry $Z_4^{\widetilde{\mathcal{I}}} \times Z_2^{\mathcal{T}}$ and Kramers doublet at inversion center has SPT-LSM anomaly, which can be trivialized either by second order SPT phase obtained by decorating an inversion symmetric plane, or by a 3D strong SPT phase.

# D SPT phases from condensation of fractional quasi-particles

In this appendix, we show that many bosonic SPT phases can be obtained by condensing fractional quasi-particle excitations. Examples of fractional quasiparticles include domain walls in 1+1D, vortex or anyon in 2+1D and dyons of compact $U(1)$ gauge field in 3+1D.

This condensation mechanism provides us hint to realize bosonic SPT phases using interacting spin models. Although all cohomological bosonic SPT phases are known to be ground states of some exact solvable models, these models usually involve interactions between many ($\sim 10$) spins, which makes them too complicated to realize. The condensation picture gives us an alternative way to realize SPT phases, and possible by simpler spin models. Furthermore, the condensation picture also helps to search "symmetry enforced" SPT phases in various dimensions, as we shown in Section 4.

In principle, given the symmetric gauge theory and a specific "condensation pattern", there should be a "formula" to calculate the resulting SPT index. However, for the purpose of this paper, we will not try to find the general answers, instead, we focus on examples in various dimensions, and leave the general framework in the future work.

Many results presented here are known, and have appeared in many literatures [11,25,32, 33,64–72]. These papers use different languages, such as decorated domain wall, condensing bound states of vortex and charge, anyon condensation, etc. We feel it is convenient to have a unified language and try to understand these mechanisms in one framework, where we present in this appendix. Besides, the condensation mechanism to obtain 3+1D SPT phases by monopole condensation may be new to readers.

Let us first describe the general idea of condensation of fractional quasiparticles (especially bosonic gauge charges here). We start from a symmetric gauge theory with Abelian gauge group $GG$ and global symmetry group $SG$. Since the gauge charges are nonlocal objects, they transform under group $PSG$, which is an extension of $SG$ by $GG$ [58], and is classified by the second cohomology group $H^2[SG, GG]$. Condensing gauge charges will Higgs gauge field, leading to symmetric short-range entangled (SRE) phases or spontaneously symmetry breaking phases. If $PSG$ is a non-trivial extension, we always get symmetry breaking phase [73]. So,

to obtain SPT phases, we require $PSG$ of gauge charges to be a trivial extension. In the trivial $PSG$ case, gauge charges are labeled by different linear representations of $SG$, and we can choose the condensed gauge charge to carry different certain representations $\mathcal{R}$. In order to obtain a fully symmetric phase, any condensed local operator should transform trivially on $SG$, which puts constraint on $\mathcal{R}$. Roughly speaking, fusion of $\mathcal{R}$ should be consistent with fusion of gauge charges.

We point out that symmetry properties of gauge charges do not fully characterize the symmetric gauge theory. Given $PSG$ of gauge charges, there may exist multiple symmetric gauge theories, which are differ by symmetry properties of gauge fluxons/monopoles. Here, gauge fluxons/monopoles may or may not be a local object. For example, for $U(1)$ gauge theory in 2+1D, monopole is identified as instanton, and thus is a local object. Instead, for $U(1)$ gauge theory in 3+1D, the monopoles are nonlocal excitations, and can be viewed as gauge charges of the dual magnetic $\widetilde{U}(1)$ gauge field. And symmetry properties of local fluxons/monopoles are characterized by its quantum number (representation) $\widetilde{\mathcal{R}}$, while for nonlocal excitations, symmetry properties are characterized by $\widetilde{PSG}$, which is $SG$ extension of the dual gauge group $\widetilde{GG}$. We mention that, in 3+1D, "fluxons" of $Z_N$ gauge theory are loops, which makes their symmetry properties more complicated. We will not consider loop excitations here, and leave them for the future exploration.

It turns out that by condensing gauge charges with trivial linear representation $\mathcal{R}$, one would always obtain a trivial SPT phase, regardless of symmetry properties of fluxons/monopoles. Besides, when symmetry properties of fluxons/monopoles are trivial[4], the symmetric gauge charge condensed phase will also be trivial SPT. Yet, when fluxons/monopoles transform non-trivially under the global symmetry, by condensing gauge charges carrying non-trivial linear representation $\mathcal{R}$, it is possible to obtain non-trivial SPT phases.

There are at least two ways to identify the SPT index: either by studying edge properties on an open boundary sample, or by studying properties of "defects" of $SG$. In this part, we try to identify the SPT phases by studying properties of symmetry defects. Let us discuss examples in various dimensions in the following.

### D.1 The simplest example: decorated domain walls in 1+1D

Consider system with $Z_2^a \times Z_2^b$ symmetry, it is well known that there is an AKLT-like SPT phase, characterized by edge modes carrying projective representation of $Z_2^a \times Z_2^b$. As shown in Ref. [32], starting from the ordered phase of $Z_2^a$, this SPT phase can be obtained by condensing the bound state of $Z_2^a$ domain wall and $Z_2^b$ charge.

Let us rephrase this process using the gauge charge condensation language. We first point out that $Z_2^a$ domain walls are identified as $\widetilde{Z_2^a}$ gauge charges, while $Z_2^a$ charges are identified as $\widetilde{Z_2^a}$ (spacetime) fluxons (or instanton). When $\widetilde{Z_2^a}$ charges are gapped, $\widetilde{Z_2^a}$ fluxon would proliferate, leading spontaneous symmetry breaking phase of $Z_2^a$. To obtain a symmetric phase of $Z_2^a$, we condense bound states of $\widetilde{Z_2^a}$ gauge charges and $Z_2^b$ charges. Notice that any condensed local operator contains even number of this bounded operators, and is uncharged under $Z_2^b$. So, this condensation would preserves $Z_2^b$ symmetry, and in the mean time, it would kill $Z_2^a$ order parameter. Furthermore, nonzero long-range correlators of this bounded operator is just the familiar string operator in the non-trivial SPT phase [74].

This decorated domain wall picture can easily be generalized to global symmetry group $Z_n \times Z_m$ [37], and we applied this method for 1D SPT-LSM system in Section 2.3.

---

[4]For local excitations, trivial symmetry property means trivial representation, while for non-local excitations, it means the corresponding $PSG$ is a trivial extension.

## D.2 Vortex condensation in 2+1D

In 2+1D system with charge conservation symmetry $U_c(1)$, the vortex excitation are identified as $U_g(1)$ gauge charge by duality mapping. Under this mapping, charged bosons are identified as magnetic monopole of $U_g(1)$ gauge theory.

Bound states of a single vortex and $n$ bosons have bosonic/fermionic statistics if $n$ is an even/odd integer. By condensing the bound state of one vortex and $2m$ bosons, we get bosonic integer quantum Hall phase with $\sigma_{xy} = 2m$ [11].

If the system also hosts time reversal symmetry, we are able to get bosonic quantum spin Hall phases by condensing vortices that are odd under time reversal [69].

## D.3 Anyon condensation in 2+1D

Now, we present the approach to get 2+1D SPT phases from symmetric anyon condensation. A complete survey based on tensor network construction can be found in Ref. [33].

Let us first consider example where global symmetry $SG = Z_2^s \times Z_2^{\mathcal{T}} = \{1, s\} \times \{1, \mathcal{T}\}$. We start from a symmetric $Z_2^g$ gauge theory (toric code), where gauge flux $m$ transforms as a Kramers doublet under $Z_2^{\mathcal{T}}$, while gauge charge $e$ transforms linearly under $SG$. To obtain a non-trivial SPT phase, we choose to condense the bound state of $e$ and $Z_2^s$ symmetry charge $\mathcal{R}_s$. To see the nature of this condensed phase, let us study properties of symmetry defect of $Z_2^s$. We first gauge symmetry $Z_2^s$, and label $s$ as $Z_2^s$ flux. After condensing the bound state of $e$ and $s$, both $Z_2^g$ gauge flux $m$ and $Z_2^s$ flux $s$ are confined, due to their non-trivial braiding statistics with the condensed particle. However, the bound state of $m$ and $s$ is a deconfined quasi-particle in the condensed phase, which is a Kramers doublet under $Z_2^{\mathcal{T}}$. Thus, in the ungauged theory, where $Z_2^s$ is treated as global symmetry, and $Z_2^s$ fluxons, which are identified as ends of $Z_2^s$ domain walls, carry Kramers doublets. In other words, $Z_2^s$ domain walls in this symmetric phase is decorated with $Z_2^T$ Haldane phase, which is the signature for a non-trivial SPT [32].

The above anyon condensation construction can be easily generalized to symmetry group $SG$, where $Z_N^s$ is a normal subgroup of $SG$. In this case, we start from $Z_N^g$ gauge theory, and gauge flux $m$ carries non-trivial projective representation of $SG$ with coefficient in $Z_N^s$. The non-trivial SPT can be obtained by condensing the bound state of gauge charge $e$ and symmetry charge of $Z_N^s$.

This construction can also capture SPT phases beyond decorated domain wall picture. For example, let us consider $SG = Z_2^s = \{1, s\}$. We start with symmetric $Z_2^g$ gauge theory, where $m$ carry half $Z_2^s$ charge ($s^2 \circ m = -m$), and $e$ carry integer $Z_2^s$ charge. By condensing bound state of $e$ and $Z_2^s$ charge, we obtain a symmetric phase, which we claim to be the famous Levin-Gu $Z_2^s$ SPT phase [54]. To see this, let us study properties of the $Z_2^s$ defect. Following similar argument above, we conclude that the "deconfined" $Z_2^s$ defect carries half $Z_2^s$ charge, which hosts topological spin $\pm i$, and is identified as semion. As shown in Ref. [54], this is the hallmark of the non-trivial SPT phase.

## D.4 Monopole condensation in 3+1D

Now, let us turn to 3+1D SPT phases. It has been explored in the past literatures to obtain bosonic SPT phases by so called dyon-condensation mechanism [68, 70, 75–77].

Here, we present an overview of SPT phases protected by $U_s(1) \times Z_2^{\mathcal{T}}$ from monopole condensation. $U_s(1)$ can be understood as spin rotation symmetry along $z$-axis, and time reversal action $\mathcal{T}$ flips spins. Cohomology group calculation gives SPT classification as [9]

$$H^4[U_s(1) \times Z_2^{\mathcal{T}}, U(1)_{\mathcal{T}}] = Z_2^3,. \tag{182}$$

Two of these three $Z_2$ root phases are due to interplay between $U_s(1)$ and $Z_2^{\mathcal{T}}$, while the third one denotes the SPT phase protected by time reversal only.

The first two $Z_2$ root phases are characterized by Witten effect [70, 78]. One introduces external compact $U(1)$ gauge field $A_s$ coupled to $U_s(1)$ charge, and studies properties of magnetic monopoles of $A_s$. Monopoles in the first $Z_2$ root SPT phase are Kramers doublets, while monopoles in the second $Z_2$ root SPT phase have fermionic statistics. In contrast, the third $Z_2$ root phase cannot detect by monopoles. The diagnostic for this phase is its anomalous surface state — the $e\mathcal{T}m\mathcal{T}$ surface topological order phase [56].

To obtain these three root phases, we start from a symmetric $U_g(1)$ gauge theory, where gauge field is labeled as $a_g$. Excitations of the deconfined phase of this gauge theory include electric charge $b_g$, magnetic monopole $M_g$, as well as gapless photon. In the condensed matter context, this phase is also named as $U_g(1)$ quantum spin liquid. And formally, the $U_g(1)$ gauge field emerges in local boson/spin systems by parton construction [79] or gauge mean field theory [80]. Here, we consider the case where both $b_g$ and $M_g$ are *bosonic* excitation.

For a given symmetry group, there are many $U_g(1)$ spin liquids, which are differed by symmetry properties of their excitations. First, we require that under the action of $\mathcal{T}$, electric field is invariant, while magnetic field changes sign. In other words, $\mathcal{T}$ reverses the magnetic charge while leaves electric charge invariant.

$b_g$ and $M_g$ are non-local objects, and in general, they carry projective representation with coefficient in $U(1)$. Here, we consider the case where $b_g$ carries non-trivial projective representation under $U_s(1) \times Z_2^{\mathcal{T}}$, while $M_g$ transform linearly under this symmetry group. Projective representation of $U_s(1) \times Z_2^{\mathcal{T}}$ is classified by the second cohomological group

$$H^2[U_s(1) \times Z_2^{\mathcal{T}}, U_{\mathcal{T}}(1)] = Z_2^2 . \tag{183}$$

The physical meaning of these two $Z_2$'s are clear. Roughly speaking, the first $Z_2$ generator indicates that $b_g$ is a Kramers doublet, while the second $Z_2$ generator means that $b_g$ carries half-integer $U_s(1)$ charge.

The magnetic monopole $M_g$ is coupled to dual gauge field $\widetilde{U_g}(1)$. Symmetry action on $M_g$ is set to be

$$\begin{aligned}
\widetilde{U_g}(\phi) &: M_g \to e^{i\phi} \cdot M_g \\
U_s(\theta) &: M_g \to M_g \\
\mathcal{T} &: M_g \to M_g^\dagger ,
\end{aligned} \tag{184}$$

where $\widetilde{U_g}(\phi)$ belongs to the dual gauge field $\widetilde{U_g}(1)$.

In the following, we will start from these symmetry enriched $U_g(1)$ quantum spin liquid with $b_g$ carry projective representation. Then, by condensing bound state of $M_g$ and certain symmetry charge, we are able to obtain SPT phases protected by $U_s(1) \times Z_2^{\mathcal{T}}$.

### D.4.1 Trivial SPT phase

Starting from any quantum spin liquids, by condensing trivial monopole $M_g$ without attaching any symmetry charge, one always obtains a trivial SPT phase.

### D.4.2 SPT phase with Kramers-doublet $U_s(1)$ monopoles

We start from spin liquid phase with $b_g$ transform as Kramers doublets under $\mathcal{T}$ and carry integer $U_s(1)$ charge. To realize it, we assign $b_g$ with spin index, labeled as $b_g \equiv (b_{g\uparrow}, b_{g\downarrow})^{\mathrm{t}}$.

The simplest symmetry assignment on $b_g$ reads

$$
\begin{aligned}
U_g(\phi) &: b_g \to \mathrm{e}^{\mathrm{i}\phi} \cdot b_g, \\
U_s(\theta) &: b_g \to b_g, \\
\mathcal{T} &: b_g \to \mathrm{i}\sigma^y \cdot b_g, \quad \mathrm{i} \to -\mathrm{i}.
\end{aligned}
\tag{185}
$$

Namely, $b_g$ carry no $U_s(1)$ charge and $\mathcal{T}^2 \circ b_g = -b_g$.

We then add another layer formed by local bosons, labeled as $B_s$, which carries unit $U_s(1)$ charge. Under $\mathcal{T}$, $B_s$ transform to $B_s^\dagger$.

By condensing the bound state of $M_g$ and $B_s$, we Higgs out $U_g(1)$ gauge group. Both $U_s(1)$ and $\mathcal{T}$ are preserved in this Higgs phase, since all condensed local operators carry no global symmetry charge.

To determine which SPT phase it belongs to, we study properties of external $U_s(1)$ monopole by coupling $U_s(1)$ charge to external gauge field $A_s$. Due to condensation of $B_s$ and $M_g$ bound states, the bare $U_s(1)$ monopole, which picks nonzero Berry phase when winding around $B_s$, is confined. The deconfined $U_s(1)$ monopole is identified as bound state of $U_s(1)$ monopole and $b_g$, which transforms as Kramers doublets under $\mathcal{T}$. This is a signature of a non-trivial bosonic SPT phase [56, 81, 82].

### D.4.3 SPT phase with fermionic $U_s(1)$ monopoles

Similar as the previous case, we start with two-component gauge charge: $b_g \equiv (b_{g\uparrow}, b_{g\downarrow})^{\mathrm{t}}$. And we require $b_g$ to carry half charge under $U_s(1)$, and transforms as a Kramers singlet under $\mathcal{T}$. Then a natural symmetry assignment on $b_g$ reads

$$
\begin{aligned}
U_g(\phi) &: b_{g\sigma} \to \mathrm{e}^{\mathrm{i}\phi} \cdot b_{g\sigma}, \\
U_s(\theta) &: b_{g\uparrow} \to \mathrm{e}^{\mathrm{i}\theta/2} \cdot b_{g\uparrow}, \qquad b_{g\downarrow} \to \mathrm{e}^{-\mathrm{i}\theta/2} \cdot b_{g\downarrow}, \\
\mathcal{T} &: b_{g\uparrow} \leftrightarrow b_{g\downarrow}, \qquad\qquad \mathrm{i} \to -\mathrm{i}.
\end{aligned}
\tag{186}
$$

Here, $b_g$ can be viewed as parton decomposition for the physical spins $\vec{S}$:

$$
\vec{S} \sim \frac{1}{2} b_g^\dagger \cdot \vec{\sigma} \cdot b_g,
\tag{187}
$$

where $b_g$'s are glued by gauge field $a_g$ to recover the physical Hilbert space.

On spin operator, $U_s(1)$ is identified as spin rotation symmetry along $z$-axis while $\mathcal{T}$ flips spin:

$$
\begin{aligned}
U_s(\theta) &: S^+ \to \mathrm{e}^{-\mathrm{i}\theta} S^+, \quad S^z \to S^z, \\
\mathcal{T} &: S^\pm \to S^\mp, \qquad S^z \to -S^z, \quad \mathrm{i} \to -\mathrm{i}.
\end{aligned}
\tag{188}
$$

We claim that the Higgs phase obtained by condensing the bound state of $M_g$ and $S^+$ would be a non-trivial SPT phase, which is characterized by "statistical Witten effect", where $U_s(1)$ monopoles have fermionic statistics [78].

To see this, let us couple $U_s(1)$ charge to an external compact gauge field $A_s$. Then, $S^+$ is identified as gauge charge for $A_s$. And due to the compactness of $A_s$, there are also monopole excitations for $A_s$.

Let us study the deconfined phase for $U_g(1)$ gauge theory first. Topological excitations, or dyons, for this deconfined phase are labeled by a four component vector $\vec{n} = (q_g, m_g; q_s, m_s)$, where $q_g$ ($q_s$) counts charge number for $a_g$ ($A_s$) gauge field, and $m_g$ ($m_s$) counts monopole number of $a_g$ ($A_s$) gauge field.

Under $\mathcal{T}$ action, gauge charge/monopole number changes as

$$\mathcal{T} : (q_g, m_g; q_s, m_s) \rightarrow (q_g, -m_g; -q_s, m_s). \tag{189}$$

The next step is to figure out quantization conditions for these dyons by studying the accumulated Berry phase when winding them around each other [83,84]. Given two arbitrary dyons labeled by $\vec{n}_1$ and $\vec{n}_2$, where $\vec{n}_i = (q_g^i, m_g^i; q_s^i, m_s^i)$, we put $\vec{n}_1$ dyon in the origin and move $\vec{n}_2$ dyon along a closed path encircling an solid angle $\Omega$ respective to the origin. Then, the Berry phase accumulated in this process is given by

$$\exp\left[\frac{i\Omega}{2}\left(q_g^2 m_g^1 - q_g^1 m_g^2 + q_s^2 m_s^1 - q_s^1 m_s^2\right)\right]. \tag{190}$$

The quantization condition comes from the fact that the above Berry phase should be invariant for the chose of $\Omega$ or $4\pi - \Omega$. So, we get the quantization condition as

$$q_g^2 m_g^1 - q_g^1 m_g^2 + q_s^2 m_s^1 - q_s^1 m_s^2 \in \mathbb{Z}. \tag{191}$$

In the case considered here, according to Eq. (186) and Eq. (188), dyonic charge for various operators is determined as

$$b_{g\uparrow} \sim (1, 0; \frac{1}{2}, 0), \quad b_{g\downarrow} \sim (1, 0; -\frac{1}{2}, 0),$$
$$M_g \sim (0, 1; 0, 0), \quad S^+ \sim (0, 0; -1, 0). \tag{192}$$

In particular, since $b_{g\sigma}$ carries half $U_s(1)$ charge, the "bare" $U_s(1)$ monopole with dyonic charge $(0, 0; 0, 1)$ would be disallowed by quantization condition Eq. (191). Instead, monopole operator

$$M_{s+} \sim (0, \frac{1}{2}; 0, 1), \quad M_{s-} \sim (0, -\frac{1}{2}; 0, 1) \tag{193}$$

are deconfined excitations. Besides, we emphasis that the quasiparticles mentioned above are all *bosonic*.

In fact, it is convenient to use $b_{g\uparrow/\downarrow}$ and $M_{s\pm}$ as basis of the four dimensional "charge-monopole lattice" for this $U_s(1) \times U_g(1)$ gauge theory. In particular, we can express $M_g$ and $S^+$ using these four basis as

$$M_g \sim M_{s+} M_{s-}^\dagger, \quad S^+ \sim b_{g\uparrow}^\dagger b_{g\downarrow}. \tag{194}$$

These four vectors can be grouped into two sets

$$\left\{ b_{g\uparrow}, M_{s+} \right\}, \quad \left\{ b_{g\downarrow}, M_{s-} \right\}. \tag{195}$$

According to Eq. (190), quasiparticles belonging to different groups are invisible to each other, while quasiparticles within one set "statistically interact" as charge and monopole.

Now, to kill gauge field $a_g$, let us condense the bound state of $M_g$ and $S^+$, which is labeled as $D \sim (0, 1; -1, 0)$. The local operator $D^\dagger D$ carries no symmetry charge, so the condensed phase should belong to some symmetric SRE phase. To see the SPT index of this symmetric SRE phase, let us study properties of monopoles of external gauge field $A_s$.

After condensation, the "deconfined" external monopole should pick trivial Berry phase when encircling around $M_g S^+$. According to Eq. (190), the simplest deconfined excitations carrying unit external monopole number are

$$M_{s+} b_{g\uparrow}^\dagger \sim (-1, \frac{1}{2}; -\frac{1}{2}, 1), \qquad M_{s-} b_{g\downarrow}^\dagger \sim (-1, -\frac{1}{2}; \frac{1}{2}, 1);$$
$$M_{s+} b_{g\downarrow}^\dagger \sim (-1, \frac{1}{2}; \frac{1}{2}, 1), \qquad M_{s-} b_{g\downarrow}^\dagger \sim (-1, -\frac{1}{2}; -\frac{1}{2}, 1). \tag{196}$$

Notice that the first two operators are only differ by the condensed object $(0, 1; -1, 0)$, and thus can be identified as the same excitation in the Higgs phase, labeled as $M_f$. Importantly, $M_f$ has fermionic statistics, as $M_{s+}$ ($M_{s-}$) and $b_{g\uparrow}^{\dagger}$ ($b_{g\downarrow}^{\dagger}$) statistically interact as charge and monopole. On the contrary, the last two excitations in Eq. (196), labeled as $M_{b+}$ and $M_{b-}$, are bosonic excitations. Under $\mathcal{T}$ action, $M_f$ is invariant, while $M_{b\pm}$ transforms to each other.

We then count $U_s(1)$ charge of $M_f$ and $M_{b\pm}$. Naively, one may think $M_f$, $M_{b\pm}$ are charge-$1/2$ excitations. However, it is no longer valid to identify $q_s$ as $U_s$ charge number after condensation In fact, in the condensed phase, the $U_s(1)$ charge carried by the excitation labeled as $(q_g, m_g; q_s, m_s)$ is

$$Q_s = q_s + m_g. \tag{197}$$

The reason is due to screening effect. In the condensed phase, $D \sim M_g S^+$ form a Debye-plasma with short-range interaction. Quasi-particle $(q_g, m_g; q_s, m_s)$ would be Debye screened by $D$'s: it will be surrounded by a cloud of $D$'s and $D^{\dagger}$'s with total $D$ number equals to $-m_g$. Since $D$ carry $U_s(1)$ charge $-1$, the screening cloud carry $U_s(1)$ charge $m_g$. And the total $U_s(1)$ charge $Q_s$ of the quasi-particle $(q_g, m_g; q_s, m_s)$ in the condensed phase is $q_s + m_g$. Then, it is straightforward to see that fermionic $A_s$ monopole $M_f$ carries no $U_s(1)$ charge, while bosonic $A_s$ monopole $M_{b\pm}$ carries plus/minus unit $U_s(1)$ charge. This phenomena is named as statistical Witten effect in Ref. [78] and is proved to be the feature of a non-trivial bosonic SPT phase.

One can also figure out statistical Witten effect by studying $\mathcal{T}$ action on monopoles. It is easy to see that

$$\mathcal{T} : M_f \to M_f, \quad M_{b+} \leftrightarrow M_{b-}, \quad S^+ \leftrightarrow S^-. \tag{198}$$

Since $U_s(1)$ electric charge changes sign under $\mathcal{T}$, $U_s(1)$ magnetic charge should be invariant under $\mathcal{T}$. Thus, $M_f$ is identified as monopole which carries unit magnetic charge, while $M_{b\pm}$ are dyons carrying both electric and magnetic charge. We then conclude the "pure" monopole $M_f$ is a fermion.

For later use, let us also discuss the surface state for this bosonic SPT phase. It has been shown that there exists a gapped symmetric surface state with toric code topological order for this phase, which is named as $e\mathcal{C}m\mathcal{C}$ [56, 78]. As suggested by the name, $e$ and $m$ both carry half-charge of $U_s(1)$. We can treat $e$ ($m$) to be a two component operator, with $e = (e_1, e_2)^{\mathrm{t}}$ and $m = (m_1, m_2)^{\mathrm{t}}$. Under global symmetry, $e$ and $m$ transforms as

$$U_s(1) : e \to \exp\left[\mathrm{i}\frac{\sigma^z}{2}\theta\right] \cdot e, \quad m \to \exp\left[\mathrm{i}\frac{\sigma^z}{2}\theta\right] \cdot m;$$
$$\mathcal{T} : e \to \sigma^x \cdot e, \qquad m \to \sigma^x \cdot m, \qquad \mathrm{i} \to -\mathrm{i}. \tag{199}$$

In a purely 2+1D bosonic system, $e\mathcal{C}m\mathcal{C}$ state can never preserve time reversal symmetry: it must supports nonzero Hall conductance in 2+1D. In this sense, the surface symmetric $e\mathcal{C}m\mathcal{C}$ topological order is anomalous.

### D.4.4 SPT phase with $e\mathcal{T}m\mathcal{T}$ surface state

Now, let us turn to the third root SPT phase of the cohomology group $H^4[U_s(1) \times Z_2^{\mathcal{T}}, U(1)]$. Unlike the previous two cases, this phase cannot be captured by bulk Witten effect: the external $U_s(1)$ monopole has the same properties as that in the trivial SPT phase. In fact, this root SPT phase is only protected by $\mathcal{T}$: even if one explicitly breaks $U_s(1)$ symmetry, we still obtain a non-trivial SPT phase.

It is argued that the surface state of this SPT phase can support an anomalous symmetry enriched toric code phase, where $e$ and $m$ are both Kramers doublet under time reversal symmetry [56, 57].

Can we still obtain this SPT phase by monopole condensation of some $U_g(1)$ quantum spin liquid? The answer is yes, and we will describe the condensation process in the following.

We could safely break $U_s(1)$ symmetry, and focus on $\mathcal{T}$ symmetry, as $U_s(1)$ symmetry plays little role in the SPT discussed here. Let us start from a $U_g(1)$ gauge theory, with $\mathcal{T}$ action defined as

$$\mathcal{T}: b_g \to \sigma^y\, b_g\,, \quad M_g \to M_g^\dagger\,, \quad \mathrm{i} \to -\mathrm{i}\,. \tag{200}$$

Here, under $\mathcal{T}$ action, $b_g = (b_{g\uparrow}, b_{g\downarrow})^{\mathrm{t}}$ transforms as a Kramers doublet, while $M_g$ is mapped to its anti-particle.

As before, one can view $b_g$ as partons for the physical spin $\vec{S}$, with $\vec{S} \sim \frac{1}{2} b_g^\dagger \cdot \vec{\sigma} \cdot b_g$. Then, physical time reversal symmetry is defined as

$$\mathcal{T}: \vec{S} \to -\vec{S}\,, \quad \mathrm{i} \to -\mathrm{i}\,. \tag{201}$$

We claim that by condensing the bound state of $M_g$ and $S^+$, the final phase would be this non-trivial SPT phase characterized by its anomalous surface state, e.g. $e\mathcal{T}m\mathcal{T}$.

To see this, let us start from the whole symmetry group $U_s(1) \times Z_2^{\mathcal{T}}$, and study $U_g(1)$ QSL with symmetry transformation rules for charge $b_g$ and monopole $M_g$ defined in Eq. (186) and Eq. (188). Let us define a new time reversal symmetry operator $\widetilde{\mathcal{T}}$ as

$$\widetilde{\mathcal{T}} \equiv U_s(\pi)\mathcal{T}\,. \tag{202}$$

According to Eq. (188) and Eq. (202), under $\widetilde{\mathcal{T}}$ action, the physical spin operator $\vec{S}$ transforms the same as that in Eq. (201). Furthermore, from Eq. (186) and Eq. (184), we are able to read out $\widetilde{\mathcal{T}}$ action on $b_g$ and $M_g$. It turns out that , $\widetilde{\mathcal{T}}$ acts the same on $b_g$ and $M_g$ as $\mathcal{T}$ in Eq. (200). In particular, $b_g$ is a Kramers singlet under the original $\mathcal{T}$ symmetry, but becomes a Kramers doublet under $\widetilde{\mathcal{T}}$ action.

As in the last part, let us condense the bound state of $M_g$ and $S^+$. We notice that under $\widetilde{\mathcal{T}}$, $M_g S^+ \to -M_g^\dagger S^-$. Here, this minus sign is physical and cannot be tuned away by magnetic $\widetilde{U_g}(1)$ gauge transformation. We call this composite object as the $\widetilde{\mathcal{T}}$-odd monopole. In the following, we will prove that this condensation pattern would give the non-trivial SPT phase with $e\mathcal{T}m\mathcal{T}$ anomalous surface state.

As shown in the last part, the condensed phase is a non-trivial SPT phase protected by symmetry $U_s(1) \times Z_2^{\mathcal{T}}$, with fermionic $U_s(1)$ monopole. Moreover, there exists a symmetric gapped surface state, dubbed "$e\mathcal{C}m\mathcal{C}$", where symmetry transformation rules are defined in Eq. (199). Then, under $\widetilde{\mathcal{T}} = U_s(\pi)\mathcal{T}$ action, $e$ and $m$ transform as

$$\widetilde{\mathcal{T}}: e \to \sigma^y \cdot e\,, \quad m \to \sigma^y \cdot m\,, \quad \mathrm{i} \to -\mathrm{i}\,, \tag{203}$$

$e$ and $m$ are both Kramers doublet under $\widetilde{\mathcal{T}}$ action, and the surface state is named as $e\mathcal{C}\widetilde{\mathcal{T}}m\mathcal{C}\widetilde{\mathcal{T}}$ [57].

Now, let us break $U_s(1)$ and $\mathcal{T}$ symmetry by hand but preserve the combination $\widetilde{\mathcal{T}}$, and perform the above condensation procedure again. We then start with a $U_g(1)$ QSL with $b_g$ being Kramers doublet under $\widetilde{\mathcal{T}}$. By condensing the $\widetilde{\mathcal{T}}$-odd monopole $M_g S^+$, we expect to get a $\widetilde{\mathcal{T}}$ SPT phase, which is characterized by the $e\widetilde{\mathcal{T}}m\widetilde{\mathcal{T}}$ surface topological order.

Lastly, we comment that there is actually additional $Z_2$ root SPT phase beyond cohomology classification, which has symmetric $efmf$ surface topological order. For the purpose of our paper, we will not discuss the possible monopole condensation mechanism for this phase here.

# E  Examples of LSM systems

In this appendix, we discuss some non-trivial LSM systems. While gauge charge condensation is a powerful tool to construct SPT-LSM system from the parent LSM system, one should be very careful about this construction. In this part, we show one example where this construction fails.

Let us consider a square lattice system with internal $Z_2^g \times Z_2^h$ symmetry as well as magnetic translation symmetry satisfying $T_x T_y = g T_y T_x$. Each site supports a qubit, with symmetry action identified as $g = \prod_j \sigma_j^x$ and $h = \prod_j \sigma_j^z$. Namely, $Z_2^g \times Z_2^h$ forms a projective representation on each site, characterized by the anti-commutator $\sigma^x \sigma^z = -\sigma^z \sigma^x$.

Naively, this system seems to be an SPT-LSM system following the anyon condensation argument. However, as discussed in Ref. [25], it actually holds conventional LSM anomaly. One way to see this LSM anomaly is to follow spectral sequence calculation discussed in Appendix C: by exhausting all kinds of SPT decoration on 1-cells and 2-cells, we find that none of these decorations lead to projective representation on 0-cells. Here, we will examine the anyon condensation mechanism more carefully, and pointing out why the argument does not work in this case.

The classification of $Z_2^g \times Z_2^h$ SPT phases is obtained by cohomological calculation as

$$
\begin{aligned}
H^3[Z_2^g \times Z_2^h, U(1)] &= H^3[Z_2^g, U(1)] \times H^3[Z_2^h, U(1)] \times H^2[Z_2^g, H^1[Z_2^h, U(1)]] \\
&= Z_2^3,
\end{aligned}
\tag{204}
$$

where the first two $Z_2$ root phases are Levin-Gu SPT protected $Z_2^g$ or $Z_2^h$ respectively, and the last root phase comes from the interplay between $Z_2^g$ and $Z_2^h$.

We start from a $Z_2$ spin liquid with $e$ transforming projective representation under $Z_2^g \times Z_2^h$: $gh \circ e = -hg \circ e$. Then, by condensing bound state of $m$ and $g$-charge, we obtain the third root SPT phase.

Now, let us add magnetic translation symmetry. Due to the background $e$ charge at each site, encircling $m$ around one site picks up $-1$ Berry phase. Notice that $g$-charge travelling around a unit cell is acted by $g$ symmetry due to the magnetic translation, which also picks $-1$ phase. So, bound state of $m$ and $g$-charge transform trivially under magnetic translation. Naively, by condensing this bound state, we obtain the SPT phase without breaking lattice symmetry.

However, the anyon condensation argument above has a fatal flaw: in fact, magnetic translation is incompatible with this particular symmetry fractionalization of $e$, and such $Z_2$ spin liquid can never be realized. To see this, we consider the following PSG equations

$$
\begin{aligned}
T_x T_y \circ e &= \eta_{xy} g T_y T_x \circ e, \\
h T_x T_y \circ e &= \eta_{h,xy} T_x T_y h \circ e,
\end{aligned}
\tag{205}
$$

where $\eta_{xy}, \eta_{h,xy}$ belong to $Z_2$ IGG, and take value $\pm 1$ when acting on $e$. Then, we have

$$
\begin{aligned}
h T_x T_y \circ e &= \eta_{xy} h g T_y T_x \circ e \\
&= -\eta_{xy} \eta_{h,xy} g T_y T_x h \circ e \\
&= -\eta_{h,xy} T_x T_y h \circ e,
\end{aligned}
\tag{206}
$$

where we use $gh \circ e = -hg \circ e$ to obtain the second line. We get inconsistency from the last lines in the above two equations. Thus, we conclude that $e$ carrying projective representation of $Z_2^g \times Z_2^h$ is incompatible with magnetic translation group.

Actually, we can use anyon condensation to show this SPT phase is realized in systems with linear representation per site. Let us start with $Z_2$ gauge theory, where $e$ carries half $g$-charge, which can be realized in system with linear representation. Unlike the previous case,

this symmetry fractionalization pattern is compatible with magnetic translation group. We then condensing bound state of $m$ and $h$-charge, which leads to the desired SPT phase without breaking lattice symmetry. Similarly, we can also construct other $Z_2^g \times Z_2^h$ SPT phases on systems with linear representations. In other words, none of SPT decorations can be supported in systems with projective representation and magnetic translation group. So, the system considered here must have LSM anomaly.

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
