# Peer review of "Generalized Lieb-Schultz-Mattis theorem on bosonic symmetry protected topological phases"

_SciPost Physics, doi:SciPost Phys. 11, 024 (2021)_

## Round 1 · Referee Report · Anonymous (Referee 1) · 2020-5-19

Strengths

  1. Provides a comprehensive understanding of so-called "LSM-SPT" phases
  2. Provides different viewpoints on the construction of these phases
  3. Includes a reasonably transparent 1D example worked out quite thoroughly.

Weaknesses

There are no obvious scientific weaknesses, but I think that the presentation could be improved somewhat. The paper is on the technical side which makes this challenging, but more figures (particularly when discussing the decoration procedures) would be helpful.

Report

This is a rather technical paper that nevertheless presents interesting and potentially important results.

The basic topic of the paper is an extension of the "Lieb Schulz Mattis" type constraints. These are constraints that require a quantum system to exhibit fractionalization/topological order/host long-range entanglement in its ground state given that it is gapped and preserves some combination (usually on-site intertwined with translational) symmetries. These can be viewed as "lattice anomaly matching" conditions on admissible ground states of local Hamiltonians equipped with some symmetry structure, and depends sensitively on the nature of the on-site Hilbert space (e.g. whether it has an odd or an even number of half-integer spins or electrons, or an integer or fractional number of bosons).

The authors extend these results to systems where a short-range entangled state is admissible (where the usual LSM constraints do not apply) but there are nevertheless obstructions to finding a fully trivial insulating state. The result is a ground state that frequently has protected gapless modes --- a symmetry-enforced property.

This has obviously interesting implications, as it gives some idea of what sort of systems/materials one might design in order to stablize e.g. a crystal-symmetry-protected topological phase of a quantum magnet. [The present paper is devoted mostly to toy models, but I mention this as an indication of the sort of application the work might have.]

Overall this seemed like an interesting, useful, and scientifically valuable contribution. I'd say it's considerable more formal than is the standard fare -- even by the relatively more mathematical standards of the subfield of topological phases -- but I think that it is reasonably well-explained so that a determined but non-expert reader would still grasp the essentials.

This would be greatly aided, however, if there were more figures -- for a paper with such a deep link to real-space structure to have only 3 figures in the main text is surprising and makes descriptions extremely unwieldy that could have been summarized graphically in a much cleaner fashion (and enhanced the comprehensibility to an audience not raised on a diet of cohomology groups).

I do think that the authors could consider linking to other perspectives given on similar problems, e.g. the lattice homotopy constraints of their cited Reference 21.

Requested changes

  1. The authors should consider adding more figures to explain their constructions.

  2. I found the appendices quite useful. It's probably a good idea for the authors to add more frequent references to the appendices where they will be useful in the text, so that the reader clearly knows what is new and what is a standard-but-possibly-unfamiliar result.

  3. There seems to be a missing reference in the opening paragraph of Section IV C 3 (the model is claimed to have been studied before but it is not clear where.)

  • validity: high
  • significance: good
  • originality: good
  • clarity: good
  • formatting: reasonable
  • grammar: acceptable

Author:  Shenghan Jiang  on 2021-05-17  [id 1427]

(in reply to Report 1 on 2020-05-19)
Category:
answer to question

We thank Referee 1 for positive comments on our work. We are so sorry for the late reply. We respond below to the specific suggestions.

  1. Thanks for the suggestions about adding more figures. We add two new figures in the main text. And we also move two figures from appendix to main text. We believe this makes our presentation better.

  2. Thanks for the suggestions. We add more references to the appendix in the main text.

  3. The model is studied in reference 25, although from another perspective. We add the missing reference in the main text.

---

## Round 1 · Referee Report · Anonymous (Referee 2) · 2020-5-22

Report

In the manuscript “Generalized Lieb-Schultz-Mattis theorem on bosonic symmetry protected topological phases, the authors provide an interesting family of generalized Lieb-Schultz-Mattis (LSM) constraints that forbid fully symmetric short-range-entangled (SRE) ground states to be “completely” trivial, in the sense that any SRE ground state in such systems has to be a symmetry-protected topological state (SPT). The authors illustrate an interesting mechanism to obtain such generalized versions of the LSM constraints via condensing topological excitations in fractionalized phases. Besides that, the authors provide a general mathematical framework to study such generalized LSM constraints and the resulting SPT states using the spectral sequence of cohomology theory, which is quite impressive. I believe the novelty of this paper matches the standard of SciPost.

Before the publication of this work, I hope the author can address the following quibbles/questions:
(1) In Sec. II A, there is a discussion (around Eq. (10)) about putting the system on the chain of 2n sites and showing that the resulting ground state is an SPT state. If I understand correctly, the reflection symmetry $\sigma$ requires a chain to have an odd number of sites. And the reflection symmetry $\sigma$ is crucial for the generalized LSM constraint to be applicable to the system, as is stated in Sec. II B. So the author may want to clarify the discussion a bit more, especially about the importance of the reflection symmetry $\sigma$ in the example in Sec. II A.
(2) In Sec IV A, the authors state that the featureless insulator on a $C_3$ symmetric honeycomb lattice with spin-1/2 per site and an odd number of sites is a second-order SPT state. It seems that any SRE featureless insulator defined on a system with an odd number of spin-1/2s will have to have some “unpaired” spin-1/2 on the boundary (because the bulk is gapped). Can I refer to all of them as second-order SPT states?
(3) In the fourth paragraph of Sec IV C, the authors make the statement how the $Z_4^{\tilde{\mathcal{I}}}$ and $Z_2^{\mathcal{T}}$ enforces any SRE state to be a SPT state with $\nu_{s \mathcal{T}}=1$. There is no mention of translation symmetries here. However, in the proof provided in Sec IV C 2, the translation symmetry seems to play a crucial role. Could the author clarify if the translation symmetry is needed?
(4) Eq. (60) is obtained via the Kunneth formula. The terms appear in the Kunneth formula, in the general context of standard bosonic SPTs, are often equipped with nice physical understandings (like domain wall decoration). Could the author explain a bit more how the term appearing in Eq. (60) is related to the condensation picture given above (60).
  • validity: high
  • significance: high
  • originality: top
  • clarity: -
  • formatting: -
  • grammar: -

Author:  Shenghan Jiang  on 2021-05-17  [id 1428]

(in reply to Report 2 on 2020-05-22)
Category:
answer to question

We appreciate the positive remarks by Referee 2, who believe the novelty of our paper matches the standard of SciPost. We are sorry for the long-period for replying. Here are our response to the specific comments and suggestions by Referee 2.

  1. We thank the referee to point out this. This SPT phase is defined on closed chains, and for a closed chain with even number of sites preserves reflection symmetry. However, for an open chain, as the referee pointed out, systems with even number of sites break the reflection symmetry $\sigma$ (and also translation symmetry). It seems that we cannot use an open chain system to study its boundary modes (see words below Eq. (10)). However, since the open chain construction is derived from closed chains, and thus have the same bulk properties as closed chain SPT phase, it is then OK to study this open chain system. We add a sentence to address this point in the main text. Also, to address the importance of $\sigma$ in this example, we add a sentence in the paragraph after Eq.(1), which states that fractional spin is not well-defined without reflection symmetry.

  2. This is a good point. We add a paragraph in the honeycomb subsection to address this point.

  3. Actually, the entanglement argument only relies on magnetic inversion, but not translation symmetry. We delete the part about translation symmetry. Sorry for the confusion caused.

  4. This is a good question. We add a sentence after this Kunneth decomposition to explain the physical meaning here.

---

## Round 2 · Referee Report · Anonymous (Referee 2) · 2021-7-19

Report

The authors have adequately responded to all the comments. I recommend the publication of this manuscript.

---

## Editorial Decision

published